# A facultative plasminogen-independent thrombolytic enzyme from *Sipunculus nudus*

Mingqing Tang[1,11], Guoxing Ma [1,2,11], Chunyan Xu[3,11], Hui Yang[4,11], Hongjun Lin[1,11], Chao Bian [5], Chengjia Hu[1], Meiling Lu[1], Lei Chen [1], Wencai Jie[6], Zhen Yue [3], Jianbo Jian [6], Yuqing Sun[1], Hui Yan[1], Jingjing Zhou[1], Xianying Zhang[1], Shengye Liao[1], Zhaofa Li[1], Shuangfeng Cai[1], Yaqing Wu [1], Kexin Yang[1], Yanan Xiong[1], Yonggang Zhao[1], Zhimin Lv[1,7], Xiaoming Xu[1,7], Chuang Liu[8], Pengliang Xin[9], Lichao Ye[10], Xiuling Cui [1] ✉, Qiong Shi [5] ✉, Xi Chen [4] ✉ & Ruian Xu [1,7] ✉

Current thrombolytic therapies primarily function by converting plasminogen into plasmin, a process dependent on the fibrin–activator complex. This dependence, coupled with the substantial molecular size of plasmin, constrains its effectiveness in degrading D-dimer and restricts its diffusion within thrombi. Here, we introduce a small facultative plasminogen-independent thrombolytic enzyme, snFPITE, isolated from *Sipunculus nudus.* Compared to traditional thrombolytic agents, snFPITE does not require plasminogen for thrombolysis, although its presence enhances lytic activity. This enzyme fully degrades cross-linked fibrin without leaving residual nondegradable D-dimer and generates a smaller fibrinolytic-active agent from plasminogen. A series of male rats and mice models further confirm that snFPITE is a safety injectable thrombolytic agent. Mechanistically, snFPITE activates plasminogen and degrades fibrin(ogen) in a multisite cleavage manner. snFPITE is inhibited by plasminogen activator inhibitor 1 and α2-antiplasmin via a competitive inhibition. We further identify 28 snFPITE candidate sequences, of which 10 are confirmed as functional genes.

Thrombotic disease is a leading health threats globally, especially in the post-COVID-19 era[1,2]. A plasminogen activator, such as tissue-type plasminogen activator (tPA), is commonly used in first-line thrombolytic treatment[3]. However, its effectiveness relies on the readiness of plasminogen[4,5]. In most reported cases of thrombosis, their internal plasminogen are found to be under-expressed, mutated, or altered, thereby impairing the formation of the classical plasminogen–tPA–fibrin ternary fibrinolytic complex[6]. Moreover,

[1]Engineering Research Centre of Molecular Medicine of Ministry of Education, Fujian Key Laboratory of Molecular Medicine, Key Laboratory of Precision Medicine and Molecular Diagnosis of Fujian Universities, Xiamen Key Laboratory of Marine and Gene Drugs, School of Biomedical Sciences, Huaqiao University, Xiamen, Fujian, China. [2]Department of Life Sciences, Tangshan Normal University, Tangshan, Hebei, China. [3]BGI Research, Sanya, Hainan, China. [4]Key Laboratory of Synthetic and Natural Functional Molecule of the Ministry of Education, College of Chemistry and Materials Science, Northwest University, Xi'an, Shanxi, China. [5]Laboratory of Aquatic Genomics, College of Life Sciences and Oceanography, Shenzhen University, Shenzhen, Guangdong, China. [6]BGI Genomics, Shenzhen, Guangdong, China. [7]Xiamen Institute of Medicine and Technology, Xiamen, Fujian, China. [8]The First Affiliated Hospital of Zhengzhou University, Zhengzhou, Henan, China. [9]Department of Haematology, Quanzhou First Hospital Affiliated to Fujian Medical University, Quanzhou, Fujian, China. [10]Department of Neurology, The Second Affiliated Hospital, The Second Clinical Medical College, Fujian Medical University, Quanzhou, Fujian, China. [11]These authors contributed equally: Mingqing Tang, Guoxing Ma, Chunyan Xu, Hui Yang, Hongjun Lin. ✉e-mail: cuixl@hqu.edu.cn; shiqiong@szu.edu.cn; xchen@nwu.edu.cn; ruianxu@hqu.edu.cn

forming such fibrinolytic complexes is a finely orchestrated multistep process[4,5,7]. The capacity of each step and the interaction among steps determine the final thrombolytic efficacy. Therefore, substantial efforts have been taken to simplify tPA-mediated thrombolysis. Theoretically, plasmin should be an ideal candidate for thrombolysis treatment if various antiplasmins do not rapidly quench the free plasmin. To overcome the current challenging problems of plasmin, such as its complicated structure, large molecular size, and less-stability, plasmin has been shortened to microplasmin at the cost of fine regulation[8]. Although some fibrinolytic enzymes exhibit both tPA and plasmin activities, such as lumbrokinase, conventional plasmin remains the activated product of plasminogen[9].

Another challenging problem is the formation of a large amount of fibrin degradation products. For example, D-dimer is a non-degradable terminal product, which is formed during tPA-and plasmin-mediated thrombolysis. Elevated D-dimer levels increase blood viscosity and exert a negative feedback on thrombolysis[10]. Although many drugs have successfully decreased D-dimer levels, only rare could efficiently degrade the formed D-dimer[11]. Therefore, thrombolytic agents with other clot lysis manner are highly desired. Consequently, an ideal thrombolytic agent should feature direct fibrinolytic activity, ability to activate and convert plasminogen into smaller fibrinolytic fragments rather than plasmin, and lysing clots without producing the D-dimer.

Previous reports have demonstrated that lumbrokinase from various earthworms has functioned as the orally administered thrombolytic drug[9]. A recent study highlighted that a popular marine worm-based snack (Tu sun dong) in South Fujian was helpful for digestion[12]. It was deduced that similar or even more complicated thrombolytic agents might exist in marine worms under the unique coastal environment. Therefore, we back on the small facultative plasminogen-independent thrombolytic enzyme (snFPITE) from a marine worm, *Sipunculus nudus*, also known as peanut worm. According to the fibrinolysis results obtained, snFPITE was found to activate and convert plasminogen into smaller and simpler fibrinolytic-active agent (Flaa) rather than conventional plasmin. Compared with traditional thrombolytic agents, snFPITE did not require plasminogen for thrombolysis but enhanced lytic activity in its presence. The X-ray crystal structure analysis revealed that snFPITE activated plasminogen and degraded fibrin(ogen) in a multisite cleavage manner. Plasminogen activator inhibitor 1 (PAI1) and α2-antiplasmin (A2AP) inhibited snFPITE through a competitive inhibition. Additionally, sequencing data unveiled that up to 28 putative *snFPITE* sequences existed in the *S. nudus* genome. Up to 10 sequences were confirmed as well functional genes by three different recombinant expression systems.

## Results

### snFPITE activated plasminogen and degraded fibrin(ogen) efficiently

The snFPITE protein sample was isolated from the fresh intestinal fluid of *S. nudus* (Fig. 1a) through chromatography. Native/SDS-PAGE, high-resolution mass spectrometry (HRMS), and ultrahigh-performance liquid chromatography (UHPLC) unveiled that snFPITE was a small monomer protein with a molecular weight of 24.925 kD (Fig. 1b–d and Supplementary Fig. 1a). This small monomeric feature of snFPITE is anticipated to enhance its permeability and penetrative capacity, potentially facilitating more efficient and rapid clot lysis compared to conventional plasmin (90–93 kD). Using the modified plasminogen-free fibrin plate, we demonstrated that the plasminogen-activating and fibrin-degrading efficacies of snFPITE exceeded those of recombinant tPA (rtPA) or plasmin at equivalent concentrations (Fig. 1e, f). Further fibrin and fibrinogen gel zymography revealed that snFPITE completely degraded the Aα, Bβ, and γ chains of fibrin and fibrinogen within 3 h (Fig. 1g). Thus, snFPITE appears to function as a hybrid of

plasmin and tPA, demonstrating effective activity in the absence of plasminogen, while exhibiting enhanced efficacy in its presence.

The enzymatic characteristics of snFPITE were also examined. Surprisingly, snFPITE exhibited excellent thermostability compared with any known thrombolytic drugs in use[13]. No significant reduction in the fibrinolytic activity of snFPITE was observed following 90 days of incubation at 37 °C (Supplementary Fig. 1b, c) or 96 h of incubation at 40 °C (Supplementary Fig. 1d). Additionally, snFPITE displayed a relatively wide adaptability across pH range of 5.0–9.0 (Supplementary Fig. 1e).

### snFPITE effectively lysed blood clots

In a study using rats with FeCl₃-induced common carotid arterial thrombosis, we found that both snFPITE and rtPA significantly improved the carotid arterial blood flow compared with the negative saline group (Fig. 2a, $p < 0.05$). The mouse tail thrombus model also reconfirmed the potent antithrombosis effects of snFPITE at a dose as low as 0.9 mg/kg when compared to the negative saline group (Fig. 2b, $p < 0.05$). Then, the clot lysis curve was established by conducting the whole blood halo assay (Fig. 2c, d). A shorter activation time ($A_t < 10$ min) was obtained for snFPITE at different doses compared with rtPA, indicating immediate initiation of thrombolysis (Fig. 2e). Such quick initiation of snFPITE-mediated clot lysis was possible because of the direct fibrinolytic activity of this enzyme. The thrombolysis rate of snFPITE declined after 20 min. Although no significant differences were observed in the maximal clot lysis rates ($CLR_{max}$) between snFPITE and rtPA (Fig. 2f), snFPITE exhibited a prolonged 50% lysis time (Fig. 2g; $T_{0.5} \approx 120$ min).

In the majority of thrombosis, plasminogen is prone to either underexpressed, mutated, or exhibits a modified conformation, thereby impeding the formation of the plasminogen−tPA−fibrin complex[6]. Even worse, the blood flow often stagnates at the clot site of thrombosis. Consequently, tPA-mediated plasminogen activation would be further restricted. snFPITE could be therefore more advantageous than current PAs for thrombolysis, especially under a defective plasminogen status (Supplementary Fig. 2a). In addition to its clot lysis effects, snFPITE exerted a pro-fibrinolytic influence on the coagulation system, as evidenced by the results of four coagulation assays conducted in a mouse model. The indices (prothrombin time, activated partial thrombin time, and thrombin time) of healthy mice were markedly prolonged after snFPITE treatment when compared with the saline group. Moreover, snFPITE exerted a better protective effect on thrombosis than rtPA (Supplementary Table 1). The improvement in the four coagulation tests suggested a potential benefit of the direct fibrinolytic activity of snFPITE, warranting further investigation in future studies.

D-dimer is a stable degradation product of clotted fibrin, typically present during plasmin-mediated fibrinolysis[14]. Elevated D-dimer concentrations in the blood serve as an indicator of thrombosis, facilitate thrombosis, and impede thrombolysis[10]. When blood samples with high D-dimer levels were co-incubated with snFPITE, snFPITE significantly decreased the D-dimer levels, although underlying reasons remain to be further explored (Supplementary Table 2). This D-dimer-degrading phenomenon might imply that snFPITE should have a broader substrate specificity than plasmin, although plasmin hydrolyses various peptide substrates[15,16]. Thus, the action of snFPITE on major other mammalian proteins was tested. The blood biochemical analysis results showed that snFPITE slightly decreased the levels of total protein and albumin in the mouse serum both in vitro (Supplementary Fig. 2b) and ex vivo (Supplementary Table 3), but not in in vivo models (Supplementary Table 4). The decreasing degradation effect of snFPITE on serum proteins in in vivo models might be an outcome of the complicated environment of the blood flow system. A similar degradation trend of collagen was also observed (Supplementary Fig. 2c). Moreover, a similar trend of rtPA-mediated degradation of

serum total protein and albumin was noted in this study, which was consistent with the substrate-degrading activity of plasmin reported in a previous study. In line with all the aforementioned findings and the detection of moderate trypsin activity (Supplementary Table 5), we hypothesize that snFPITE is a special thrombolytic enzyme with broader and moderate protease activities and may be beneficial for the food digestion. Such broader protease activities of snFPITE may not only strengthen its thrombolytic properties, could also raise potential bleeding issues, which will need to be carefully assessed in future exploration.

### snFPITE is an injectable thrombolytic agent

Intravenous injection continues to be the preferred method for administering the majority of thrombolytic agents. Hence, a systematic analysis of the biodistribution, pharmacokinetics, and toxicity of snFPITE was conducted to evaluate its appropriateness for this route of administration. The experimental rats were intravenously injected with the Cy7-conjugated snFPITE. The biodistribution and pharmacokinetics of snFPITE were investigated in rats through fluorescence imaging of the living animal and ex vivo organ at certain time points. snFPITE was rapidly distributed in the circulation system within 10 min, and enriched in heart, liver, and kidney stepwise. However, no fluorescence signals were noted in the brain (Fig. 3a). Therefore, akin to

most proteins, snFPITE underwent primary metabolism and excretion via hepatic and renal pathways, and was unable to traverse the blood-brain barrier. Fluorescence data of blood samples indicated that the concentration of snFPITE peaked at 10 min post-administration, exhibited a marked decline by 30 min, and subsequently stabilized up to 8 h, aligning with the findings from imaging studies (Supplementary Table 6). The extended half-life of snFPITE in blood circulation might confer certain advantages in maintaining effective concentrations in the bloodstream.

Then, the toxicity of snFPITE was evaluated using 293 cell line and mouse models. The CCK8 assay results revealed no significant change in the viability of 293 cells even after treatment with 50 μg/mL of snFPITE (Supplementary Fig. 3a). In the mouse model, the mice were intravenously injected with snFPITE or rtPA for five days. Meantime, the body weight of the mice (Fig. 3b), amount of water drunk (Supplementary Fig. 3b), and food consumed (Supplementary Fig. 3c) were recorded daily, which revealed no significant differences among the experimental groups. Bleeding and coagulation indices were examined using tail bleeding and glass-based coagulation assays after five days of treatment. The results displayed that both snFPITE and rtPA marginally prolonged bleeding and coagulation times at high concentrations ($P > 0.05$). However, these effects were rapidly alleviated at lower concentrations (Supplementary Fig. 3d). Meanwhile, the vein blood

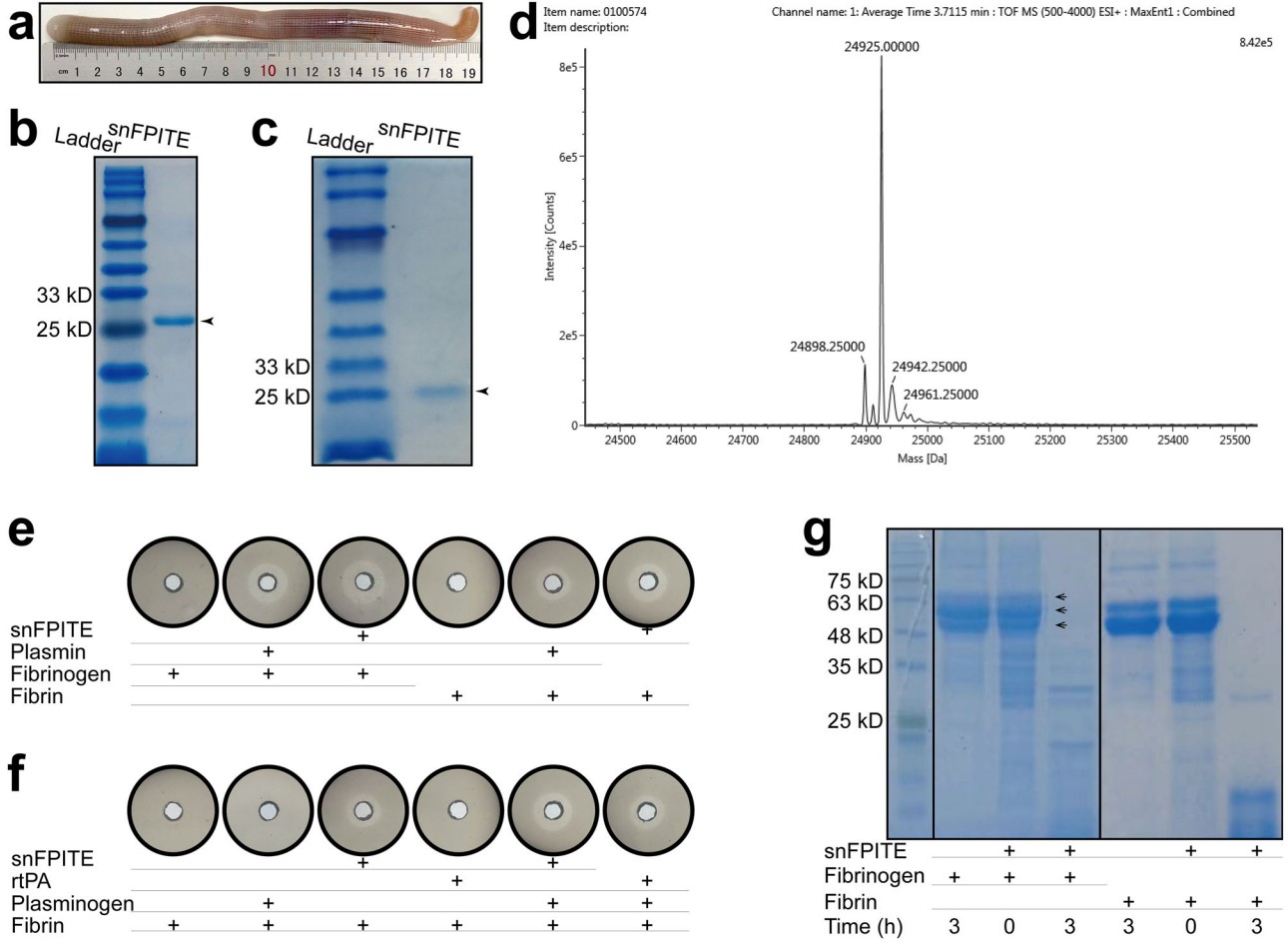

**Fig. 1 | Characterization of the snFPITE and thrombolysis potency tests. a** Fresh *S. nudus*. **b** SDS-PAGE of snFPITE. Arrow indicates snFPITE. snFPITE: 1 μg/well. **c** Native-PAGE of snFPITE. Arrow indicates snFPITE. snFPITE: 1 μg/well. **d** High-resolution mass spectrometry and ultrahigh-performance liquid chromatography of snFPITE. **e** Fibrin(ogen) degradation potency of snFPITE and plasmin. White halos indicate the lysis of fibrin(ogen). snFPITE and plasmin: 1 μg/well.

**f** Plasminogen-activating potency of snFPITE and rtPA. White halos indicate the activation of plasminogen. snFPITE and rtPA: 1 μg/well. **g** Time-dependent degradation products of snFPITE toward fibrin(ogen). Arrows indicate the Aα, Bβ, and γ chains of fibrin and fibrinogen. Fibrin(ogen): 2 μg/well, snFPITE: 0.05 μg/well. Four independent experiments were repeated with similar results (**b, c, g**). Source data are provided as a Source Data file.

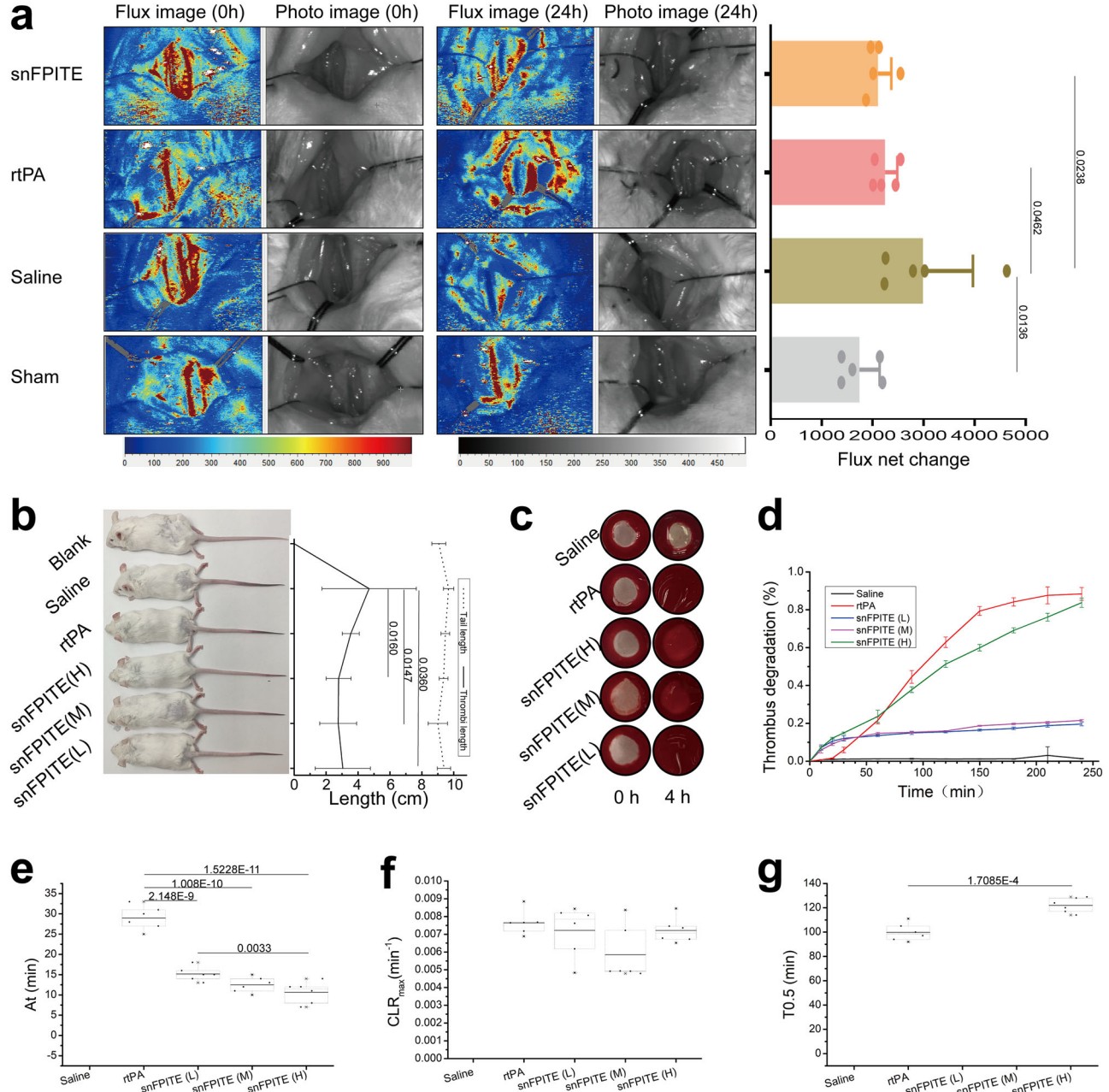

**Fig. 2 | Thrombolytic activity of snFPITE. a** Thrombolytic performance of snFPITE and rtPA on carotid arterial thrombosis model in rat. Changes in blood flux of the carotid artery before (left) and after (middle) exposure to FeCl₃. Net change in the blood flux of the carotid artery exposed to FeCl₃ (right). snFPITE and rtPA: 0.9 mg/kg. $n = 5$ independent experiments. **b** Thrombolytic performance of snFPITE and rtPA on a mouse tail thrombosis model. Carrageenan-induced tail thrombosis (left). The length of the whole tail and thrombus in mice (right). rtPA: 0.9 mg/kg, snFPITE (H): 0.9 mg/kg, snFPITE (M): 0.45 mg/kg, and snFPITE (L): 0.23 mg/kg. $n = 8$ independent experiments. **c** Blood-dissolving performance of snFPITE and rtPA on a murine blood halo model. rtPA: 72 ng/mL, snFPITE (H): 72 ng/mL, snFPITE (M): 36 ng/mL, snFPITE (L): 18 ng/mL. $n = 6$ independent experiments. **d** Thrombolytic profile of snFPITE and rtPA on a murine blood halo model. The percentage of degradation was obtained from normalization to the non-clotted blood sample. rtPA: 72 ng/mL, snFPITE (H): 72 ng/mL, snFPITE (M): 36 ng/mL, snFPITE (L): 18 ng/mL. $n = 6$ independent experiments. **e** Activation times ($A_t$) of snFPITE and rtPA. $n = 6$ independent experiments. **f** Maximal clot lysis rates ($CLR_{max}$) of snFPITE and rtPA. rtPA: 72 ng/mL, snFPITE (H): 72 ng/mL, snFPITE (M): 36 ng/mL, snFPITE (L): 18 ng/mL. $n = 6$ independent experiments. **g** 50% lysis ($T_{0.5}$) of snFPITE and rtPA. $n = 6$ independent experiments. Significance was calculated by repeated-measures one-way ANOVA followed by Tukey and LSD's multiple comparisons test (**a**, **e**), or by Kruskal–Wallis test followed by Dunn's multiple comparisons test (**b**). Data are presented as the mean ±SD (**a**, **b**, **d**). Box is bounded by the first and third quartile with a horizontal line at the median and whiskers extend to the maximum and minimum value (**e**, **f**, **g**). Source data are provided as a Source Data file.

was extracted, and one portion of this blood was used for examining routine blood parameters. No significant differences were observed among all experimental groups (Supplementary Table 7). Another portion of the collected vein blood was subjected to platelet activation detection and four coagulation tests. No effect of snFPITE on platelet activation was noted. However, the inhibitory effects of snFPITE were detected using flow cytometry (Fig. 3c), which might be attributed to its direct interaction with platelets or to alterations in the blood microenvironment that subsequently inhibited platelet activation. The values of prothrombin time, activated partial thrombin time, and

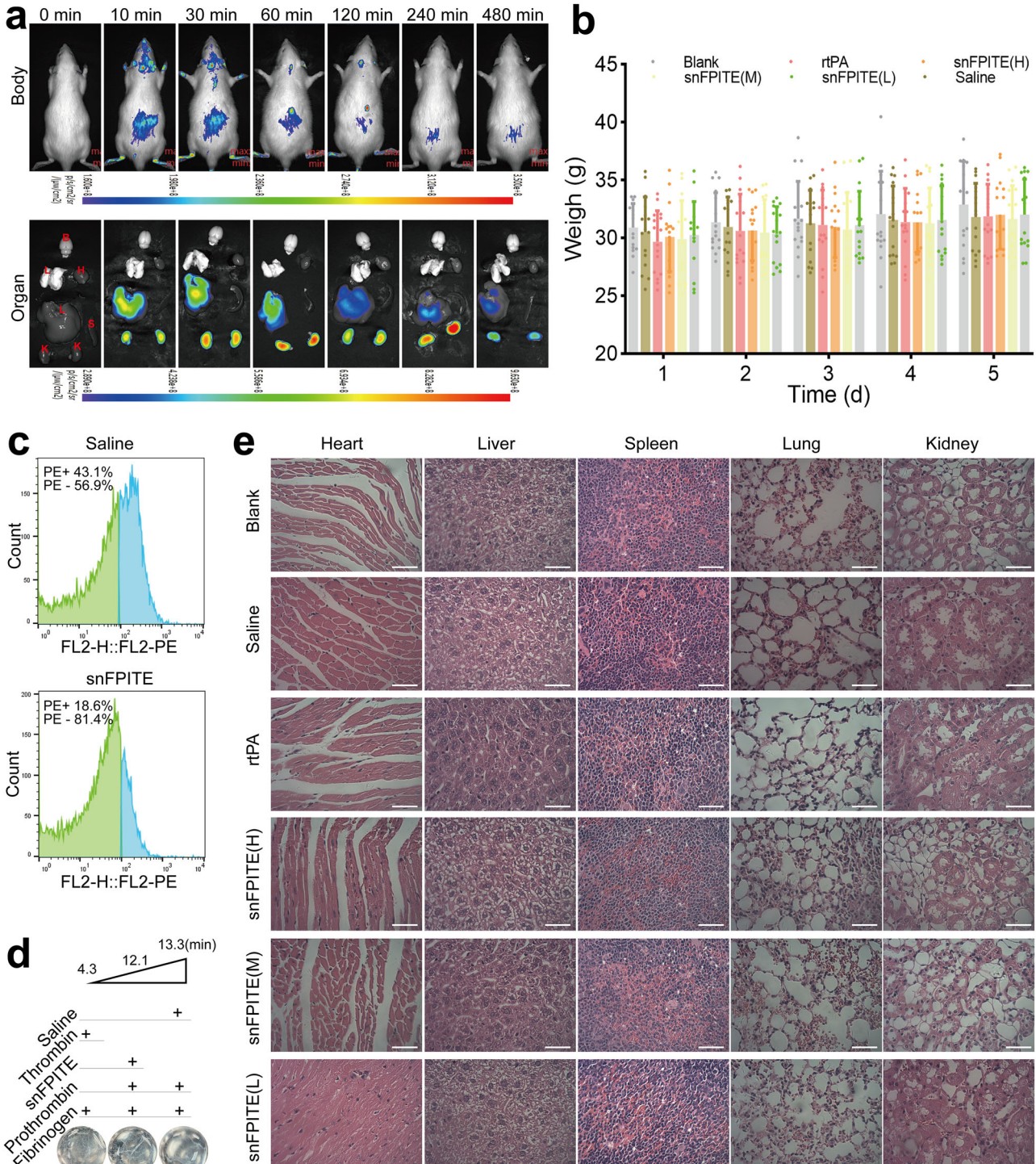

**Fig. 3 | Safety and pharmacokinetic analysis of snFPITE. a** Biodistribution of Cy7-conjugated snFPITE in a rat model. Fluorescence imaging of a live rat (top). Fluorescence imaging of dissected organs (bottom). Red letters (B, L, H, L, S, and K) represent the organs (brain, liver, heart, lung, spleen, and kidney, respectively). snFPITE: 5 mg/kg ($n = 3$). **b** Effects of snFPITE on the body weight of the experimental mice. rtPA: 0.9 mg/kg, snFPITE (H): 0.9 mg/kg, snFPITE (M): 0.45 mg/kg, and snFPITE (L): 0.23 mg/kg. Data were presented as the mean± SD; $p > 0.05$ by one-way ANOVA ($n = 16$ independent experiments). **c** snFPITE inhibited platelet activation in a mouse model. The green curve of flow cytometry represents the nonactivated

platelet (PE−). The blue curve of flow cytometry represents the activated platelet (PE + ) ($n = 6$). **d** snFPITE slightly promoted prothrombin activation. Numerical values (top) indicate the time of the first fibrin formed in the U-plate (bottom). Thrombin or prothrombin: 1 U/well, snFPITE: 0.1 µg/well, and fibrinogen: 400 µg/well. Data were presented as the mean ($n = 6$). **e** HE staining of the five major organs dissected from a mouse after treatment for five days. Magnification = ×400. rtPA: 0.9 mg/kg, snFPITE (H): 0.9 mg/kg, snFPITE (M): 0.45 mg/kg, and snFPITE (L): 0.23 mg/kg ($n = 6$). Scale bar: 50 µm. Three independent experiments were repeated with similar results (**e**). Source data are provided as a Source Data file.

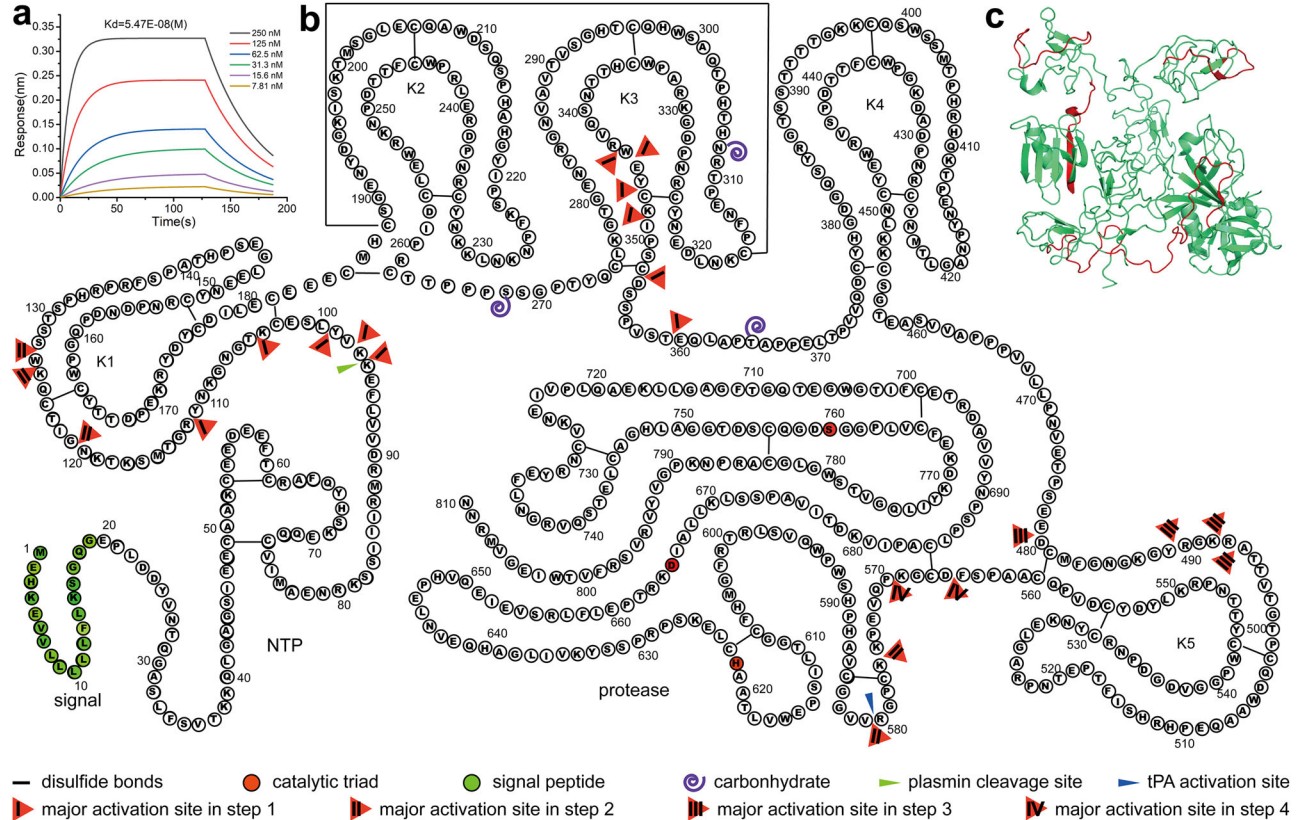

**Fig. 4 | Activation sites and process of snFPITE on plasminogen. a** Direct molecular interaction of snFPITE and plasminogen. Three independent experiments were repeated with similar results. **b** Schematic representation of the cleavage steps and sites of snFPITE on plasminogen. **c** Location of cleavage fragments (red) on plasminogen (green). Source data are provided as a Source Data file.

thrombin time increased, whereas the fibrinogen levels decreased. In the light of all the aforementioned findings, drawing a direct conclusion that snFPITE activates prothrombin requires more supporting evidence. Therefore, the co-incubation of snFPITE with the prothrombin complex was conducted next, and snFPITE was found to only slightly activate prothrombin (Fig. 3d). Furthermore, at the end of the treatment, samples from major organs, including the heart, liver, spleen, lungs, and kidneys, were collected and subjected to microscopic examination. Hematoxylin and eosin (HE) staining revealed no significant structural damage or inflammatory cell infiltration post-treatment (Fig. 3e). In summary, the safety profile of snFPITE was comparable to that of rtPA.

### snFPITE activated plasminogen and degraded fibrin(ogen) in a multisite cleavage manner

Initially, GatorPivot was used to assess whether snFPITE could bind to plasminogen. The low $K_d$ value (5.47E-08 M) suggested a direct interaction between snFPTTE and plasminogen (Fig. 4a). Additionally, protein sequencing of the co-incubation products of snFPITE–plasminogen showed that snFPITE could cleave plasminogen at up to 22 sites, including the plasmin's cleavage site Lys[96]-Lys[97] and tPA's activation site Arg[580]-Val[581] (Fig. 4b, c, and Supplementary Table 8). Moreover, snFPITE-mediated processing of plasminogen did not generate the normal plasmin, but a 26.83 kD free Flaa that was analogous to the protease domain of plasmin (Fig. 4b, Supplementary Fig. 4), implying the existence of a different plasminogen-activating pathway[5]. As Flaa was considerably smaller than plasmin, it might penetrate clots more efficiently, making lysis within the clots easier.

Following crystallization, an intact crystal structure of the fibrinolytic enzyme snFPITE (snFPITE-n1) was revealed at a high resolution

of 2.0 Å (PDB: 8ZVS, Fig. 5a, and Supplementary Table 9) along with its detailed sequence information (Supplementary Fig. 5a). Similarly, the second snFPITE crystal structure (snFPITE-n2) was also revealed at a higher resolution of 1.5 Å (PDB: 8ZVX, Fig. 5b, and Supplementary Table 9), and its detailed sequence information was obtained (Supplementary Fig. 5a). Moreover, snFPITE-n2 consisted of two molecules (A and B) in an asymmetric unit. snFPITE-n2-B (molecule B) was the same as snFPITE-n1, but 28 amino acids differed between molecules A and B. These two molecules formed a heterodimer through hydrogen bonding with two separate catalytic triads available for substrates, indicating that snFPITE-n2 could function comparably to snFPITE-n1. This unique structure of snFPITE-n2 might shed light on preventing the loss of activity through dimerization or polymerization during freeze–thaw cycles. Prior to crystallization, the protein samples were treated with the serine protease inhibitor phenylmethylsulphonyl-fluoride (PMSF). This treatment facilitated the clear observation of electron densities corresponding to the PMSF molecule covalently bonded to Ser[189] of snFPITE-n1, snFPITE-n2-A, and snFPITE-n2-B within the PMSF-snFPITE-n1/n2 cocrystal structures (Supplementary Fig. 5b–d).

Based on protein sequence alignment, the closest homolog of snFPITE-n1 was found to be the fibrinolytic enzyme from *Enchytraeus japonensis* (44% identity). Furthermore, the most structurally similar protein with a known crystal structure of snFPITE-n1 was the earthworm fibrinolytic enzyme component A (EFEa, 41% identity; PDB: 1M9U) (Supplementary Fig. 5e). However, based on the relatively low-root-mean-square deviation (RMSD) value (0.575-Å) between snFPITE-n1 and EFEa, it was inferred that both enzymes shared crucial structures and functions. Furthermore, both these proteins belonged to the serine protease family, with the spherical characteristic and

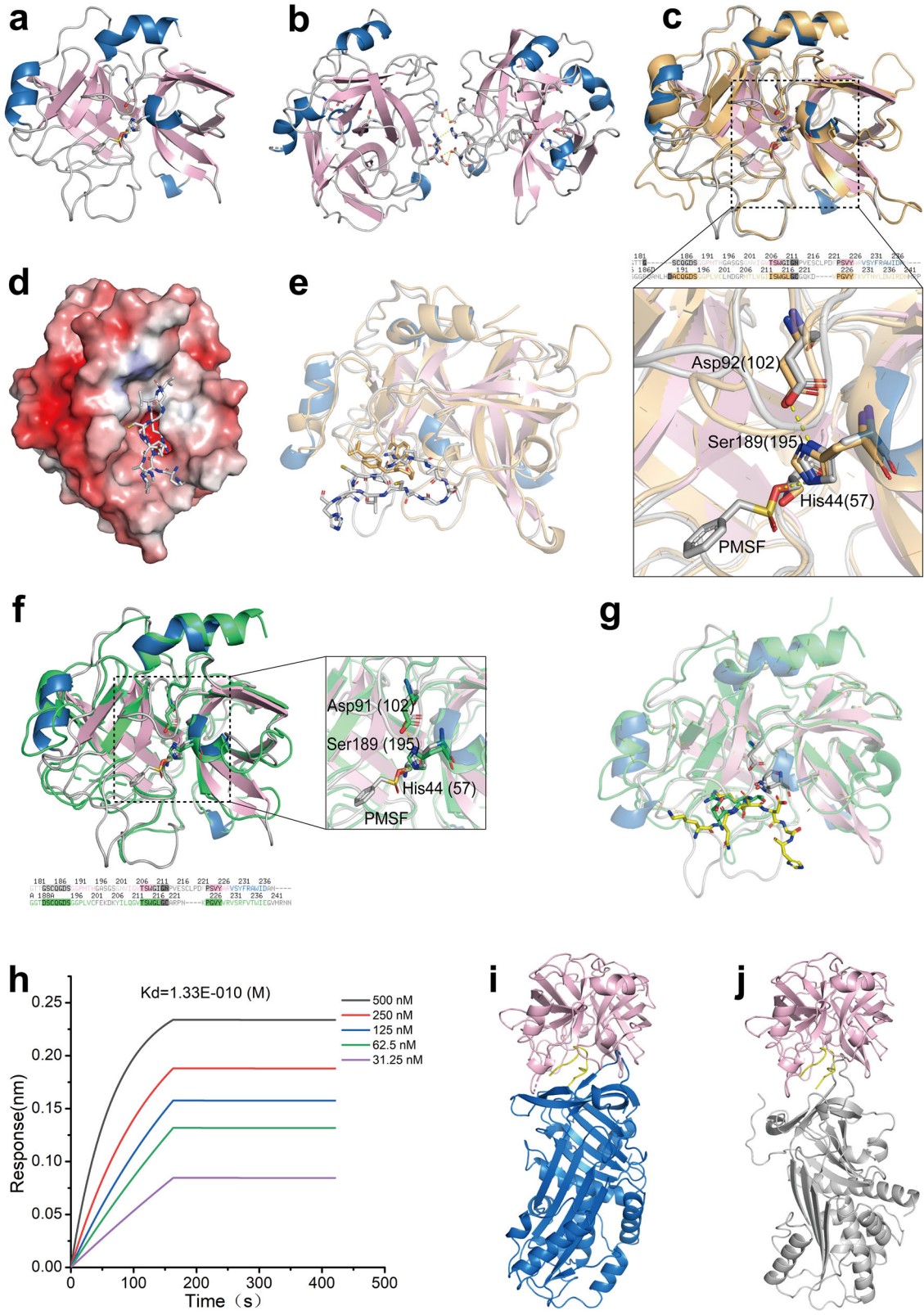

six-stranded barrel-like folds connected by three trans-domain straps[17]. For EFEa and related proteins, the catalytic triads within this family are composed of His[57], Asp[102], and Ser[195]. In contrast, the analogous residues in snFPITE-n1 were identified as His[44], Asp[91], and Ser[189]. Notably, both Asp[91] and Ser[189] established hydrogen bonds with His[44], a feature that is essential for facilitating the catalytic interaction (Supplementary Fig. 5f).

Further, snFPITE-n1 and tPA (PDB: 1A5H) structures were compared to clarify how snFPITE activates plasminogen (Fig. 5c). In general, the RMSD value was 0.824 Å between the two structures and the catalytic triads almost overlapped. These S1 pocket-building residues were highly conserved between the two enzymes. Asp[189] at the S1 pocket's base in tPA with a negative charge, which allowed salt bridge formation with basic residues, was crucial for the binding to tPA

**Fig. 5 | Molecular mechanism of snFPITE-mediated thrombolysis. a** Crystal structure of snFPITE-n1. The α-helices of the structure are indicated in blue, the β-sheets in pink, and the loops in gray. **b** Crystal structure of snFPITE-n2. The α-helices of the structure are indicated in blue, the β-sheets in pink, and the loops in gray. **c** Structure of snFPITE-n1 (shown as a pink cartoon) superimposed with tPA (PDB: 1A5H, shown as a green cartoon). **d** Model of snFPITE-n1 in complex with a canonical human plasminogen peptide 576-KCPGRVVGGCAH-588 (gray sticks) showed that there was no steric clash between them. **e** Model of snFPITE-n1 superimposed with the tPA in complex with an inhibitor (PDB: 1A5H). **f** Structure of snFPITE-n1 superimposed with plasmin (PDB:1BUI, shown as an orange cartoon).

**g** Structure of snFPITE-n1 superimposed with plasmin in complex with a peptide mimic inhibitor (PDB: 1BUI). **h** Molecular interaction between snFPITE and fibrinogen. Three independent experiments were repeated with similar results. **i** Model of snFPITE-n1 in complex with PAI1 and plasminogen peptide. PAI1 (blue) blocked the way such that the plasminogen peptide 576-KCPGRVVGGCAH-588 (yellow) was bound to the snFPITE-n1 (pink). **j** Model of snFPITE-n1 in complex with A2AP and fibrinogen peptide. A2AP (gray) competed with the fibrinogen peptide 576-KCPGRVVGGCAH-588 (yellow) bound to the snFPITE-n1 (pink). Source data are provided as a Source Data file.

substrate Arg[18]. The counterpart for this residue in snFPITE-n1 was Gly[183], which was small and nonpolar and should easily embrace a Lys or Arg residue theoretically. This was consistent with the fact that Lys or Arg inhibited snFPITE-n1 (Supplementary Fig. 6a) and Arg-Sepharose 4B (Supplementary Fig. 6b-left) and Lys-Sepharose 4B affinity-purified it (Supplementary Fig. 6b-right). The S2 pocket of tPA contained Tyr[99], leading to a preference for small residues theoretically, such as Gly at the P2 position[18]. The corresponding residue in snFPITE-n1 was Ile[88], which was also compatible with a small residue theoretically. Furthermore, the model of snFPITE-n1 complexed with a canonical human plasminogen peptide displayed that a steric clash between the peptide and snFPITE-n1 was absent (Fig. 5d), suggesting that this peptide could be an appropriate substrate for snFPITE-n1. This peptide was bound to snFPITE-n1 with the Arg[580] residue inserted into the negatively charged pocket, as in tPA. Moreover, this model was also superimposed with the tPA complexed with an inhibitor (PDB: 1A5H), and the peptide substrate overlapped well with this inhibitor (Fig. 5e). Similarly, up to 12 cleavage sites of snFPITE on plasminogen were further corroborates with the corresponding models (Supplementary Fig. 6c). Thus, snFPITE-n1 activated plasminogen into Flaa rather than the conventional plasmin in a multisite cleavage manner. Compared with the high-site specificity of tPA, the loss of site specificity snFPITE might mainly be caused by the lack of corresponding F, E, K1, and K2 domain structures of tPA, which were critical for target selection[19]. Meanwhile, bulkier P1 residues (such as Cys, Phe, Tyr and Trp) could be inserted into the S1 pocket because of the key small residue Gly18 in this pocket, along with the basic residues Arg or Lys. Moreover, Ile[88] in the S2 pocket embraced bigger P2 residues other than Gly for tPA (Fig. 5c). Furthermore, four additional residues (Cys, Leu, Pro, and Asp) were inserted into the S1 pocket, which might extend this pocket and provide an additional substrate-binding site[17].

Next, the mechanism underlying the direct lysis of fibrin(ogen) caused by snFPITE was tried to clarify. A model of snFPITE-n1 superimposed with plasmin (PDB: 1BUI, RMSD value: 0.828 Å) overlapped with the catalytic triads and a similar S1 pocket (Fig. 5f). Meanwhile, six snFPITE-n1 models with six 7-mer peptide substrates from α, β, and γ chains of fibrinogen revealed the absence of steric clash between snFPITE-n1 and these peptide substrates, suggesting that snFPITE-n1 could accommodate substrates from the fibrinogen chains (Supplementary Fig. 7a). In all these models, the Lys or Arg residue was buried in the negatively charged pocket. The structure of snFPITE-n1 containing one peptide also superimposed with plasmin complexed with a peptide mimic inhibitor (PDB: 1BUI), and the peptide overlapped well with the inhibitor (Fig. 5g). Moreover, the Quartz Crystal Microbalance (Supplementary Fig. 7b) and GatorPivot (Fig. 5h, $K_d = 1.33E\text{-}10$ M) results further revealed that snFPITE-n1 directly bound to fibrinogen. Plasmin did not exhibit strict target selectivity because its cleavage sites were lenient[15,16]. However, it could not further dissolute the D-dimer completely. Thus, D-dimer digestion indicated that snFPITE-n1 had a wider range in terms of target selectivity. The lack of a corresponding N-terminal pre-activation peptide domain and five kringle domains (K1–K5) in plasmin might resulted in the loss of target selectivity. These domains were crucial for target selection and regulation implied by previous researches[16]. Meanwhile, the pockets of

snFPITE-n1, which were less charged than those of tPA and plasmin, were compatible with the broader substrate selectivity of this enzyme (Supplementary Fig. 7c).

## PAI1 and A2AP inhibited snFPITE through a competitive inhibition

PAI1 and A2AP are primary inhibitors of tPA and plasmin, respectively, and follow the same single-use "suicide" inhibition[18,19]. Here, we found that (1) PAI1 had no effect on the cleavage of plasminogen, but slightly impaired the generation of Flaa; (2) PAI1 inhibited the degradation of fibrin(ogen); (3) A2AP inhibited the cleavage fibrin(ogen) and plasminogen; and (4) PAI1 and A2AP mainly suppressed the cleavage of Bβ and γ chains rather than Aα of fibrin(ogen) (Supplementary Fig. 8). This slight inhibitory effect of PAI1 was consistent with its large $Ki$ value (Supplementary Fig. 9a, b). Compared with tPA and plasmin, snFPITE formed no irreversible covalent snFPITE–PAI1 or snFPITE–A2AP complex, but digested A2AP (Supplementary Fig. 9c) in an unknown manner. In this context, A2AP seemed to be a substrate of snFPITE. Therefore, the snFPITE-n1–PAI1 and snFPITE-n1–A2AP models were therefore established. Six hydrogen bonds were found between snFPITE-n1 and PAI1, indicating the occurrence of moderate interactions (Supplementary Fig. 9d). When the structure of snFPITE-n1 complexed with PAI1 was compared with the model of snFPITE-n1 complexed with a peptide substrate, PAI1 was found to obstruct the binding of the peptide substrate to snFPITE-n1 (Fig. 5i). Given the moderate interaction between PAI1 and snFPITE, it was plausible that PAI1 could inhibit the function of snFPITE specifically in the generation of Flaa, while not affecting its ability to cleave plasminogen.

Meanwhile, eight hydrogen bonds were present at the interface between snFPITE-n1 and A2AP (Supplementary Fig. 9e). A stable plasmin–A2AP complex might be formed by a plasmic attack at the reactive center loop (RCL) of A2AP (Arg[376]-Met[377]). A covalent bond was formed between the active site seryl residue (Ser[741]) in plasmin and the carbonyl group of this specific residue (Arg[376]) in A2AP[20,21]. However, the catalytic Ser[189] of snFPITE-n1 was found not to form a covalent bond with the Arg[376] of A2AP. The RCL cleavage-triggered conformational change or the competition between RCL and the fibrin(ogen) substrate might render snFPITE-n1 inactive (Fig. 5j).

In circulation, A2AP can rapidly inactivate free active plasmin, which is a major obstacle in thrombolytic treatment involving direct plasmin use[22]. However, A2AP showed some inhibitory effect on snFPITE. This inhibition did not directly involve plasmin. Moreover, the effectiveness of A2AP was inversely related to the presence of fibrin. A2AP competed with fibrin to inhibit snFPITE; hence, higher fibrin levels resulted in reduced A2AP potency. Therefore, the manner of interaction of snFPITE with PAI1 and A2AP might endow this enzyme with a lower plasma–protein binding ability, a markedly longer half-life time in blood circulation as mentioned earlier, and some fibrin(ogen)-sensing intelligence.

## Identification of 28 candidate snFPITE sequences in *S. nudus*

The presence of snFPITE-n1 and snFPITE-n2 implied that more snFPITE sequences might exist in *S. nudus*. Subsequently, 10 more putative *snFPITE*-coding sequences (*snFPITE-c1* to *snFPITE-c10*) were identified

through molecular cloning based on the snFPITE-n1, snFPITE-n2, and transcriptome sequences (Supplementary Table 10). However, the deduced protein sequences of other candidate snFPITE sequences exhibited no match with snFPITE-n1 or snFPITE-n2, except for snFPITE-c2, which had an identical protein sequence to that of snFPITE-n1. The whole-genome and transcriptome sequencing were performed, and a high-quality genome assembly was achieved (NCBI BioProject Accession: PRJNA997755, CNGB Accession: CNP0003708, Supplementary Fig. 10a–d, and Supplementary Tables 11–18). Obviously, this genome had a high heterozygosity (-3.1%; Fig. 6a). A phylogenetic tree indicated that *S. nudus* was a sister group to two other Annelida species (*Capitella telet* and *Helobdella robusta*) with a low evolution rate (Supplementary Fig. 10e). Moreover, it diverged from both species approximately 595–650 million years ago (Fig. 6b), which was a little earlier than the oldest fossil record[23]. A marked enrichment of the complement and coagulation cascade pathways was noted, indicating the crucial role of these special fibrinolytic enzymes in *S. nudus* (Fig. 6c, Supplementary Table 19, and Supplementary Data 3).

Subsequently, an additional 16 candidate *snFPITE* genes were identified within the assembled genome, including one putative pseudogene. The proteins encoded by these genes consistently exhibited the conserved GDSGGP motif, Among them, seven proteins also contained the conserved AAHC motif, while four proteins displayed the IVGG, AAHC, and GDSGGP conserved sequences. Notably, the AAHC and GDSGGP motifs were co-located at the same active site, underscoring their conserved functional significance (Supplementary Fig. 11). Nonetheless, none of the deduced protein sequences from the *snFPITE* candidate genes precisely matched the previously identified 12 snFPITE-coding sequences (99.61% for the highest identity; Supplementary Table 20). Such low identity might be attributed to the high heterozygosity of the sequenced genomes and variable processing at the transcriptional, translational, and/or post-translational levels, based on our previous study on cone snails[24]. Five candidate *snFPITE* genes with tandem duplication in the scaffold332 were also observed (Fig. 6d), suggesting a clustered expansion of *snFPITE* genes in the *S. nudus* genome with more vital roles. Moreover, a homology search (identity > 30%, *e* value < 1e-10) demonstrated the presence of three homologous genes (*snPAI1 1-3*) for *PAI1*, one (*snAP*) for *A2AP*, one (*sntPA*) for *tPA*, and one (*snuPA*) for urokinase-type plasminogen activator (*uPA*). However, the deduced sntPA and snuPA sequences were only partially aligned and exhibited low similarity to references from other examined animals (Supplementary Fig. 12). This agreed well with the direct fibrinolytic activity of snFPITE.

A phylogenetic analysis of 17 snFPITEs and 237 other representative plasminogens/fibrinolytic enzymes (Supplementary Data 4 and 5) showed that they were primarily clustered into four clades (Fig. 6e). The snFPITE enzymes within one branch were the closest to earthworms. This indicated that these enzymes might have more similar functions and evolutionary trajectories, which are consistent with the dual fibrinolytic activities in *Eisenia fetida*[25].

### Identification of 14 putative *snFPITE* transcripts in *S. nudus* transcriptomes

The transcriptional profiles of various candidate *snFPITE* genes were assessed *via* transcriptome sequencing across diverse tissues subjected to different stimuli. Ultimately, 14 full-length transcripts associated with snFPITE were identified (Supplementary Tables 21 and 22), with genomic evidence supporting nine of these transcripts (Supplementary Table 23). Additionally, eight *snFPITE* genes were highly transcribed in intestinal, ganglion, and protractor cells, indicating a marked tissue-specific expression pattern. Moreover, three genes were specifically stimulated by exogenous ammonium salt, blood clot, and sulfide, while five genes were exclusively stimulated by a blood clot (Fig. 7a). Further, snPAI1 1–3 and snAP exhibited high expression levels in ganglion, intestinal, and nephridial cells. However, their

transcription levels decreased in response to stimuli such as ammonium salt, blood clot, and sulfide, showcasing a negative correlation with *snFPITE* genes (Fig. 7b). The snFPITE-n1–snPAI and snFPITE-n1–snAP models further supported this regulatory network (Supplementary Fig. 13), exhibiting molecular interactions analogous to those of their homologous human PAI1 and human A2AP.

### Functional verification of 10 recombinant snFPITE proteins

One of the candidate *snFPITE* genes was expressed in the *Pichia pastoris* system, exhibiting fibrinolytic transient activity (Fig. 7c-top, Supplementary Fig. 14a). Additionally, eight other candidate *snFPITE* genes were transiently expressed in the HEK293 system, also exhibiting fibrinolytic activities (Fig. 7c-middle, Supplementary Fig. 14b), thereby indicating their identity as true *snFPITE* genes. Subsequently, two of these genes (*snFPITE-8* and *snFPITE-15*) were stably expressed in the HEK293 system (Fig. 7c-middle). Western blot analysis revealed the stable expression of both *snFPITE* genes (Supplementary Fig. 14c-bottom). Rat blood clot lysis results showed that all clots in each sample tube were completely dissolved, with the exception of those in the negative control group (Supplementary Fig. 14c-top). Further SDS-PAGE analysis revealed that snFPITE-8 and snFPITE-15 effectively degraded the Aα, Bβ and γ chains of fibrin, except for the negative control (Supplementary Fig. 14d). A subsequent validation experiment involving the *microplasmin* gene (*ocriplasmin*, *oPlm*) and the *snFPITE-n1* gene, both expressed in the Tn-baculovirus system (Fig. 7c-bottom), further confirmed the functionality of these *snFPITE* (Fig. 7d, Supplementary Fig. 14e).

## Discussion

Plasminogen can be directly or indirectly activated by various plasminogen activators. However, the plasminogen-mediated dissolution of clots depends on plasmin formation[16]. Compared with the known plasminogen activators, snFPITEs cleave plasminogen into fibrinolytic Flaa rather than the conventional plasmin. Flaa is considerably smaller than plasmin, endowing Flaa with a great advantage in mobility and penetration theoretically. The activated plasmin further accelerated plasminogen activation by changing the Glu-plasminogen to Lys-plasminogen[21]. The plasmin site was included in the activity range of snFPITE. However, whether such an accelerating effect exist in snFPITE needs verification. The Flaa sequence only lacks four N-terminal amino acids of oPlm, a truncated form of plasmin approved by the Food and Drug Administration for vitreomacular traction treatment in clinics[26]. The recombinant oPlm (Supplementary Fig. 15a) demonstrated similarities to snFPITE in various aspects, including molecular weight, direct fibrinolytic activities, and enzyme kinetic parameters (Supplementary Figs. 14e and 15b). This suggests that both enzymes shared functional characteristics that are critical for their roles in fibrinolysis.

In general, tPA and uPA can directly activate plasminogen predominantly through the Arg$^{580}$–Val$^{581}$ activation site. Alternatively, plasminogen is indirectly activated through other activators by forming an active activator–plasminogen complex, or its activation is accelerated by plasmin through the cleavage site of Lys$^{96}$–Lys$^{97}$[18,21]. In contrast, based on our results, a plasminogen activation pathway with multisite cleavage was established. The stepwise degradation and sequencing data of products validated that snFPITE-mediated activation of plasminogen mainly involved four stages and up to 22 sites. High-resolution crystal structures of both snFPITE-n1 and snFPITE-n2 reconfirmed that their catalytic triads consisted of His$^{44}$, Asp$^{91}$, and Ser$^{189}$, and the weak substrate specificity might be caused by Gly$^{183}$ and Ile$^{88}$ residues in the binding pockets S1 and S2, or the four additional residues (Cys, Leu, Pro, and Asp) inserted into the S1 pocket of snFPITEs. Overall, the charges of the substrate-binding pockets of snFPITEs were more prone to be neutral, which might be a potential cause of indiscrimination.

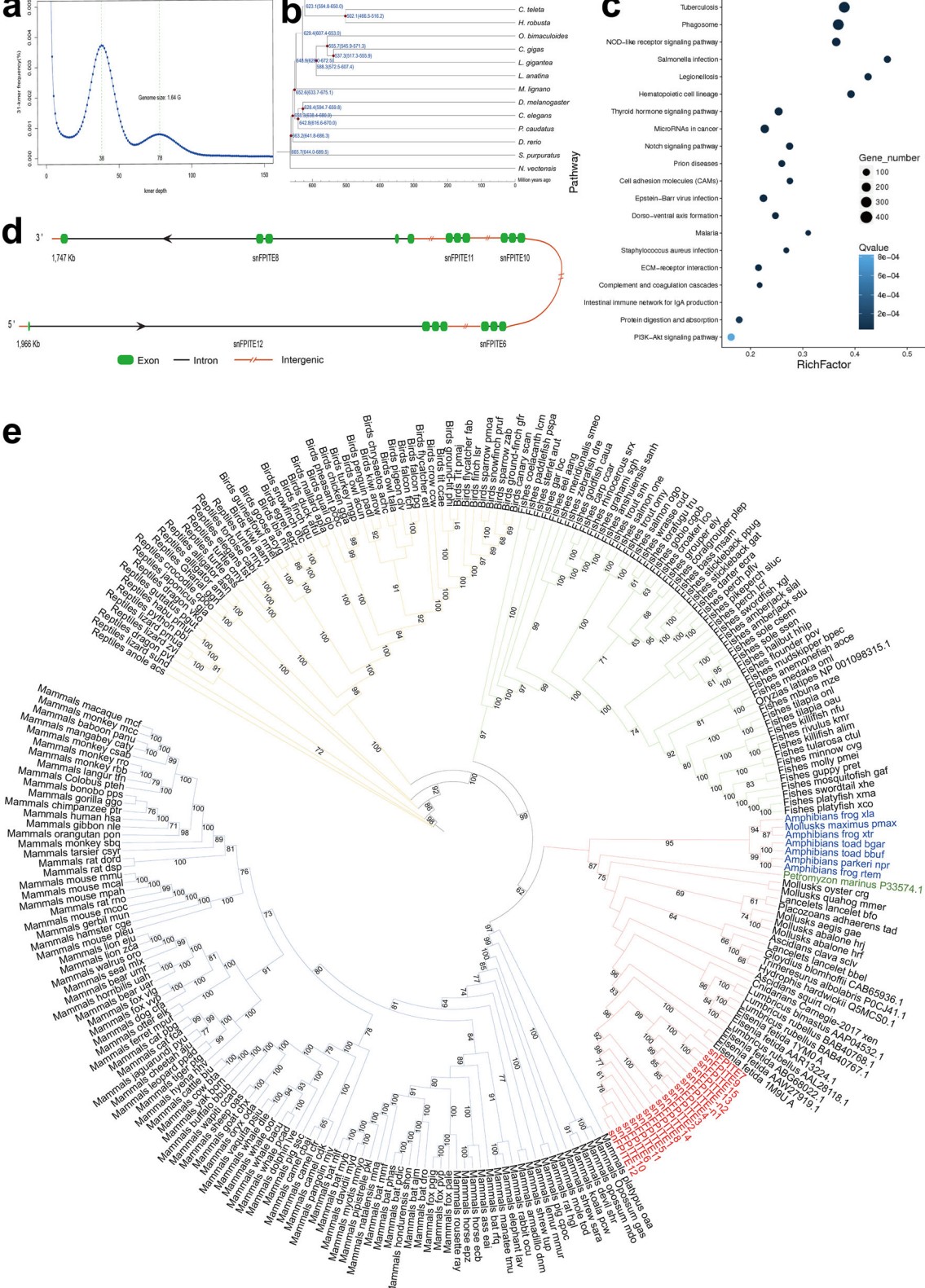

**Fig. 6 | snFPITE identified by genome sequencing and phylogenetic analysis.** **a** *S. nudus* genome has a high heterozygosity. **b** Estimation of the divergence time of *S. nudus* with 13 representative species based on orthologous relationships. Blue numbers at the nodes represent divergence times to the present (million years ago, Mya). Red dots represent the used calibration times, as derived from the TimeTree database. **c** *S. nudus* had enriched complement and coagulation cascade pathways. *Q* values represent the significance of enrichment. Circles indicate the target genes, and the size is proportional to the number of genes. **d** Tandem duplication of five *snFPITE* genes in the scaffold332. **e** Phylogenetic tree of the reported fibrinolytic enzymes. These genes were mainly clustered into four large clades (presented in different colors). The *snFPITE* genes of *S. nudus* are marked in red. The bootstrap values (>60) are indicated at each branch.

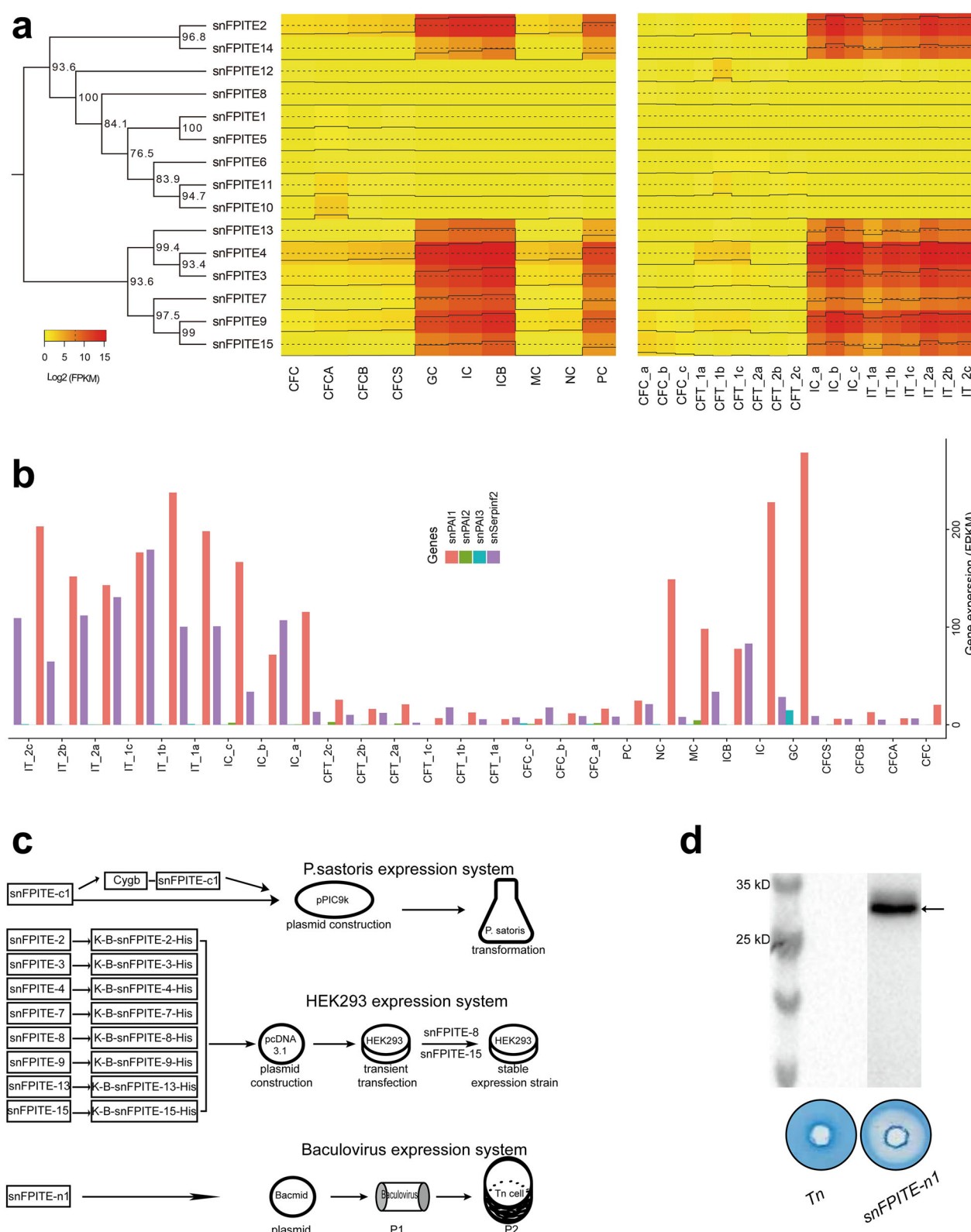

Dissolution of clots with plasmin or oPlm is a stepwise and complicated procedure that mainly occurs through hydrolysis at Arg and Lys sites to release various nondegradable fibrin degradation products such as D-dimer[21,27]. Although no direct clinical evidence is available to clarify whether high D-dimer levels substantially inhibit fibrinolysis, low D-dimer concentrations may result in rapid fibrinolysis through a feedback regulation. The present study demonstrated that snFPITEs

could degrade D-dimer. The findings might aid in ensuring a more efficient and rapid thrombolysis. In this study, the Quartz Crystal Microbalance and GatorPivot results also demonstrated direct binding between snFPITEs and fibrinogen. Well-overlapping catalytic triads and similar S1 pockets between snFPITEs and plasmin were determined based on the analyses of snFPITE crystal structures[16]. This structural similarity was also consistent with the wider substrate selectivity of

**Fig. 7 | Transcriptome analysis and functional validation of snFPITE. a** A transcription heat map of 15 *snFPITE* genes from 10 diverse tissue samples (left) and 18 treatment samples (right). CFC, coelomic fluid cell; CFCA, coelomic fluid cell stimulated by exogenous ammonium salt; CFCB, coelomic fluid cell stimulated by blood clot; CFCS, coelomic fluid cell stimulated by sulfide; GC, ganglion cell; IC, intestine cell; ICB, intestine cell stimulated by blood clot; MC, muscle cell; NC, nephridial cell; and PC, protractor cell. Lower-case letters **a–c** represent different individual *S. nudus* strain. Arabic numerals 1 and 2 represent swine blood meal injection 0.5 and 11 h later, respectively. **b** Transcription levels of 4 snFPITE-regulating genes from 10 diverse tissue samples and 18 treatment samples. Abbreviations are the same as those used in **a. c** A schematic diagram of the recombinant expression of *snFPITE* genes. These genes were validated by recombinant expression in the *Pichia pastoris* (top) or HEK293 system (middle) or Tn-baculovirus system (bottom). Cell lysates and cell culture media were isolated for fibrinolytic activity analysis, using nontransfected host cell groups as blank control. **d** Recombinant expression and function validation of *snFPITE-n1* genes in Tn-baculovirus system. Western blot analysis of recombinant snFPITE-n1 protein (top). Arrow indicates snFPITE-n1 protein. Fibrin plate analysis of snFPITE-n1 protein (bottom). snFPITE-n1: 2 μg/well. Tn: nontransfected Tn cell culture. Five independent experiments were repeated with similar results (**d**). Source data are provided as a Source Data file.

snFPITE, as plasmin is a strong protease with broad target selectivity[15,16]. Furthermore, the less charged property of the substrate-binding pockets might allow snFPITEs to cleave more substrates than plasmin, such as D-dimer. The broad-range protease activity of snFPITE could act as a double-edged sword while it enhances the speed and potency of thrombolysis. On the one hand, it allows for more efficient and rapid clot breakdown. On the other hand, this increased activity also raises the risk of adverse side effects.

Various PAs and antiplasmins in the body usually regulate fibrinolysis[28]. Generally, the administered tPAs have to overwhelm relatively small amounts (-0.12–1.5 nM) of endogenous PAI1 to generate sufficient active plasmin so as to overcome the endogenous A2AP. Therefore, high doses (50–300 nM or 0.9 mg/kg) and long half-life (more than 1 h) of tPA are usually administered for sufficient thrombolysis[29]. In the present study, the plasminogen-activating ability snFPITE was only slightly inhibited by A2AP, implying that snFPITEs could activate plasminogen with shorter response time and higher efficiency. Hydrolysis and structural modeling experiments unveiled that this inhibitory way might differ from the "suicide" manner of plasmin-A2AP[30,31]. The stepwise cleavage results showed that snFPITE-n1 might first cleave the Arg[376]–Met[377] scissile bond of A2AP. However, Ser[189] of snFPITE-n1, which had a catalytic activity, did not form any covalent bond with Arg[376] from A2AP. RCL cleavage-triggered conformational change or the competition between RCL and the fibrin(ogen) substrate might render snFPITE-n1 inactive. Some simple similar substrates of A2AP, for example, substrates containing only the minimal RC, need to be designed and produced in the future to further clarify this competitive inhibition. Therefore, this competitive inhibition presents snFPITE as a promising fibrin-sensing candidate in treating thrombosis. This is because large amounts of endogenous plasminogen (2 μM) can temporarily block the fibrinolytic activity of free snFPITEs in blood circulation, whereas high fibrin levels in clots enhance the fibrinolytic activity of snFPITEs. The distribution of Cy7-conjugated snFPITE clearly showed that the liver was the major enriched organ, implying that the liver might clear snFPITE as plasmin. However, snFPITE could not form an irreversible snFPITE–PAI1 complex. Therefore, the classic PAI1-dependent pathway could not clear the enzyme[32,33], but some other PAI1-independent clearance pathways might clear it[34–37].

Following the common drug delivery theory, the drug particle size affects the efficiency of reaching and penetrating a target tissue, such as the blood clot[38]. Currently, snFPITEs are the smallest among serine fibrinolytic enzymes exhibiting both plasminogen-activating and fibrin-degrading activities.

An interesting phenomenon was noted when the thrombolytic activity of snFPITE was compared with that of rtPA. In the whole-blood ex vivo model, snFPITE had a stronger fibrinolytic activity than rtPA in the first half-stage, but the performance reversed in the second half-stage. This change was chiefly attributed to the direct fibrinolytic activity of snFPITE. However, the thrombolytic activity of snFPITE was remarkably more potent than that of rtPA in mouse and rat models, probably because of the inhibitory effect of PAI1/A2AP on snFPITE or the longer half-life time of snFPITE. This indicated some additional advantages of intravenously administered snFPITE in vivo. Immunogenicity is another issue that cannot be ignored. Attributed to weak immunogenicity, the most successful thrombolytic drugs have been derived by modifying of the human-sourced tPA[39–44]. The enzyme snFPITE from *S. nudus* is classified as an exogenous protein. Despite its external origin, intravenous administration of snFPITE for a duration of five days resulted in no significant alterations in immune cell populations. This observation suggested that snFPITE was a thrombolytic agent with minimal immune response risk.

Many non-food sources of fibrinolytic enzymes, such as snake venom and earthworms, are associated with toxicity and immunogenicity, limit their therapeutic application. An oral lumbrokinase enteric-coated capsule product is commercially available[42]; however, its clinical application is limited, due to allergies and the restricted scope of patient suitability for oral delivery. This clinical application is partially ascribed to the raw applied material[45]. On the contrary, *S. nudus* used in this study has been a traditionally popular seafood in the coastal areas of Southern China for over several hundreds of years. The biosafety and biocompatibility of components from this organism were well-established[12]. Hence, snFPITE had a great safety advantage over other non-food-derived agents when used as an oral agent. The present study provided encouraging evidence that the safety of snFPITE was comparable to that of the well-established rtPA. Moreover, the wide distribution and abundance of natural resources of *S. nudus*, and the high snFPITE content in the *S. nudus* intestine, are beneficial for further exploration.

## Methods
### Human samples
Citrated plasma samples with high D-dimer levels (above 3 mg/L) were obtained from patients at the Quanzhou First Hospital Affiliated to Fujian Medical University. All studies were performed according to the guidelines of the Animal and Medical Ethics Committee of Huaqiao University (No. M2021017). Informed consents were obtained from patients. All procedures were performed in compliance with all relevant regulations regarding the use of human study participants and were conducted in accordance to the criteria set by the Declaration of Helsinki. Recruitment in this cohort was independent of sex, age, and types of illness, as this study was only focused on the lytic potential of snFPITE on D-dimer. Information on race, ethnicity, or other socio-economical parameters was not collected for this cohort. Given the limited size of the cohort, we did not conduct any analyses stratified by sex. Patients who met the inclusion criteria outlined above were recruited for this study, we therefore did not anticipate any self-selection or other biases.

### Animals
All studies were performed according to the guidelines of the Animal and Medical Ethics Committee of Huaqiao University (No. A202104). The experiments were reviewed and approved by the Ethics Committee of Huaqiao University. All procedures were performed in compliance with relevant ethical guidelines. Mice, rats, and roosters were housed under the standard laboratory conditions with a 12 h light/dark

cycle. The ambient temperature was maintained at $22 \pm 2\,°C$, with a relative humidity of $50 \pm 10\%$. Mice and rats were purchased from Wu Animal Center of Fuzhou University. Roosters and peanut worms (NCBI taxonomy ID 6446) were purchased from local market in Xiamen City, Fujian Province, China. Adult fresh peanut worms were starved for 48 h before the experiment. For clot lysis, tail thrombosis analysis, biodistribution, common carotid arterial thrombosis, safety, and platelet activation experiments, only male mice/rats were selected due to that they are more likely to clot and thrombosis. The mice, rats, and roosters were euthanized by isofluorane overdose after experiments.

## Isolation and purification of snFPITE
Three snFPITE purification methods were established by modifying procedures described in previous studies[46–49].

The first method employed the common multi-chromatography purification system. The intestine of the worms was carefully dissected, homogenized, and centrifuged at $12,000\,g$ for 10 min, at $4\,°C$. The supernatant was precipitated by adding 90% ammonium sulfate (final concentration). The precipitate was dissolved in Tris-HCl (0.02 M, pH 7.4) and desalted through Sephadex G-25 (GE Healthcare, IL, USA, #17-0033) by eluting with Tris-HCl (0.02 M, pH 7.4). A single peak of snFPITE was subjected to DEAE-Sepharose Fast Flow (GE Healthcare, IL, USA, #17-0709) and washed with Tris-HCl (0.02 M, pH 8.0). Also, snFPITE was further purified through a Superdex75 Increase 10/300GL gel-filtration column (GE Healthcare, IL, USA, #17-5174-01) and anion exchange column Resource Q (GE Healthcare, IL, USA, #17-1179) by eluting with Tris-HCl (0.02 M, pH 7.4) and NaCl (0.5 M). The samples from the five peaks were pooled for concentration.

The single lysine affinity chromatography system was used for the second method. The Lysine-Sepharose 4B affinity chromatography column (Solarbio, Beijing, China, #S8911) was first equilibrated with Tris-HCl (0.02 M, pH 7.4). After loading the crude extract of snFPITE, the column was eluted first with Tris-HCl (0.02 M, pH 7.4), and then with NaCl elution buffer (0.15 M). The elution peak was collected. The column was further eluted with the NaCl elution buffer (0.25 M).

The arginine affinity chromatography system was used for the third method. This system was the same as the lysine system, with only Lysine-Sepharose 4B being changed with the Argine-Sephrose 4B column (Solarbio, Beijing, China, #S4560).

## Molecular weight analysis
The approximate molecular weight of snFPITE was first calculated using SDS-PAGE[50]. Purified snFPITE was resolved through 12% SDS-PAGE (Solarbio, Beijing, China, #P1200) and stained with Coomassie brilliant blue R250 (Solarbio, Beijing, China, #C8430). A regression equation ($Y = -0.0055X + 2.0847$, $R^2 = 0.9499$) was built by calculating the retention factor value (Rf) and the relative molecular mass of the bands of the prestained protein ladder (Beyotime Biotechnology, Shanghai, China, #P0068). Then, the approximate molecular weight of snFPITE was estimated using the equation.

The precise molecular weight of snFPITE was calculated through HRMS and UHPLC at Shanghai Applied Protein Technology Co. Ltd. (Shanghai, China).

## Fibrin(ogen) lysis analysis
The fibrinolytic activity was analyzed using the refined fibrin plate method[51]. The fibrin plate was made of Tris-HCl buffer (0.02 M, pH 7.4), 20 mg/mL agarose (Solarbio, Beijing, China, #A8201), 1 mg/mL plasminogen-free fibrinogen (Sigma–Aldrich, MO, United States, 341578), 2 U/mL thrombin (Sigma–Aldrich, MO, United States, #T6884), and 50 μg/mL ampicillin (Solarbio, Beijing, China, #A8180). The samples were dropped into holes (30 μL per hole) made with a clean puncher ($\Phi = 5\,mm$) and incubated at $37\,°C$. The approximate fibrinolytic activity was roughly calculated based on the size of the lysis zone ($\pi \cdot r^2$). Three technical replicates were performed.

Fibrinolytic activity was also more accurately measured using SDS-PAGE of the fibrin hydrolysis products. Plasminogen-free fibrinogen (0.1 μg/μL) was incubated with(out) thrombin (2 U/mL) at $37\,°C$. Then, fibrinolytic agents (concentration was varied with different experiments, which has been indicated in each figure legend) were added and further incubated for 0-4 h. After that, the incubation products were resolved through 12% SDS-PAGE (25 μL per lane). Finally, the degradation results of α, β, and γ chains of fibrin, were used to calculate their fibrinolytic activity.

## Kinetics analysis
The inhibition constant ($Ki$) of PAI1 towards snFPITE was also evaluated according to the method described by Niego's et al.[52] modified with a specific substrate. Briefly, a tissue type plasminogen activator activity assay kit (Colorimetric) (Abcam, MA, United States, #ab108905) was used to analyze the $Ki$ of PAI1 towards snFPITE' plasminogen activating activity. A plasmin activity assay kit (Colorimetric) (Abcam, MA, United States, #ab273301) was used to analyze the $Ki$ of PAI1 towards snFPITE's direct fibrinolytic activity. All these experiments were performed according to the manufacturer's protocols. Briefly, a standard curve of standard agent was established first by measuring the absorbance (405 nm) with a microplate reader. For the $Ki$ experiment, 0, 1.8795, 3.759, and 7.518 μL PAI1 (0.05 μg/μL) were added into the 96-well plate. The total reaction volume was adjusted to 30 μL using PBS. Further 20 μL of snFPITE (12.5 μg/mL) was added to each well. After incubation at $37\,°C$ for 30 min, 20 μL of specific tPA or plasmin substrate (1 μg/μL) was added to each well. The samples were measured immediately using a plate reader at 405 nm. The $Ki$ was calculated by nonlinear regression according to the substrate inhibition model using Origin 8.5. Six technical replicates were performed.

## D-dimer lysis analysis
Blood samples with high D-dimer levels were collected from five recruited patients. The plasma was used for the D-dimer test. The samples were treated with saline or snFPITE for 3 h at $37\,°C$. Then, their D-dimer levels were analyzed on the ACLTOP750[LAS] automatic coagulation analyzer (Werfen, Barcelona, Spain) according to the instrument reagent and instructions. Five biological replicates were used.

## Plasminogen activation
Two approaches were used to determine plasminogen activation. In the first approach, plasminogen-free fibrin plate was prepared as aforementioned. 1 μg snFPITE, or 1 μg plasminogen (Sigma–Aldrich, MO, United States, SRP6518), or 1 μg rtPA (Actilyse, Boehringer Ingelheim, Ingelheim, Germany, #S20110052), or 1 μg snFPITE+1 μg plasminogen, or 1 μg rtPA+1 μg plasminogen was added into the premade hole in fibrin plate. The lysis zone of snFPITE combined with plasminogen was considerably larger than those of snFPITE and plasminogen separately, indicating enhanced plasminogen activation by snFPITE. Therefore, the plasminogen-activating ability was evaluated using the equation: activity = size of the lysis zone (snFPITE combined with plasminogen)− the size of the lysis zone (snFPITE)−the size of the lysis zone (plasminogen). Three technical replicates were performed.

The second approach was modified from our previous method[53], 0.5 μg snFPITE, or 0.5 μg snFPITE+ 10 μg plasminogen, or 0.5 μg rtPA+ 10 μg plasminogen was incubated for 2 h at $37\,°C$. Then, the incubation products were separated through 12% native-PAGE. The gel of each lane was carefully transferred to a plasminogen-free fibrin plate to observe the lysis zone. When snFPITE could activate plasminogen, the co-incubation samples appeared as a new or larger lysis zone. Similarly, the plasminogen-activating ability was also evaluated using the aforementioned equation. Three technical replicates were performed.

## Cleavage sites of snFPITE on plasminogen

Plasminogen (2 µg) was incubated with snFPITE (0.1 µg) for certain periods (0, 10, 30, 60, 90, 120 min), and each incubated solution was separated using 12% SDS-PAGE. All the major bands were extracted separately. The N-, C-, and protein sequencing were performed by Beijing Protein Innovation Co., Ltd. (Beijing, China). The detected sequence was compared with the plasminogen reference sequence (GeneBank: AAA60113.1) to determine the cleavage sites. A schematic of the cleavage sites was drawn from ref. 16. The plasminogen (PDB: 4DUR) structure was generated using PyMOL (https://www.pymol.org), and the cleavage fragments were presented on the plasminogen structure.

## Clot lysis

The halo whole blood clot lysis assay[54] was conducted to evaluate clot lysis. Citrate anticoagulated blood samples were obtained from three healthy male Kunming strain mice (6–8 weeks, 22.0 ± 3.0 g) and then pooled together. Initially, 5 µL droplets of the clotting mixture (Innovin + $CaCl_2$) were deposited on the bottom edge of the wells of a 96 well plate. The clotting mixture was spread around the edge of the wells with the tip of a P100 micropipette containing 25 µL of blood. After incubating the plates at 37 °C for 4 h, the samples were gently introduced into their central clear and empty areas and promptly measured using a plate reader (Tecan Infinete M200, Beijing, China) at 510 nm. Five groups were set as rtPA: 72 ng/mL, snFPITE (H): 72 ng/mL, snFPITE (M): 36 ng/mL, snFPITE (L): 18 ng/mL. Saline was used as negative control. Six technical replicates were performed. The maximum degradation ($D_{max}$) is the maximum degradation value obtained from the experiment. The maximum clot lysis rate ($CLR_{max}$) reached over the experiment corresponds to the maximum value of Dx(t)/dt and was determined on the first derivative analysis of the degradation profile. The activation time ($A_t$) required to induce degradation was defined as the first value of $t$ (in minutes) for which D(t)/dt > 1. The time elapsed until 50% degradation ($T_{0.5}$) was obtained from the intersection of the dotted line and the degradation curve.

## Analysis of enzymatic property

The enzymatic properties of snFPITE were examined from four aspects: temperature, pH value, inhibitor, and stability. The effects of temperature and pH on snFPITE were examined by determining residual activities based on the fibrin plate method following the pre-incubation of snFPITE (1 µg) at different temperatures (20, 30, 40, 50, 60, 70, 80 °C) or with different pH buffers (pH=2, 3, 4, 5, 6, 7, 8, 9, 10, 11, 12) for 30 min. Three biological replicates were used. Then, the effects of protease inhibitors were explored by co-incubating snFPITE (0.1 µg) with 2.5 mM of different protease inhibitors [EDTA (Sigma–Aldrich, MO, USA, #03609), leupeptin (Solarbio, Beijing, China, #L8110), pepstatin (Solarbio, Beijing, China, #P8100), PMSF (Solarbio, Beijing, China, #P0100), 4-(2-aminoethyl) benzenesulfonyl fluoride hydrochloride (AEBSF; Solarbio, Beijing, China, #IA0110), 4-bromobenzamide (Pbpb; Sigma–Aldrich, MO, USA, #190772), benzamidine (Sigma–Aldrich, MO, USA, #12072), iodoacetamide (IAM; Beyotime Biotechnology, Shanghai, China, #ST1411), lysine (Solarbio, Beijing, China, #P8130), arginine (Solarbio, Beijing, China, #A0013), PAI1 (Sigma–Aldrich, MO, USA, #A8111), and A2AP (Sigma–Aldrich, MO, USA, #A8849)] for 30 min. Then, the fibrin lysis or plasminogen-activating activities of snFPITE were analyzed using the aforementioned SDS-PAGE methods. Three biological replicates were used. The stability of snFPITE was examined by storing the sterile samples (1 µg) at 37 °C for 1–90 days. Then, the residual activities were determined by plasminogen-rich fibrin plate to evaluate the properties of snFPITE. Three biological replicates were used.

## Mouse tail thrombosis analysis

After one week of acclimatization, 48 male Kunming strain mice (6–8 weeks, 25.0 ± 3.0 g) were divided randomly into blank control (blank), negative control (saline), positive control (rtPA: 0.9 mg/kg), and treatment (snFPITE-H: 0.9 mg/kg, snFPITE-M: 0.45 mg/kg, snFPITE-H: 0.23 mg/kg) groups. All agents were intravenously (i.v.) injected into the mice 24 h after the carragenin (Solarbio, Beijing, China, #C8830) injection (i.p.). The lengths of healthy and embolized tails were measured 72 h after administration. The ratio of healthy tail to embolized tail length was used to evaluate the antithrombotic effects of different treatments. Eight biological replicates were used.

## Biodistribution of snFPITE in rats

After one week of acclimatization, 21 male SD strain rats (5–7 weeks, 180.0–200.0 g) were divided into 7 groups (0, 10, 30, 60, 120, 240, 480 min). Purified snFPITE was conjugated with Cy7-E SE (UE, Jiangsu, China, #C5046) according to its instructions. The unconjugated Cy7 was removed by 10 rounds of ultrafiltration using 10 kD MWCO centrifugal filter (Millipore, MA, USA) at 4500 r/min for 20 min. Cy7 conjugated snFPITE (5 mg/kg) was i.v injected into the rats. Fluorescence imaging (745/774 nm) of the live rat was collected at defined time point with Aniview Kirin (BLT, Guangzhou, China). Then, their organs were dissected and fluorescence imaging was collected again. Three biological replicates were used.

## FeCl3-induced common carotid arterial thrombosis in rats

Twenty male SD strain rats (5–7 weeks, 180.0–200.0 g) were divided randomly into sham-operated control (sham), negative control (saline), positive control (rtPA: 0.9 mg/kg), and treatment (snFPITE: 0.9 mg/kg) groups one week after acclimatization. All agents were i.v injected 1 h before the surgery. $FeCl_3$-induced common carotid arterial thrombosis established in the rats by ZHBY Biotech Co. Ltd (Jiangxi, China). The blood flow was measured using a laser Doppler perfusion imager (PeriScan PSI; Perimed, Stockholm, Sweden) after 0 and 24 h. The thrombosis ability was calculated by the increasement of net flux change. Five biological replicates were used.

## Molecular interaction

The interaction between snFPITE and fibrinogen was determined using the Q-sense E4 QCM-D method[55]. The Au-coated chip (Göteborg, Sweden, QSX 301 Au Batch: #16M415) was prepared following the manufacturer's protocols (Biolin Scientific, Göteborg, Sweden). The chip was equilibrated with Tris-HCl (0.02 M, pH=7.4) at a flow rate of 80 µL/min until the frequency value ($f$) stabilized. The flow rate was lowered to 50 µL/min, and the fibrinogen (0.2 mg/mL) was loaded until $f$ became stable. The chip was blocked with 5% skimmed milk powder (Solarbio, Beijing, China, #D8340) at a flow rate of 80 µL/min until $f$ became stable. Then, snFPITE (0.028 mg/mL) was loaded at a flow rate of 20 µL/min. The molecule interaction was evaluated on the basis of the change in $f$ values. Three technical replicates were performed.

The direct interactions between snFPITE and plasminogen (or fibrinogen) were also determined using the GatorPrime instrument (Gator Bio, CA, USA) according to the manufacturer's protocols by using streptavidin (SA-XT) biosensors[56]. All steps were performed at 30 °C and 1000 rpm. The biosensors were equilibrated in the buffer for 10 min. Biotin-labeled plasminogen or fibrinogen was immobilized on the biosensors, followed by washing for 2 min. Then the plasminogen or fibrinogen-loaded biosensors were submerged in the solutions of snFPITE, for 2–3 min and transferred to the buffer for 2–5 min to measure the association and dissociation kinetics. The data was analyzed using GatorOne software (v2.10.4) and the results were plotted using Origin (v8.0). Six technical replicates were performed.

## Cytotoxicity assay

Human umbilical vein endothelial cells (ATCC, VA, USA) were seeded (1e + 4 cells/well) into the middle of the total area of a 96-orifice plate (Corning, NY, USA, #7007) and cultured overnight. The cells were further cultured for 0, 24, 48, 72, and 96 h after adding snFPITE

(0, 0.1, 1, 10, 50, 100 μg/mL). The CCK8 test kit (Solarbio, Beijing, China, #CK04) was used to determine the cell survival rate. Six technical replicates were performed.

## Mouse toxicity evaluation

After one week of acclimatization, 112 male Kunming strain mice (6–8 weeks, 25.0 ± 3.0 g) were divided randomly into 7 groups (rtPA: 0.9 mg/kg, snFPITE-H: 0.9 mg/kg, snFPITE-M: 0.45 mg/kg, and snFPITE-L: 0.23 mg/kg). All agents were i.v injected into the mice at the same time every day for 5 consecutive days. The amount of water drunk, food consumed, body weight and death were recorded every day. At day 5, bleeding and coagulation indices were examined using tail bleeding and glass-based coagulation assays first. Then, the vein blood was extracted, and some portion of this blood was used for examining routine blood parameters. Another portion of the collected vein blood was subjected to platelet activation detection and four coagulation tests. Furthermore, major organs including the heart, liver, spleen, lungs, and kidney samples were collected and subjected for HE staining. Sixteen technical replicates were performed.

## Platelet activation assay

After one week of acclimatization, 36 male Kunming strain mice (6–8 weeks, 25.0 ± 3.0 g) were divided randomly into blank control (blank), negative control (saline), positive control (rtPA: 0.9 mg/kg), and treatment (snFPITE-H: 0.9 mg/kg, snFPITE-M: 0.45 mg/kg, snFPITE-H: 0.23 mg/kg) groups. All agents were i.v injected into the mice at the same time every day for 5 consecutive days. Citrate-anticoagulated blood samples were obtained from the treated mice. The platelets were separated immediately using the platelet isolation kit (Solarbio, Beijing, China, #P8570) following the manufacturer's protocols. The samples were coated with the PE-conjugated CD62P (P-selectin) monoclonal antibody (ThermoFisher, MA, USA, #12-0626-82), and then analyzed with BD FACSCalibur flow cytometry (BD Biosciences, CA, USA). Flow cytometry gating strategy were showed in Supplementary Fig. 16. Six biological replicates were used.

## Prothrombin activation assay

The human prothrombin complex (Hualan Biological Engineering, Inc.) was dialyzed with distilled water first. Then, 1 U prothrombin was co-incubated with 0.5 μg snFPITE at 37 °C for 2 h. Saline was set as a negative control. Next, 400 μL/well fibrinogen (1 mg/mL) was added to the 24-well plate. Then, 1 U of snFPITE/saline treated prothrombin or thrombin (Solarbio, Beijing, China, #T8021) was added immediately. Further, a 10-μL pipette tip was used to check the presence of white fibrin fibers. The time of the first formation of fibrin fibers was recorded as coagulation time. Six technical replicates were performed.

## Crystallization and mass spectral analysis

The protein was digested with trypsin (Sigma–Aldrich, MO, USA, #T4549) to identify the sequence, and the peptides were analyzed through mass spectrophotometry at the Shanghai Applied Protein Technology Co. Ltd. (Shanghai, China). The sample sequence was determined by comparing the results obtained with our snFPITE database. Data are available via ProteomeXchange with identifier PXD043675.

PMSF (Solarbio, Beijing, China, #P0100) was added to the protein sample with a final concentration of 1 mM before setting up crystallization drops. After 2 days, crystals appeared under the condition of 0.1 M Tris (pH 8.5) and 2 M $(NH_4)_2SO_4$ (Sigma–Aldrich, MO, USA, #A5132), which was further optimized to 0.1 M Tris (pH 9.0) and 2 M $(NH_4)_2SO_4$ with 0.2 M NaSCN (Sigma–Aldrich, MO, USA, #467871). Before the sample was flash-frozen with liquid nitrogen, single crystals were soaked in a cryo-protectant with 20% glycerol (Sigma–Aldrich, MO, USA, #V900860) for a few seconds. The crystals were stored in Uni-Pucks (Molecular Dimensions, OH, USA) before data collection.

The second batch of crystals was formed under a similar condition. The crystallization condition was 2 M $Li_2SO_4$ (Sigma–Aldrich, MO, USA, #920339), 100 mM Tris (pH 8.5; Sigma–Aldrich, MO, USA, #TRIS-RO), 2% v/v polyethylene glycol 400 (Sigma–Aldrich, MO, USA, #807485), and the cryoprotectant contained 25% glycerol.

The x-ray diffraction data were collected from a single crystal to a 1.99 Å resolution at beamline BL18U1 by the Shanghai Synchrotron Radiation Facility (Shanghai, China). The data were collected at 100 K by using a monochromatic x-ray at 0.97915 Å. The data were indexed and integrated using HKL3000[57] and then scaled and merged using AIMLESS[58]. The snFPITE-n1 structure was determined through molecular replacement using the program MOLREP[59]. For this determination, the crystal structure of the earthworm fibrinolytic enzyme component A from *E. fetida* (PDB: 1M9U) was used as a search model. The model was built using Coot[60] and ARP/wARP[61] and refined using Phenix[62] and REFMAC5 (CCP4)[63]. Statistics for data collection and refinement were summarized. The same method was employed to determine the snFPITE-n2 structure, except that the template used was the snFPITE-n1 structure. In the Ramachandran plot, 97.05% of residues were in the favored region, and 2.95% of residues were in the allowed region, but without outliers, for the snFPITE-n1 structure, further 97.46% of residues were in the favored region, and 2.54% of residues were in the allowed region, but without outliers for the snFPITE-n2 structure.

## Structural modeling

Figures related to all protein structures were generated using PyMOL (www.pymol.org). The crystal structure of bovine trypsin complexed with the analogues of sunflower inhibitor 1 (PDB: 4XOJ) was used as the template, and the peptide inhibitor in this structure was replaced with plasminogen substrate peptides to model the binding of plasminogen substrate to snFPITE-n1. Structural minimization using UCSF Chimera[64] was performed for these models, and no steric clash was noted in the final models. A similar method was applied to build the models of snFPITE-n1 complexed with fibrinogen α, β, and γ chain substrates.

The structures of snPAI and snA2AP (snAP) were built through homology modeling using SWISS-MODEL[65]. The structure of human A2AP was built using AlphaFold[66]. The structures of snFPITE complexed with snPAI and snAP were built using the Rosetta online server[67] for protein–protein docking. The structures of snFPITE-n1 complexed with human PAI1 and human A2AP were built by superimposing the structures of human PAI1 (PDB: 1C5G) and human A2AP with those of previous snFPITE-n1–snPAI and snFPITE-n1–snAP complexes. The structures were minimized using UCSF Chimera.

## Whole-genome sequencing

The genome size of peanut worm was calculated by flow cytometry of the single cell suspension of peanut worm. The single cell suspension of rooster red blood cells (50 mL blood was drawn from the ear vein of 3 roosters) was used as the internal standard. Genomic DNA was extracted from an adult peanut worm starved for at least 24 h in artificial seawater. We then sequenced 551.8 Gb of four paired-end libraries and five mate-pair libraries with insert sizes of 450 bp to 40 kb using an Illumina HiSeq 2000 platform (Illumina Inc., CA, USA). After filtering the sequencing data, 331.5 Gb of clean reads was obtained for subsequent assembling. The genome exhibited higher heterozygosity (~3.1%) compared with the marine genome data reported earlier, which served as a big challenge for genome assembling. Furthermore, 36.17 Gb of long reads was also sequenced using a PacBio RSII sequencing platform (Pacific Biosciences Inc., CA, USA) to improve the assembly quality of this complex genome.

## De novo genome assembly

An initial genome assembly of 2.2 Gb in total was achieved by integrating the clean reads from both Illumina and PacBio sequencing

technologies, which was markedly larger than the genome size of 1.67 or 1.64 Gb, as estimated through flow cytometry[68] or a genome survey with a 31-mer analysis[69], respectively. Subsequently, redundant sequences were removed using the program TrimDup from Rabbit (https://github.com/gigascience/rabbit-genome-assembler). Finally, a high-quality genome assembly with a full length of 1.73 Gb was obtained, and the contig and scaffold N50 values were 36.8 and 651 kb, respectively.

The completeness of our genome assembly was evaluated by applying the BUSCO method[70] to identify the conserved single-copy orthologs of the metazoa lineage, of which 82.7% were single-copy and 6.2% were duplicated BUSCOs. Our genome assembly was also evaluated using the combined unigene set from 10 transcriptomes. More than 87.5% of the total unigenes (59,993) were mapped onto the assembled genome, indicating the high completeness and accuracy of our present genome assembly.

### Repetitive elements annotation
A total of 641.45 Mb was annotated as repetitive elements, which accounted for 37.13% of the assembled genome, by using a combination of de novo and homology-based approaches.

### Gene annotation
The MAKER pipeline[71] was employed to annotate gene models from genomic protein-based homologs, RNA-seq transcriptomes, and full-length transcripts from PacBio long-read transcriptome sequencing (Iso-Seq). A confident gene set of 45,896 genes was predicted, with an average length of 13.85 kb. The average length of the coding sequences was 1022 bp, with an average of 5.8 exons per gene.

After searching several public databases for functional annotation, 61.69% of the total predicted genes were found to be annotated in the KEGG database[72] and 72.12% had motif or domain annotations in the InterProScan database[73].

### Orthologous gene clustering and phylogenetic analysis
The gene families were clustered using TreeFam[74]. A total of 51,276 gene families were identified in the 14 examined species. Among them, the *S. nudus* had 15,607 gene families with an average gene number of 2.07, of which 1941 unique gene families containing 7482 genes were specific to *S. nudus*. A functional enrichment analysis (http://david.abcc.ncifcrf.gov/) of these unique gene families was performed. They belonged to 24 significantly enriched pathways (*Q*-value < 0.01). A phylogenetic relationship between *S. nudus* and 13 other representative species was established by using a genome-wide set of 192 bidirectional-best BLAST hits of orthologous genes.

### Divergence time estimation and gene gain/loss prediction
The divergence times of the peanut worm with 13 representative marine species were estimated using PAML MCMCTree[75]. The reference of nine branches, cited from TimeTree (http://www.timetree.org/), was used for calibrating time. Through comparison with other 13 invertebrate species, 14 significantly expanded (*p* < 0.05) and 65 significantly contracted (*p* < 0.05) families were identified in the peanut worm. A total of 327 genes in the significantly expanded gene families were mapped to the KEGG pathways for further functional enrichment analysis. Then, 76 enriched pathways were determined.

### *snFPITE* gene identification in the assembled genome
Two cloned snFPITE protein sequences (239 and 261 aa, respectively) were used as the reference for identifying homologous sequences in our assembled genome by using Exonerate software v2.4.0[76]. Subsequently, these candidate genes were filtered using the conserved domain (GDSGGP), and the identity between these newly predicted and reference sequences was calculated. Genes with aligned identity >80% and coverage >50% were considered as putative *snFPITE* genes.

### Transcription of *snFPITE* genes
The transcriptome data of 18 treatment samples at different time points and 10 diverse tissue samples were sequenced and mapped onto the peanut worm genome and gene set by using HISAT2 v2.0.4[77] and Bowtie2 v2.3.4[78], respectively. RSEM v1.2.12[79] was used to calculate the transcription values. The detailed gene tree and expression heatmaps were built for comparison.

### snFPITE transcript identification from full-length transcriptomes
Nine cloned *snFPITE* nucleotide sequences were used as the reference for aligning the sequenced full-length transcriptomes employing BLASTn v2.9.0[80], with the threshold of identity >80% and coverage >50%.

### Phylogenetic analysis of snFPITE and other reported fibrinolytic enzymes
The protein sequences of 15 *snFPITE* genes were aligned using Clustal Omega[81], and the alignment was then trimmed using trimAl (-gappyout, v1.4; http://trimal.cgenomics.org/). Finally, FastTree v2.1.8[82] was used to construct a phylogenetic tree.

A total of 223 plasminogen (fibrinolytic enzyme) proteins of K01315 [EC:3.4.21.7] were downloaded from the KEGG database (release 102), with one protein selected as a representative of each species. Moreover, another 14 plasminogen proteins of other species were downloaded from public databases, such as NCBI, based on relevant literature. Then, the downloaded plasminogen proteins with 15 snFPITE and 2 crystal proteins of *S. nudus* were used for constructing the phylogenetic tree of the fibrinolytic enzyme genes. First, MUSCLE[83] (v 3.8.31) was employed for multiple alignment of protein sequences. Subsequently, trimAl[84] (v 1.4.1, -automated1) was applied to remove spurious sequences or poorly aligned regions. Finally, IQTREE[85] (v 2.1.3, -m MFP -B 1000) was selected to construct the phylogenetic tree with the best-fit model and 1000 replicates.

### Recombinant expression of *snFPITE* genes
Three expression systems were used in this study. In the *Pichia pastoris* system, the plasmid was constructed by cloning the target sequence (*snFPITE-c1*) into the expression plasmid *pPIC9k* (Invitrogen, CA, USA, #V17520). *Pichia pastoris* GS115 (Invitrogen, CA, USA, #C18100) was then transfected with the expression plasmid. In a fermenter under optimized conditions, the positive transformants were fermented in a BSM medium supplemented with PTM1 medium and biotin (Solarbio, Beijing, China, #D8150). Non-transformed *Pichia pastoris* GS115 was set as blank control. Cell lysates and culture media were taken to fibrinolytic analysis.

In the HEK293 system (ATCC, VA, USA, #CRL-1537), maintenance and transfection were performed following standard protocols[86–88]. The Kozak enhancer sequence, BSA-SP signal sequence, and His-tag were added to those candidate genes (*snFPITE-2, -3, -4, -7, -8, -9, -13, -15*) and cloned into the expression plasmid *pcDNA3.1* (Invitrogen, CA, USA, #V79520). Recombinant plasmids were replicated in the *Escherichia. coli* DH5α (Invitrogen, CA, USA, #EC0112). HEK293 cells were transfected with the purified plasmids, and the culture media were collected on day 5 after transfection. The stable cell line was established following 2 months of G418 (Solarbio, Beijing, China, #IG0010) selection. Nontransfected HEK293 was set as the blank control. The cell lysates and culture media were used for fibrinolytic analysis. The expression of those snFPITE proteins was analyzed with WB using Anti-6* His (HHHHHH) monoclonal antibody (1A065) (Antibody System, Paris, France, #MGK07901) under 1:1000 dilution.

In the Tn-baculovirus expression system, synthesized *snFPITE-n1* and *microplasmin* genes were subcloned into the pFastBac1 vector (Invitrogen, CA, USA, #10712024) and transfected into competent *E. coli* to obtain the expression vector Bacmid. Then, the cells were used

to produce the p1 baculovirus through Bacmid transfection. Validated p1 viruses were then used to produce p2 baculovirus through p1 transfection in Tn cell (Trichoplusia ni, ATCC, VA, USA, #CRL-10859). The culture media were collected 48 h after transfection and subjected to Strep-based purification and functional validation. Nontransformed Tn was set as the blank control. The cell lysates and culture media were taken to fibrinolytic analysis. The expression of snFPITE-n1 protein was analyzed with WB using Anti-Strep Tag (WSHPOFEK) monoclonal antibody (C23.21) (Antibody System, Paris, France, #RGK26101) under 1:1000 dilution.

## Statistical analysis

The experimental data were presented as the mean ± SD. of three independent experiments unless otherwise noted. $p < 0.05$ indicated statistically significant difference. Significance levels were denoted as follows: $^*p < 0.05$, $^{**}p < 0.01$, $^{***}p < 0.001$, $^{****}p < 0.0001$. Statistical analysis was performed using one-way analysis of variance.

## Reporting summary

Further information on research design is available in the Nature Portfolio Reporting Summary linked to this article.

## Data availability

All data supporting the findings of this study are available within the article and its Supplementary Information files or Source Data files. All genome and transcriptome sequence data are available in the NCBI Sequence Read Archive (SRA) under accession number PRJNA997755. This dataset was also deposited in CNGB Nucleotide Sequence Archive under accession code CNP0003708. Crystal data of snFPITE-1 and snFPITE-2 are freely accessible in PDB (Protein Data Bank) under accession codes 8ZVS and 8ZVX, respectively. The sequence identification data (HRMass spectra) of snFPITE-n1 and snFPITE-n2 are available within the Supplementary Data 1 and 2. These data also have been deposited in ProteomeXchange with identifier PXD043675. Crystal data of of earthworm fibrinolytic Enzyme component a from *Eisenia fetida*, catalytic domain of human two-chain tissue plasminogen activator complex of a bis-benzamidine, and ternary microplasmin-staphylokinase-microplasmin complex: a proteinase-cofactor-substrate complex in action, are freely accessible in PDB under accession codes 1M9U, 1A5H, and 1BUI. Source data are provided with this paper.

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

## Acknowledgements

We thank Meimei Liang, Yuming Yan, Yimin Xiao (Huaqiao University) for animal maintenance; Jia Liu (Huaqiao University) for instruments maintenance; Wenzhao Cheng, Fang Bao (Second Affiliated Hospital of Fujian Medical University) and Zhitao Lou (Huaqiao University) for image and data processing; Xiaofang Cheng (Gator Bio Inc.) for molecular interaction analysis. We thank technical assistance from Shiyuan Jiang, Bin Hu, Chunguang Wen, ERCMM and School of Biomedical Sciences during this project was performed. We thank the staff members at BL18U1 beamline of the National Facility for Protein Science in Shanghai (NFPS), Shanghai Advanced Research Institute, Chinese Academy of Sciences, for providing technical support and assistance in data collection and analysis, and Mr. Wulin Song and William Xu for their kind help. Special thanks to Professor Yunqing Yang for his critical reading and comment on manuscript, to Dr. Jia Luo for her critical reading manuscript and to Professor Binghua Jiao for his full support during performance of this project. This research is supported by the Ocean and Fishery Bureau of Fujian Province (2014FJPT08 to R.A.X.), Science and Technology Bureau of Fujian Province (2021Y0027 to Z.F.L., 2019J01070 to M.Q.T.), Fujian Province (3502ZCQXT2021006 to X.L.C.), Science and Technology Bureau of Xiamen City (3502Z20227197 to M.Q.T.), Science and Technology Bureau of Quanzhou City (2018Z012 to M.Q.T.), Scientific Research Foundation of Tangshan Normal University (2020A10 to G.X.M.), National Key Research and Development Program of China (2022YFC2106100 to R.A.X.), National Natural Science Foundation of China (22377098 and 21807088 to X.C.), Scholarship Program for Science and Technology Activities of Returned Overseas Scholars (2022-004 to X.C.).

## Author contributions

R.A.X. initiated the project. R.A.X. and X.L.C. supervised the project. R.A.X., G.X.M., and M.Q.T. designed the study. M.Q.T., R.A.X., G.X.M., X.C., X.L.C., and Q.S. conceptualized the experiments. M.Q.T., G.X.M., H.J.L., H.Y., and C.Y.X. performed most of the experiments. C.J.H. calculated the kinetics. J.J. Z. purified the snFPITE. X.Y. Z. and S.Y.L. performed fibrin(ogen) lysis and D-dimer lysis. C.B., W.C.J., Z.Y., and J.B.J. performed the genome and transcriptome sequencing. M.L.L., L.C., and Y.Q.S. performed the recombinant expression in *P. stroris* and HEK293 system. H.Y. performed clot lysis. Z.F.L. performed PAGE and WB. S.F.C. and Y.Q.W. performed biodistribution analysis and HE. Y.N.X. carried out the safety analysis. Y.G.Z. and Z.M.L. maintained the animals. X.M.X. conducted the platelet activation assay. K.X.Y. performed the recombinant expression in baculovirus system. C.L. performed data analysis. P.L.X. and L.C.Y. collected the clinical samples. M.Q.T. wrote the manuscript with assistance from all authors. R.A.X., X.L.C., G.X.M., Q.S. and X.C. provided funding and resources. R.A.X., X.L.C., M.Q.T., G.X.M., Q.S. and X.C. revised manuscript.

## Competing interests

All authors declare no competing interests.
