## [Transparent Peer Review file · Nature Communications]

A novel facultative plasminogen-independent thrombolytic enzyme from *Sipunculus nudus*

Corresponding Author: Professor Ruian XU

Version 0:

Reviewer comments:

Reviewer #1

(Remarks to the Author)

This extensive project by Ma et al. describes the identification of a novel, multi-modal thrombolytic enzyme, termed snFPITE, that was isolated from an ancient marine organism known as the 'peanut worm' (*Sipunculus Nudus*). Using state-of-the-art genomic, transcriptomic, phylogenetic and proteomic approaches, including crystallography and extensive bioinformatic tools, the authors have dissected the full genome of the worm, identified, isolated and expressed numerous variants of snFPITE, and modelled its unique structure and enzymatic properties. snFPITE is described as a chimeric plasminogen-activating enzyme that also functions as a direct fibrinolytic agent; it uniquely cleaves plasminogen in multi cleavage sites to form a new version of truncated active plasmin, called Flaa; furthermore, snFPITE displays a rather low substrate specificity, degrades fibrinogen, fibrin and D-dimer, and is non-covalently inhibited by PAI-1 and alpha2 antiplasmin (A2AP). The overall picture emerging is of a fascinating, potent and ancient thrombolytic enzyme that the authors claim could be harnessed for future therapeutic purposes to treat thrombosis.

The omics and discovery parts of this work (which will not be part of this review) are undoubtedly impressive in their extent, capabilities and qualities, and the authors deserve a massive credit for their expertise and efforts. Yet, the enzymatic characterisations, both in vitro and in vivo, of snFPITE are rather crude, making it challenging to fully assess the true clinical potential of this enzyme against 'traditional' thrombolytic agents (e.g. t-PA, u-PA and plasmin). Overall, the 'jump' from the snFPITE discovery to in vivo applications and possible clinical utilisation is rather ambitious and could be precepted as too preliminary at this stage. This is especially the case since important aspects of snFPITE's behaviour in vivo have not been studied (see below). A possible idea is to tone down the in vivo and clinical relevance of this paper, while presenting these important additional experiments in a new manuscript.

Major comments

1. As the authors themselves note in the Results, similar enzymes from other worms harbouring both plasminogen-activating and direct fibrinolytic actions have been described in the literature (e.g., Lumbrokinases from earthworm (*Eisenia fetida*) or annelid worms (*Enchytraeus japonensis*); doi.org/10.1371/journal.pone.0053110; doi.org/10.3389/fmolb.2021.680397). This context is missing and is essential in the Introduction. What makes snFPITE stand out in its novelty relative to these related enzymes?
2. Given the broad substrate specificity and multi-site cleavage activity of snFPITE on plasminogen, fibrin, fibrinogen and D-dimer, a question is raised whether the protein is in fact a thrombolytic enzyme, or a much broader and potent protease. Can the authors provide data on the action of snFPITE towards major other mammalian proteins, such as albumin and collagen, to provide some insight into this question?
3. The fibrin plate assay used and western blotting are clear, but rather crude and semi-quantitative when it comes to characterisation of new enzymes. What is the K_m , K_{cat} and V_{max} of snFPITE towards fibrin, fibrinogen and plasminogen? How do they compare to the efficiency of plasmin, t-PA and u-PA? What is the inhibitor constant (K_i) of PAI-1 and A2AP towards snFPITE? Is A2AP inhibition relevant for snFPITE, given that snFPITE generates plasmin (Flaa) and that the association of A2AP with plasmin is extremely fast (overall rate constant $2 \times 10^7 \text{ mol}^{-1} \cdot \text{s}^{-1}$)? These types of questions are

lacking and will be required to truly assess the in vivo behaviour of snFPITE and its clinical potential.

4. Similarly, there are much more advanced ways to measure and compare clot degradation between thrombolytic enzymes in vitro and ex vivo (relative to basic end-point images of clots in well plates). Assays like S-2251-based plasminogen activation assays (to determine plasminogen activation rates using purified proteins), plasma clot lysis assays (Niego et al. Blood Coagul Fibrinolysis. 2008), whole blood Halo assays (Bonnard et al. Sci. Rep 2017) and/or plasma Halo assays (Palazzolo JS et al. RPTH 2022) could be used to accurately determine the kinetics and potency of snFPITE relative to the main thrombolytic enzymes, to better predict its potential in vivo.

5. What are the biodistribution and pharmacokinetic profiles of snFPITE in the mouse? Plasma clearance rates and toxicity studies to key organs (heart, brain) and major clearance organs (liver, kidney) are not described. What is the half-life in the blood circulation? Does it cross the blood-brain barrier?

6. Given that snFPITE degrades fibrinogen and consumes plasminogen, and is not covalently inhibited by A2AP and PAI-1, major haemostatic safety concerns are immediately raised with regards to bleeding complications and/or Disseminated Intravascular Coagulation (DIC). Are there data to indicate the likely safety profile of this agent? Tail-bleeding assays? Platelet activation assays? Does it activate prothrombin?

7. In Figure 1d (blood clot degradation), why was urokinase used as a reference and not t-PA (fibrin-specific), and/or microplasmin? Perhaps the closest agent to Flaa is microplasmin, which contains only the protease domain of plasmin. The current comparison is incomplete and possibly less relevant to thrombolysis in vivo.

8. The fascinating observation of a normal fibrinolytic activity of snFPITE-p1, which lacks the critical Ser189 in its catalytic triad, could suggest that snFPITE may not actually operate via this residue. Can the authors extend on this point?

Additional comments

1. In the in vivo mouse study, why was snFPITE given intraperitoneally and not intravenously?

2. Immunogenicity of snFPITE could be a major issue when it comes to future clinical use of a foreign protein. The authors claim in the Discussion that its consumption as a food source can minimise this issue. How can this be the case given that snFPITE is designated to be administered intravenously, but the protein is degraded in the digestive system?

3. The Figure Legends are unnecessary long and could be shortened by a better use of annotation on the figures themselves (e.g., plus or minus symbols under each lane in gel presentation, to indicate the combination of materials tested). Furthermore, it is difficult to read the roman numerals under the wells in the fibrin plates (better to add the number next to each well). Together with larger fonts in all figures, the article will become much more reader-friendly.

4. The constant 'jumps' (out of order) between main figures and extended figures is confusing and makes it hard to navigate the paper. Is there a way to simplify this?

5. Fibrin gel zymography will be preferable to assess the major actions of snFPITE, for example in Fig. 2A and 3B.

6. Contrary to what is written in lines 322-324, plasmin does not activate plasminogen, but rather cleaves it to generate Lys-plasminogen from Glu-plasminogen. This cleavage relaxes the plasminogen molecule and renders it 10-20 times more activatable, while also providing it with higher affinities to fibrin and cell-surface receptors (Cesarman-Maus and Hajjar 2005). Please correct.

7. Line 340 – plasmin does not have a strict target selectivity. Instead, the requirements for a plasmin cleavage site are lenient (i.e., a basic residue at P1, an aromatic residue at P2 and a non-specific residue at P3), making it a potent protease with a broad substrate specificity (Schaller and Gerber 2011). This is also why a strict control by A2AP is required.

8. Lines 353-355 – Therapeutic administration of t-PA will raise t-PA concentrations in blood to ~50-300 nM and maintain them over 1 hour. The levels of endogenous PAI-1 (~0.12-1.5 nM) are insufficient to control this amount of exogenous t-PA, in contrast to the text. Please amend.

9. Serpin-PA/plasmin complexes assist in their clearance via LRP-1. What are the implications for clearance if snFPITE is not covalently inhibited by the serpins?

Reviewer #2

(Remarks to the Author)

Thrombotic diseases remain leading causes of death and disability. Fibrinolytic therapy with plasminogen activators has been widely used for thrombosis. Here, Ma et al described a new type of plasminogen independent thrombolytic molecule from a marine worm that could activate plasminogen resulting a new product (so called fibrinolytic active agent, Flaa, which may have advantages over the D-dimer from the plasmin mediated thrombolysis), and directly degrade fibrin. Then they extended searching the organism genome and discovered 10 more functional enzymes. These new discoveries gerent future development for a better thrombosis therapeutics. Biochemically, the novel enzyme cleavage sites are well characterized. The high resolution of crystal structures further mechanistically explained the broad cleavage specificity. All experiments are carefully designed, well performed, and comprehensively included genome sequencing, basic biochemistry, structural biology, mouse model and human clinical samples. The manuscript presentation can be improved to make it outstanding. Here are some minor points might be considered in revising the manuscript.

1. May be a good idea to omit the comma in the title.

2. Try to avoid over use of non-standard abbreviations, especially those only used once or few times, like thrombotic disease (TD) ----only used once? urokinase (UK),---- better to use standard uPA (urokinase-type plasminogen activator)?

3. The abstract, the structural information can be briefly stated, as good amount of structural works are described in the result section (example, high resolution structures revealed possible broad substrate cleavage mechanism). If the space is limited, details of snFPITE candidate sequences can be shortened (like "Further more snFPITE candidates from the worm were discovered through genome sequencing, mass spectroscopy and functional tests").

4. Introduction, in need of brief introduction of why marine worm is a good source for fibrinolytic molecule discovery and a brief summary of fibrinolytic enzymes discovered from other worms. In line 65, "advantages have been achieved" may be

changed to “advances have been made”.

5. Figure legends, “Figure 1| The excellent thrombolysis properties of snFPITE.” May be changed to “Characterization of the snFPITE and thrombolysis potency tests”.

Reviewer #3

(Remarks to the Author)

This manuscript provides an impressive array of experiments characterizing a new series of enzymes that act in assays as fibrinolytics. They have several unusual properties that may eventually prove useful in the further treatment of thromboses. Overall, it is a very interesting manuscript.

There are concerns with some experiments that should be addressed, as well as some minor points.

The activity of snFPITE-p1, lacking active site Ser, is quite striking. Can other snFPITE variants be mutagenized to perform the Ser-Ala mutation, which is known to inactivate serine proteases? Can the gel for this protein in Extended Figure 16a be expanded so that the quality of this protein and the data can be clearly discerned? This is an important point, because the active site serine should be absolutely required for activity, so this either indicates something truly and unexpectedly novel or it reveals an underlying problem in the methodology -- why is proteolytic activity observed when none should be there. Also, maybe the alignment can be provided?

For many of the plate assays shown in this paper, there is a circle drawn around areas that are said to be active. Can these circles be removed, at least from an SI version? It is unclear from the images with the drawn circles whether any of the treatments are actually active. By contrast, extended figures 16b and c clearly show visible zones that are readily interpreted.

“Ancient” may not be appropriate to describe the species *Sipunculus nudus*. By definition, since the species currently exists, it is modern. The group of sipunculids is thought to be relatively ancient. Similarly, in the discussion of the enzymes themselves (lines 333-338), the argument seems to be that since the enzymes are ancient, then they are likely to become multifunctional. This argument is inconsistent with observations. Consider that all of the central metabolic enzymes in the animal are equally ancient (at least), so this argument would imply that all other enzymes are multifunctional (they are probably not).

The work could be placed in better context. There are many thrombolytic enzymes identified from annelids and other organisms, and likely the context could be better placed. Since many fibrinolytic enzymes have been identified in worms and other invertebrates, what is the limitation in their clinical application? Also, is it important that these are food organisms?

Extended data figure 2a: lanes 2-4 are not defined in the legend.

Lines 227 and 1159: It is not totally clear whether PMSF “can be” covalently linked to snFPITE-n1, or whether it was actually covalently linked and directly observed in the crystal structure.

Extended data table 1 – not enough information is given to understand this table.

Lines 318-320: is this known from experiment, or is this a guess based upon presumed properties?

PaA11 and A2AP inhibitory activities: inhibition is said to exist but to be weaker and reversible, but is there any quantitation such as potentially K_i/K_d for either of these molecules? There are other possibilities, for example irreversible inhibition of a minor confounding component, that could be cleared up through quantitation? Based upon the cleavage of A2AP, are those simply substrates that decrease the rate in your assays because of competitive reactivity?

Version 1:

Reviewer comments:

Reviewer #1

(Remarks to the Author)

In their revised manuscript, Ma et al. made a thorough effort to address most of the comments previously raised. The revised manuscript has improved overall, especially in its characterisation of the enzymatic properties of snFPITE, the clarity of figures and the readability of the text. Nevertheless, several additional issues were identified, mainly in the new data added, which should be clarified and carefully addressed before the manuscript can be approved for publication:

Major comments

1. Enzyme characterisation methods: While the authors simplified the figure legends and methodologies (as recommended), these paragraphs do not contain the essential information that is required for reproduction of each experiment. Dose/concentration/volumes of reagents, incubation times, statistical analyses and the number of independent experiments are all critically missing (especially in the newly added methods). The biochemical methods and figure legends (both in the

main manuscript and the supp file) require a thorough revision.

2. Please add a dedicated "Statistical analysis" paragraph in the Methods and perform + display the stats in relevant figures and legends (also see in the comments below).
3. It is not always clear what the authors make of the new enzyme data that was included (e.g., the degradation of albumin and collagen). Please add a clear concluding statement following each set of experiments, and more clarity in the Discussion regarding the strength and limitation of snFPITE as a possible new thrombolytic agent.
4. Fig. 1e and 1f: The lysis zones are very difficult to observe. Can the contrast be improved (similar to Fig. 7d)? The same issue exists with Supp Fig. 1d and 1e.
5. Fig. 1g and 1h, and Supp Fig. 1b and 1c: Why did the authors vary the concentrations of S-2251 and S-2288 to find K_m and V_{max} of tPA and snFPITE towards plasminogen and fibrin? This seems incorrect. Instead, the concentrations of tPA, snFPITE and the chromogenic peptide needs to be fixed, and the substrate (for example, plasminogen) needs to be varied (see the method in <https://pubmed.ncbi.nlm.nih.gov/18469556/>). Can you please clarify. Additionally, add units to the values of K_m and V_{max} , also in Supp. Fig. 15b. Please also check the methodology for Si evaluation in Supp Fig. 9a and 9b.
6. Fig. 1i, Fig. 3d, Supp Fig. 4, Supp Fig. 8a (especially important) – while the use of (+) symbols under each lane has improved the figures, it is hard sometimes to associate the (+) with the material used due to the panel length. Please add lines under each reagent to assist with this.
7. Fig. 2a and 2b: Please add statistical analysis and "n" numbers in the legends.
8. Fig. 2d: The Halo whole blood assay is not only quantitative, but also qualitative (e.g., T0.5, CLRmax and 'Activation time' can be extracted). Can the authors display some quantitative data and statistical analysis?
9. Fig. 3b – this column graph is very dense. Is there a way to separate it to the different parameters tested?
10. Fig. 3c – Can the authors display the two overlapping histograms for each platelet group? Also, it is unclear how the platelet experiment was performed, since there is no method description. Please add. Finally, what is the explanation for the apparent platelet inhibition by snFPITE?
11. Fig. 3d – If snFPITE activates prothrombin, why is the 'time of the first fibrin formed' is extended in its presence?
12. Fig. 7d – why does snFPITE-p1 needs to be mutated, if it has no Serine residue in its catalytic triad anyway (apologies if there is a misunderstanding here, but worth clarifying).
13. Supp Fig. 8a: Lane 3 – plasminogen degradation by snFPITE does not seem to be blocked by PAI-1 at all, and only partially by A2AP (lane 4). The same is true with fibrinogen (last 4 lanes). Can the authors modify their statements to reflect this? In fact, the authors should examine carefully their claims with regards to these inhibitions. As mentioned by the other reviewers, A2AP most likely serves as a substrate for A2AP, rather than an inhibitor (also based on Supp Fig. 9c).
14. The Abstract, Introduction and to a lesser extent other sections require another round of English editing. For example:
 - "the plasminogen" should be changed to "plasminogen" (lines 39, 54, 72, 82, 87, 391)
 - Line 38: "rather than the conventional plasmin" → "than conventional plasmin"
 - Line 46: "study" → "studies"
 - Line 56: "finely" → "a finely"
 - Line 63: "some" → "while some"
 - Line 64: "the conventional plasmin" → "conventional plasmin"; "activation" → "activating"
 - Line 68: "have" → "that have"
 - Lines 73, 83: "decompose" → "convert"
 - Line 76: "a clinical orally thrombolytic drug lumbrokinase" – rephrase
 - Line 79: "a unique coastal tidal flat survival environment – unclear, rephrase
 - Line 80: "We therefore successfully identified" → "We therefore studied"
 - Line 95: Remove "of"
 - Line 133: "current tPAs" → "current PAs"
 - Line 191: "is very difficult" → "requires further examination"; "Next, the co-incubation on snFPITE" → "Next, co-incubation of snFPITE"
 - Line 196: "In a word" → "In summary"
 - Line 295: Remove "how"
 - Line 302: Remove "therefore"
 - Line 306: Remove "the"
 - Line 383: "Amazingly" → "Strikingly"
 - Line 420: "inhibit fibrinolysis by far" → "substantially inhibit fibrinolysis"
 - Line 466: "Whether" → "Is"
 - Line 487: "years" → "of years"
 - Line 492: "nature" → "natural"
 - Line 493: "anti-thrombin" → "anti-thrombotic, or thrombolytic"

Additional comments

1. Line 137: "markedly improved patients" – what patients? Please clarify.
2. Line 138: "a better protective effect than rtPA" – protective effect on what?
3. The argument in paragraph 433- 436 does not make sense. Clinically administered tPA (dose: 0.9 mg/kg over 1 h of infusion for stroke) easily overcomes endogenous PAI-1 and creates a large amount of plasmin. The clinical dose of tPA is definitely sufficient in this context. Please amend.
4. Line 450: There are large plasminogen levels in the circulation (2 μ M). Please amend the argument.
5. Line 464: Do you mean "serine-free" (i.e., a protease without serine)?
6. Line 727: "S-2251" need to be replaced with "S-2288".
7. Supp Fig. 1d and 1e legend: Change "9=day 10" to "9=day 100".
8. Supp Fig. 2 legend: It is unclear what is "Abnormal plasminogen". Also, there is no obvious difference between the

'Dashed circle' for abnormal plasminogen and 'Solid circle' for normal plasminogen; they both appear as dashed circles.

9. Supp. Fig. 3b: Any stats performed?

10. Supp Fig. 4, right panel: There seems to be a small lytic zone in the Flaa region also with plasminogen alone (lane 1). How can the authors explain this?

Reviewer #2

(Remarks to the Author)

The revised version is much improved. All the reviewers' questions are adequately addressed, no more question from me.

Reviewer #3

(Remarks to the Author)

The authors have done much to improve the manuscript. There is a lot to like about it, and it is pretty interesting. Now that some points have been clarified, it is easier to find some of these underlying issues with the characterization of enzymes.

kinetics -- issues include (including what is shown in the main text and SI figures):

where is the experimental methods section for kinetics (anywhere k_i , v_{max} , or k_m is shown)?

no units for v_{max} , k_i , or k_m

no error bars

no indication of (or difficult to find) number of replicates per point

too many sig figs (implied precision greater than possible)

figure 5h: no indication of where it came from in terms of experimental method.

where is the method written up underlying this figure?

no mention of k_d in manuscript

no replicates indicated

I am still nervous about the activity of serine proteases when serine is not present. This should not happen. Could the authors more clearly present the controls, methods, and underlying data supporting this aspect?

Version 2:

Reviewer comments:

Reviewer #1

(Remarks to the Author)

In their 2nd revision, Ma et al. made another solid attempt to address the concerns raised and the manuscript has overall improved. Unfortunately, significant mistakes and inaccuracies are still present, as highlighted below, which the authors need to correct before publication. It will be appreciated if the next revision will include all the correct information to allow the conclusion of this review.

Comments

1. In Fig. 1g and 1h, the high molar values of K_m (8.71 M for S-2251 and 41.39 M for S-2288) clearly do not represent the K_m of snFPITE towards its substrates, plasminogen and fibrin, but instead of plasmin and fibrin towards their amylolytic peptides.

As highlighted to the authors previously, these assays were designed incorrectly (see comment #5 in the previous set of comments) and the data generated was not relevant to the questions it was meant to test. The authors need to correct these assays in the following way:

- Vary plasminogen and fibrin concentrations on a fixed snFPITE and reporter peptide concentrations.
- These reactions are from a second order (i.e., snFPITE activates a substrate, which then activates a reporter peptide)
- Fit the raw parabolic curves (before the reaction "curls" due to reporter peptide depletion) to a polynomial second order equation ($y = AX^2 + Bx + C$).
- Plot the activation rate V (represented by the constant A ; in $\Delta Abs_{405nm} s^{-2}$) against the plasminogen or fibrin concentration to calculate the K_m and V_{max} of snFPITE towards these natural substrates (not towards the reporter peptides).
- Please refer to Niego et al. Blood Coagul Fibrinolysis 2008 for example.

2. The contrast in Fig. 1e and 1f, as well as Supp Fig. 1d and 1e, is still not satisfying. However, if this is the best the authors can do, then it could be accepted.

3. New lines 160-162, please amend the sentence: "Such broader protease activities of snFPITE may not only strengthen its thrombolytic properties, but could also raise potential bleeding issues, which will need to be carefully assessed in future clinical exploration".

4. New line 460: "...of snFPITE could act as a double-edged sword while it enhances the speed and potency..."

5. As requested before, please add at the end of each figure legend and supplementary figure legend a sentence describing the statistical test and level of significance (e.g., Bars represent Mean \pm SD; * $p < 0.05$, ** $p < 0.01$ by one-way ANOVA). The

- legends need to contain all the statistical information irrespective of the Statistical Analysis section.
6. It is recommended to use colour bars for new Fig. 3b, similar to Supp Fig. 3b and 3c.
 7. The colour description in the legend of Fig. 3c has been switched: The **green** curve of flow cytometry represents the nonactivated platelet (PE-). The **blue** curve of flow cytometry represents the activated platelet (PE+). Please correct.
 8. Fig. 3d has not been changed despite the authors' claim. the 'time of the first fibrin formed' is still extended in the presence of snFPITE relative to saline.
 9. Please specify what is the "negative control group" in page 8, line 138. Is this a group of healthy patients without thrombosis?
 10. Supp Fig. 4, middle and right panels: There still seems to be a small lytic zone in the Flaa region also with snFPITE alone (lane 1). How can the authors explain this? According to the data currently presented, Flaa may in fact be an active fragment of snFPITE, which might be generated by plasmin after plasminogen activation by snFPITE. This possibility requires a careful examination.

Reviewer #3

(Remarks to the Author)

I liked this paper on the initial review and will not restate the reasons why here. The authors have done a reasonable job of trying to address reviewer comments. I would urge the authors to consider the following:

- 1) Some of the conclusions are stated maybe more firmly than the data support. Examples:
 - a) Line 133 "...is therefore more advantageous.... in patients." Preliminary experiments demonstrating initial efficacy and pharmacology in animal models probably should not be used to state that it would be advantageous in patients. A more cautious approach is suggested.
 - b) Line 140, the data do not confirm an advantage in thrombosis treatment. They instead show that the proteins exhibit the desired properties in the assays that you are performing, suggesting that it is worth investigating whether they might have an advantage in thrombosis treatment.
 - c) Line 119, snFPITE is said to do "considerably better", but no significance is indicated between the conditions in question.
 - d) And elsewhere, I would suggest greater caution in making concluding statements.
- 2) The methods could use improvement. They are not well linked to the figures and main text; as one example, sometimes, a concentration for an experiment or the number of replicates performed is given in the figure legend, but not in methods text. Not all the conditions are listed in the methods, so that I could not always picture how to do the experiments. Also, I did not see the mouse toxicity methods, but maybe I just missed them. If the authors could take one more look and try to address any issues that they see, it would be helpful.
- 3) The new enzyme mechanism is still preliminary.
 - a) In the papers cited by the authors, the proteolysis absent an active site serine is very minor - 10^5 or 10^6 times slower. Since the method followed release of fluorescent groups, such a large difference makes one wish that these experiments were redone with modern methods and with more proteins. Moreover, such a large difference would be considered to be inactivating in biochemical studies done with less precision. The authors of those papers themselves describe rigorously removing potential sources of protease contamination to be sure. Also, in the papers talking about wild-type proteins with different mechanisms, they were confirmed using solid biochemical experiments.
 - b) Typically, one would want the purified protein and significant characterization in order to conclude that a new mechanism is present. The proxies described here are initial steps, but they do not prove a new mechanism. There is still a reasonable chance that something unexpected is going on.
 - c) For these reasons, I think it's fine for the authors to describe the data, but if I were them I would be more cautious in the interpretation. At this point, I suggest it's at the authors' discretion.

Version 3:

Reviewer comments:

Reviewer #1

(Remarks to the Author)

Thank you for your patience and sincere attempts to address the issues raised. Please see the document attached for final comments and suggestions.

If the kinetics assays are too complex, it could be considered to remove them altogether. However, in their correct form they still require revision.

Reviewer #4

(Remarks to the Author)

This manuscript describes the discovery and characterization of a plasminogen-independent thrombolytic protease(snFPITE) isolated from *Sipunculus nudus*. The authors provide evidence that the enzyme hydrolyzes fibrin and generates a novel, smaller fibrinolytic-active agent from plasminogen. Specific comments concerning the enzymology are

listed below.

Comments:

1. Figure 1 E, F and Suppl Figure 1 B, C. It is not clear why V_{max} is reported here rather than k_{cat} . As seen in Fig. 1 B,C, the enzyme has been purified. If the purified enzyme is used, it would be helpful to calculate k_{cat} , which is much more instructive than V_{max} .
2. The methods section describing the enzyme kinetics results in Fig. 1 B,C are incomplete. It is assumed that the purified enzyme was used, but this is not explicitly stated. Also, what concentration of purified enzyme was used? What buffer and pH was used for the reaction?
3. Supplemental Fig. 9 A,B. It is not clear how the K_i is determined since there does not seem to be any dose response in the presence of inhibitor. The units for K_i are not given. Is it ng/ul or a μM value? It would be better if all values were given in molar units so the reader does not have to calculate it. The model used for fitting should be provided rather than referencing a software package.
4. Lines 316-317 states, "However, the catalytic Ser189 of snFPITE-n1 was found not to form a covalent bond with the Arg376 of A2AP, indicating a different manner of interaction". Since A2AP is a substrate (Fig. 9C), the covalent acyl-enzyme intermediate may not be detected in the time frame of the experiment because it is deacylated to form the product. This does not imply a different manner of interaction but rather simply that A2AP is being turned over. However, this is also a hypothesis and the data presented does not allow mechanistic interpretation.
5. Supplemental Figures 5A and 11. It would be helpful if the catalytic triad Asp, His, Ser could be labeled directly in the sequence alignment. The numbering scheme in Fig S11 does not place a serine at position 189, which makes it difficult to identify the triad residues.

Response to Reviewer #1

Dear Reviewer,

We thank your extremely valuable and critical comments and suggestions. In line with your comments and suggestions, our addresses are as follow:

Changes were highlighted in yellow in corresponding main text, figure legends, and table titles.

Overall comments:

This extensive project by Ma et al. describes the identification of a novel, multi-modal thrombolytic enzyme, termed snFPITE, that was isolated from an ancient marine organism known as the ‘peanut worm’ (*Sipunculus Nudus*). Using state-of-the-art genomic, transcriptomic, phylogenetic and proteomic approaches, including crystallography and extensive bioinformatic tools, the authors have dissected the full genome of the worm, identified, isolated and expressed numerous variants of snFPITE, and modelled its unique structure and enzymatic properties. snFPITE is described as a chimeric plasminogen-activating enzyme that also functions as a direct fibrinolytic agent; it uniquely cleaves plasminogen in multi cleavage sites to form a new version of truncated active plasmin, called Flaa; furthermore, snFPITE displays a rather low substrate specificity, degrades fibrinogen, fibrin and D-dimer, and is non-covalently inhibited by PAI-1 and alpha2 antiplasmin (A2AP). The overall picture emerging is of a fascinating, potent and ancient thrombolytic enzyme that the authors claim could be harnessed for future therapeutic purposes to treat thrombosis.

The omics and discovery parts of this work (which will not be part of this review) are undoubtedly impressive in their extent, capabilities and qualities, and the authors deserve a massive credit for their expertise and efforts. Yet, the enzymatic characterisations, both in vitro and in vivo, of snFPITE are rather crude, making it challenging to fully assess the true clinical potential of this enzyme against ‘traditional’ thrombolytic agents (e.g. t-PA, u-PA and plasmin). Overall, the ‘jump’ from the snFPITE discovery to in vivo applications and possible clinical utilisation is rather ambitious and could be precepted as too preliminary at this stage. This is especially the case since important aspects of snFPITE’s behaviour in vivo have not been studied (see below). A possible idea is to tone down the in vivo and clinical relevance of this paper, while presenting these important additional experiments in a new manuscript.

Response:

Thank you very much for positive assessment on our work. A set of experiments to further characterize this enzyme. Our works include:

1. The S-2251 and S-2288 based plate assay were introduced to precisely quantify the direct fibrinolytic activity and plasminogen activation activity of snFPITE. Please see Fig. 1g, h and Supplementary Fig. 1b, c.
2. Whole blood halo assay was employed to real-time quantitatively study on the thrombolytic activity of snFPITE. Please see Fig. 2c, d.

3. The distribution, clearance and blood life span of snFPITE were evaluated by the fluorescence of the Cy7 conjugated snFPITE in *ex vivo* and *in vivo* model on rats. Please see Fig. 3a, b and Supplementary Table 6.
4. The anti-thrombosis of snFPITE was verified with tail thrombosis in mouse model and FeCl₃ induced common carotid arterial thrombosis in rat model. Please see Fig. 2a, b.
4. The effects of snFPITE on the blood components were examined with *in vitro*, *ex vivo* and *in vivo* models on mice and rats. Please see Supplementary Table 2-4, Supplementary Table 7, and Supplementary Fig. 2b, c.
5. The safety issue was evaluated by changes in organ structure, blood bleeding, blood coagulation, blood composition, mouse behavior, body weight, prothrombin and platelet activation as well as amount of water and food consumption after mice were injected with snFPITE for five consecutive days. Please see Fig. 3b-e, Supplementary Fig. 3a, b, and Supplementary table 1.
6. All *ex vivo* clinical human samples were replaced with mouse and rat models. Please see Fig. 2a, b, 3a-e.

Major comment 1:

As the authors themselves note in the Results, similar enzymes from other worms harbouring both plasminogen-activating and direct fibrinolytic actions have been described in the literature (e.g., Lumbrokinases from earthworm (*Eisenia fetida*) or annelid worms (*Enchytraeus japonensis*); doi.org/10.1371/journal.pone.0053110; doi.org/10.3389/fmolb.2021.680397). This context is missing and is essential in the Introduction. What makes snFPITE stand out in its novelty relative to these related enzymes?

Response:

Thank for your valuable suggestions.

The information of lumbrokinase has been added in the Introduction section. Please see Page 4 Line 63-64.

snFPITE is of novelty in two aspects relative to other related known enzymes.

First, the other known enzymes activate plasminogen and degrade it into the classical fibrinolytic agent plasmin. Plasmin, a large molecule, is rapidly quenched by various antiplasmins in circulation system; it is also unable to digest the D-dimer. In contrast, snFPITE activates plasminogen into a new smaller fibrinolytic agent Flaa. Flaa is considerably smaller than plasmin, it might penetrate clots more efficiently than plasmin, which would make lysis within the clots easier.

In addition, many non-food sources of fibrinolytic enzymes, such as snake venom and earthworms, are associated with toxicity and immunogenicity, which limit their therapeutic use. While the pean worm is a very popular seafood in coastal areas of the Southern China for hundred years as oyster in the Western countries. The biosafety and biocompatibility of components from this traditional seafood have thus been well-established. Please see Page 4, 5, 11, 25 Line 59-67, 75-80, 204-207, 481-488.

Major comment 2:

Given the broad substrate specificity and multi-site cleavage activity of snFPITE on plasminogen, fibrin, fibrinogen and D-dimer, a question is raised whether the protein is in fact a thrombolytic enzyme, or a much broader and potent protease. Can the authors provide data on the action of snFPITE towards major other mammalian proteins, such as albumin and collagen, to provide some insight into this question?

Response:

Thank for your extremely valuable comment and suggestions. We have been trying to further define snFPITE.

Yes, snFPITE should belong to the serine thrombolytic enzyme category. On the basis of our experimental evidences of substrates digestion, structure modeling and moderate trypsin, snFPITE should define as a special thrombolytic enzyme with broader and moderate protease activities. Please see Page 9, 12, 13 Line 148-154, 225-230, 234, 235.

The *in vitro*, *ex vivo* and *in vivo* experiments on the action of snFPITE towards total protein, albumin and collagen were carried out. Please see Page 8, 9 Line 148-154, Supplementary Fig 2b, c, and Supplementary table 3-5.

Major comment 3:

The fibrin plate assay used and western blotting are clear, but rather crude and semi-quantitative when it comes to characterisation of new enzymes. What is the K_m , K_{cat} and V_{max} of snFPITE towards fibrin, fibrinogen and plasminogen? How do they compare to the efficiency of plasmin, t-PA and u-PA? What is the inhibitor constant (K_i) of PAI-1 and A2AP towards snFPITE? Is A2AP inhibition relevant for snFPITE, given that snFPITE generates plasmin (Flaa) and that the association of A2AP with plasmin is extremely fast (overall rate constant $2 \times 10^7 \text{ mol}^{-1} \cdot \text{s}^{-1}$)? These types of questions are lacking and will be required to truly assess the *in vivo* behaviour of snFPITE and its clinical potential.

Response:

Related experiments were established as suggested.

Besides the fibrin plate assay and western blotting experiments with higher quality (Please see Fig 1e, 2c, 3d, and 7d), the S-2251 and S-2288 based plate assay were introduced to precisely quantify the direct fibrinolytic activity and plasminogen activation activity of snFPITE (Please see Fig. 1g, h and Supplementary Fig. 1b, c). Whole blood halo assay was employed to real-time quantitatively study on the thrombolytic activity of snFPITE (Please see Fig. 2c, d).

The K_m and V_{max} of snFPITE towards fibrinogen and plasminogen were determined. Please see Fig. 1g, h.

The K_m and V_{max} of plasmin towards fibrinogen were tested. Please see Supplementary Fig. 1b. While The K_m and V_{max} of rtPA towards plasminogen were tested. Please see Supplementary Fig. 1c.

The inhibitor constant (K_i) of PAI-1 towards snFPITE was measured. Please see Supplementary Fig. 9a, b.

A2AP was found to inhibit both snFPITE degrading fibrin and activating plasminogen through the activity inhibition and K_i experiments. Please see Supplementary Fig. 8a, 9c. The corresponding structure modeling also confirm the inhibition of A2AP on snFPITE. Please see Fig. 5i, j and Supplementary Fig. 8b-d, 9d, e. Thus, it is reasonably conjectured that A2AP inhibiting snFPITE's fibrin degradation is only relevant for snFPITE. On the contrary, A2AP inhibiting snFPITE's plasminogen activation is relevant for both snFPITE and the Flaa, as the similarity found for Flaa and oPlm (the proteinase kringe of plasmin). Please see Supplementary Fig. 15a, b.

Major comment 4:

Similarly, there are much more advanced ways to measure and compare clot degradation between thrombolytic enzymes in vitro and ex vivo (relative to basic end-point images of clots in well plates). Assays like S-2251-based plasminogen activation assays (to determine plasminogen activation rates using purified proteins), plasma clot lysis assays (Niego et al. Blood Coagul Fibrinolysis. 2008), whole blood Halo assays (Bonnard et al. Sci. Rep 2017) and/or plasma Halo assays (Palazzolo JS et al. RPTH 2022) could be used to accurately determine the kinetics and potency of snFPITE relative to the main thrombolytic enzymes, to better predict it's potential in vivo.

Response:

Thanks for valuable suggestion.

Done. Please see Fig. 1g, h, 2d and Supplementary Fig. 1b, c, 9a, b, 15b. While whole blood halo assay was employed to real-time quantitatively study on the thrombolytic activity of snFPITE.

Major comment 5:

What are the biodistribution and pharmacokinetic profiles of snFPITE in the mouse? Plasma clearance rates and toxicity studies to key organs (heart, brain) and major clearance organs (liver, kidney) are not described. What is the half-life in the blood circulation? Does it cross the blood-brain barrier?

Response:

Thank for comment. Several experiments were added.

The biodistribution, clearance and blood life span of snFPITE were evaluated by the fluorescence of the Cy7-conjugated snFPITE in *ex vivo* and *in vivo* model on rats. Please see Page 9, 10 Line 166-172, Fig. 3a, and Supplementary table 6.

Toxicity of snFPITE was examined by effects on 293 cells, blood components and five key organs. Through systematic comparison, we found that the toxicity of snFPITE was similar to that of the clinical drug rtPA. Please see Page 10, 11 Line 175-196, Fig. 3b-e, Supplementary Fig. 3a, b, and Supplementary table 1.

It is clearly that snFPITE could not cross the blood-brain barrier from the fluorescence image of the living rat and dissected organs of the experimental animals. Please see Page 9, 10 Line 168-170, and Fig. 3a.

Major comment 6:

Given that snFPITE degrades fibrinogen and consumes plasminogen, and is not covalently inhibited by A2AP and PAI-1, major haemostatic safety concerns are immediately raised with regards to bleeding complications and/or Disseminated Intravascular Coagulation (DIC). Are there data to indicate the likely safety profile of this agent? Tail-bleeding assays? Platelet activation assays? Does it activate prothrombin?

Response:

To ascertain concerns that raise, the four coagulation tests were established. The results from the four coagulation tests indicated that the index of prothrombin time, activated partial thrombin time, fibrinogen levels, and thrombin time were markedly improved for those animals after the treatment with snFPITE when compared with the negative control group. Please see Page Line 134-138 and Supplementary Table 1.

The bleeding and coagulation indexes were examined with tail-bleeding assay and glass-based coagulation assay after treatment for 5 days. The data of bleeding and coagulation clearly showed that the snFPITE increased both the bleeding and coagulation time at the high concentration, while these affects were alleviated quickly at lower concentration. Please see Page Line 181-183 and Supplementary Fig. 3b.

We did not find platelet activation effect of snFPITE, but instead, inhibition effects were observed by the flow cytometry. Please see Page 8 Line 187-188 and Fig. 3c.

Although the values of prothrombin time, activation part thrombin time and thrombin time increase respectively, but the fibrinogen value decrease. Taken all data above together it is extremely difficult for us to draw a direct conclusion that whether snFPITE activates prothrombin. Therefore, next, we did a co-incubation experiment of snFPITE with prothrombin complex, and found that the snFPITE only activated prothrombin slightly. Please see Page 10, 11 Line 188-193 and Fig. 3d.

In a word, the safety of snFPITE is comparable to that of rtPA.

Major comment 7:

In Figure 1d (blood clot degradation), why was urokinase used as a reference and not t-PA (fibrin-specific), and/or microplasmin? Perhaps the closest agent to Flaa is microplasmin, which contains only the protease domain of plasmin. The current comparison is incomplete and possibly less relevant to thrombolysis in vivo.

Response:

The former routinely positive control agent (urokinase-type plasminogen activator, uPA) has been substituted with recombinant tissue-type plasminogen activator (rtPA), a popular clinical thrombolytic drug (Please see Page 6-11 Line 104-107, 108, 109, 122-127, 138, 148-151, 176-196 and Fig. 1f, 2a-d, 3b-e, Supplementary Fig. 1b, 3b, 4, Supplementary table 1, 3, 4, 7).

Yes, the closest agent to Flaa is microplasmin. Based on the sequence comparison, we found that the sequence of Flaa was only lacking 4 N-terminal amino acids of microplasmin (ocriplasmin, oPlm), a truncated form of plasmin which has

been approved by FDA for the clinical treatment of vitreomacular traction. Furthermore, we constructed oPlm in the Tn/baculovirus expression system, and found that oPlm was similar to snFPITE in some aspects, such as molecular weight, direct fibrinolytic activities, and enzyme kinetic parameters. Please see Page 21 Line 399-404 and Supplementary Fig. 15a, b.

Major comment 8:

The fascinating observation of a normal fibrinolytic activity of snFPITE-p1, which lacks the critical Ser189 in its catalytic triad, could suggest that snFPITE may not actually operate via this residue. Can the authors extend on this point?

Response:

Many thanks for your encouragements.

Yes, indeed. this is a very fascinating observation.

To further confirm this point, 2 *snFPITE* genes (*snFPITE-n1* and *snFPITE-p1*) and their mutations (*snFPITE-n1*^{565T>G} and *snFPITE-p1*^{377A>G}) were subjected to recombinant expression in the Tn-baculovirus system. Amazingly, all the 4 *snFPITE* genes selected still remain fibrinolytic activities. Because both *snFPITE-n1* and *snFPITE-n1*^{565T>G} are pre-terminated at amino acid 125 lacking the key amino acid Ser189, and while the Ser189 in *snFPITE-p1*^{377A>G} has been changed to Ala, thus we speculated that there might routinely occur in *S. nudus* in which fibrinolytic enzyme could still function well even without the canonical catalytic triad structure. As a novel Ser189-independent fibrinolytic enzyme, further exploration is necessary to clarify its molecular pathway. Please see Page 20, 21 Line 374-387, Fig. 7c, d, and Supplementary Fig. 14.

Additional comment 1:

In the in vivo mouse study, why was snFPITE given intraperitoneally and not intravenously?

Response:

The modified mouse and rat models were done according to your kind suggestion. snFPITE and control agents were intravenously administrated. Please see Fig. 2a, b, 3a-c, e, and Supplementary table 1, 4, 6, 7.

Additional comment 2:

Immunogenicity of snFPITE could be a major issue when it comes to future clinical use of a foreign protein. The authors claim in the Discussion that its consumption as a food source can minimise this issue. How can this be the case given that snFPITE is designated to be administered intravenously, but the protein is degraded in the digestive system?

Response:

Thank for your extremely valuable comment.

The immunogenicity of snFPITE has been comprehensively evaluated in the revised manuscript. We found that the immune cells of mice did not change

significantly after intravenous administrated with snFPITE for 5 days, implying that there might be low immunogenicity for snFPITE in mice. Please see Page 10 Line 183-185 and Supplementary table 7.

Additional comment 3:

The Figure Legends are unnecessary long and could be shortened by a better use of annotation on the figures themselves (e.g., plus or minus symbols under each lane in gel presentation, to indicate the combination of materials tested). Furthermore, it is difficult to read the roman numerals under the wells in the fibrin plates (better to add the number next to each well). Together with larger fonts in all figures, the article will become much more reader-friendly.

Response:

Correct.

Please see Fig. 1e, f, 3b, d, 7d, Supplementary Fig. 1d, e, 2b, c, 4, 8a, 9c, 14a, Figure Legends 1-7, and Supplementary Figure Legends 1-15.

Additional comment 4:

The constant 'jumps' (out of order) between main figures and extended figures is confusing and makes it hard to navigate the paper. Is there a way to simplify this?

Response:

Correct.

Please see the revised manuscript.

Additional comment 5:

Fibrin gel zymography will be preferable to assess the major actions of snFPITE, for example in Fig. 2A and 3B.

Response:

Correct.

Please see Fig. 1i, and Supplementary Fig. 2b, c, 4, 6a, 8a, 9c, 14c.

Additional comment 6:

Contrary to what is written in lines 322-324, plasmin does not activate plasminogen, but rather cleaves it to generate Lys-plasminogen from Glu-plasminogen. This cleavage relaxes the plasminogen molecule and renders it 10-20 times more activatable, while also providing it with higher affinities to fibrin and cell-surface receptors (Cesarman-Maus and Hajjar 2005). Please correct.

Response:

Corrected.

Please see Page 21, 22 Line 394-397, 407-408.

Additional comment 7:

Line 340 – plasmin does not have a strict target selectivity. Instead, the requirements for a plasmin cleavage site are lenient (i.e., a basic residue at P1, an aromatic residue

at P2 and a non-specific residue at P3), making it a potent protease with a broad substrate specificity (Schaller and Gerber 2011). This is also why a strict control by A2AP is required.

Response:

Yes, free plasmin does not have a strict target selectivity.

Corrected.

Please see Page 8, 9, 15, 22, 23 Line 146-148, 154-156, 277-279, 427-430.

Additional comment 8:

Lines 353-355 – Therapeutic administration of t-PA will raise t-PA concentrations in blood to ~50-300 nM and maintain them over 1 hour. The levels of endogenous PAI-1 (~0.12-1.5 nM) are insufficient to control this amount of exogenous t-PA, in contrast to the text. Please amend.

Response:

Corrected.

Please see Page 23 Line 433-436.

Additional comment 9:

Serpin-PA/plasmin complexes assist in their clearance via LRP-1. What are the implications for clearance if snFPITE is not covalently inhibited by the serpins?

Response:

Based on the observation that snFPITE mainly enriched in the liver (Please see Page 9 Line 166-167, and Fig. 3a) and did not covalent bind to PAI1 (Please see Page 15, 16 Line 287-290, and Supplementary Fig. 9c), we hypothesized that snFPITE could not be cleared by the classic PAI1-dependended pathway, but by some other PAI1-independent clearance pathways. Please see Page 24 Line 452-457.

Kind regards,
Ruian Xu, PhD

Response to Reviewer #2

Dear Reviewer,

We thank your extremely valuable and critical comments and suggestions. Our addresses are as follow:

Overall comments:

Thrombotic diseases remain leading causes of death and disability. Fibrinolytic therapy with plasminogen activators has been widely used for thrombosis. Here, Ma et al described a new type of plasminogen independent thrombolytic molecule from a marine worm that could activate plasminogen resulting a new product (so called fibrinolytic active agent, Flaa, which may have advantages over the D-dimer from the plasmin mediated thrombolysis), and directly degrade fibrin. Then they extended searching the organism genome and discovered 10 more functional enzymes. These new discoveries gerent future development for a better thrombosis therapeutics. Biochemically, the novel enzyme cleavage sites are well characterized. The high resolution of crystal structures further mechanistically explained the broad cleavage specificity. All experiments are carefully designed, well performed, and comprehensively included genome sequencing, basic biochemistry, structural biology, mouse model and human clinical samples. The manuscript presentation can be improved to make it outstanding. Here are some minor points might be considered in revising the manuscript.

Response:

Thanks for your professional assessment and valuable comments. We have revised thoroughly according to your suggestions. Changes are highlighted in yellow in corresponding main text, figure legends, and table titles.

Comment 1:

May be a good idea to omit the comma in the title.

Response:

Thanks, Done.

Comment 2:

Try to avoid over use of non-standard abbreviations, especially those only used once or few times, like thrombotic disease (TD) ----only used once? urokinase (UK), better to use standard uPA (urokinase-type plasminogen activator)?

Response:

Thank for your suggestion.

All the non-standard abbreviations have been replaced with their full name, except some essential and frequently used terms, such as snFPITE, tPA, Flaa, PAI1, A2AP, HRMS, UHPLC, rtPA, EFEa, RMSD, PMSF, RCL, snPAI, snAP, sntPA, snuPA, oPlm, FDA. For the details, please see Page 3-6, 12, 15, 16, 18, 21 Line 34, 51, 83, 88, 99, 100, 106, 224-225, 285, 299, 344, 345, 400 in the revised manuscript.

Comment 3:

The abstract, the structural information can be briefly stated, as good amount of structural works are described in the result section (example, high resolution structures revealed possible broad substrate cleavage mechanism). If the space is limited, details of snFPITE candidate sequences can be shortened (like “Further more snFPITE candidates from the worm were discovered through genome sequencing, mass spectroscopy and functional tests”).

Response:

Thanks for your suggestion. We have rewritten it accordingly. Please see the new Abstract section.

Comment 4:

Introduction, in need of brief introduction of why marine worm is a good source for fibrinolytic molecule discovery and a brief summary of fibrinolytic enzymes discovered from other worms. In line 65, “advantages have been achieved” may be changed to “advances have been made”.

Response:

Thank for your suggestion.

The reason to choose peanut worm as an ideal source for fibrinolytic molecule discovery was addressed in Introduction section.

Please see Page 3 Line 75-80.

For line 65, correct.

Comment 5:

Figure legends, “Figure 1| The excellent thrombolysis properties of snFPITE.” May be changed to “Characterization of the snFPITE and thrombolysis potency tests”.

Response:

Done. Please see Page 31 Line 597.

Kind regards,
Ruian Xu, PhD

Response to Reviewer #3

Dear Reviewer,

We thank your extremely valuable and critical comments and suggestions. In the light of your comments and suggestions, our addresses are as follow:

Overall comments:

This manuscript provides an impressive array of experiments characterizing a new series of enzymes that act in assays as fibrinolytics. They have several unusual properties that may eventually prove useful in the further treatment of thromboses. Overall, it is a very interesting manuscript.

There are concerns with some experiments that should be addressed, as well as some minor points.

Response:

Many thanks for your kind comments and deliberate suggestions. Several experiments were carried out. The manuscript was rewritten thoroughly according to your comments point-by-point as following. Changes were highlighted in yellow in corresponding main text, figure legends, and table titles.

Comment 1:

The activity of snFPITE-p1, lacking active site Ser, is quite striking. Can other snFPITE variants be mutagenized to perform the Ser-Ala mutation, which is known to inactivate serine proteases? Can the gel for this protein in Extended Figure 16a be expanded so that the quality of this protein and the data can be clearly discerned? This is an important point, because the active site serine should be absolutely required for activity, so this either indicates something truly and unexpectedly novel or it reveals an underlying problem in the methodology -- why is proteolytic activity observed when none should be there. Also, maybe the alignment can be provided?

Response:

Yes, this is a really exciting observation.

To further confirm this outcome, two *snFPITE* genes (*snFPITE-n1* and *snFPITE-p1*) and their mutations (*snFPITE-n1*^{565T>G} and *snFPITE-p1*^{377A>G}) were subjected to construct into the Tn-baculovirus expression system. As a result, all 4 *snFPITE* genes selected were still of fibrinolytic activities (Please see the Fig. 7d). Since *snFPITE-n1* and *snFPITE-n1*^{565T>G} would pre-terminated at amino acid 125 lacking the key amino acid Ser189, while Ser189 in *snFPITE-p1*^{377A>G} was replaced with Ala, we therefore speculated that there might be routine situation for *S. nudus* in which fibrinolytic enzyme could function well even without the canonical catalytic triad structure. Please see Page 20 Line 374-387, Fig. 7c, d, and Supplementary Fig. 14.

Comment 2:

For many of the plate assays shown in this paper, there is a circle drawn around areas that are said to be active. Can these circles be removed, at least from an SI version? It

is unclear from the images with the drawn circles whether any of the treatments are actually active. By contrast, extended figures 16b and c clearly show visible zones that are readily interpreted.

Response:

Thank you for your suggestion. Done.

We have also improved the plate assay methods, making them more visible and discernable. Please see Fig. 1e, f, 2c, 3d, 7d and Supplementary Fig. 1d, e in the revised manuscript.

Furthermore, the S-2251 and S-2288 based plate assay were introduced to quantify the direct fibrinolytic activity and plasminogen activation activity of snFPITE more reliably (Please see Fig. 1g, h and Supplementary Fig. 1b, c), while whole blood halo assay was employed for real-time quantitative study on the thrombolytic activity of snFPITE.

Comment 3:

“Ancient” may not be appropriate to describe the species *Sipunculus nudus*. By definition, since the species currently exists, it is modern. The group of sipunculids is thought to be relatively ancient. Similarly, in the discussion of the enzymes themselves (lines 333-338), the argument seems to be that since the enzymes are ancient, then they are likely to become multifunctional. This argument is inconsistent with observations. Consider that all of the central metabolic enzymes in the animal are equally ancient (at least), so this argument would imply that all other enzymes are multifunctional (they are probably not).

Response:

Thanks for comments, corrected. Please see the Title, Introduction and Discussion sections in the revised manuscript.

Comment 4:

The work could be placed in better context. There are many thrombolytic enzymes identified from annelids and other organisms, and likely the context could be better placed. Since many fibrinolytic enzymes have been identified in worms and other invertebrates, what is the limitation in their clinical application? Also, is it important that these are food organisms?

Response:

Thanks for your valuable comments.

The reason to choose peanut worm as an ideal source for fibrinolytic molecule discovery was addressed in the Introduction section. In comparison to established thrombolytic enzymes, snFPITE does not necessitate plasminogen for thrombolysis but demonstrates enhanced lytic activity in its presence. This was evidenced by the complete dissolution of cross-linked fibrin without leaving any residual undegradable D-dimer. Meanwhile, it further activated the plasminogen into a novel smaller fibrinolytic-active agent rather than the conventional plasmin.

Please see Page 3 Line 34-39.

Many non-food sources of fibrinolytic enzymes, such as snake venom and earthworms, are associated with toxicity and immunogenicity, which limit their therapeutic use. However, there is a commercially available orally lumbrokinase product named Boluoke®. The limitation of its clinical application is because of allergies and that scope of patient application by oral delivery. While the pean worm is a very popular and important seafood in the coastal areas of the Southern China for hundred years like seafood oyster in the Western countries. Furthermore, the data from this present study show that the safety of snFPITE is comparable to that of rtPA, implying that snFPITE should be of potential as an injectable thrombolytic agent. Please see Page 4, 5, 11, 25 Line 59-67, 75-80, 204-207, 481-488.

Comment 5:

Extended data figure 2a: lanes 2-4 are not defined in the legend.

Response:

The Extended data Figure 2 was replaced with a new Supplementary Fig. 2. All the legends also were revised accordingly. Please see Supplementary Fig. 2 in the latest edition.

Comment 6:

Lines 227 and 1159: It is not totally clear whether PMSF “can be” covalently linked to snFPITE-n1, or whether it was actually covalently linked and directly observed in the crystal structure.

Response:

Three new images have shown clearly that PMSF is covalently linked to snFPITE-n1. Please see Supplementary Fig. 8b. snFPITE-n2-A and snFPITE-n2-B were also provided. Please see Supplementary Fig. 8c, d.

Comment 7:

Extended data table 1 – not enough information is given to understand this table.

Response:

Extended data Table 1 was reconstructed. Please see Supplementary Table 5 in the revised manuscript.

Comment 8:

Lines 318-320: is this known from experiment, or is this a guess based upon presumed properties?

Response:

These sentences were modified. Please see Page 21 Line 394-395 in the revised manuscript.

Comment 9:

PAI1 and A2AP inhibitory activities: inhibition is said to exist but to be weaker and reversible, but is there any quantitation such as potentially K_i/K_d for either of these molecules? There are other possibilities, for example irreversible inhibition of a minor

confounding component, that could be cleared up through quantitation? Based upon the cleavage of A2AP, are those simply substrates that decrease the rate in your assays because of competitive reactivity?

Response:

Many thanks for your comments.

The K_i of PAI1 on snFPITE's fibrinolytic activity and plasminogen activation potency are assessed with S-2251 and S-2288 based plate assay. Please see Page 15 Line 287- 288 and Supplementary Fig. 9a, b.

We ascertain that snFPITE should work as a monomer based upon a set of experiments, implying no minor confounding component in the snFPITE molecule. We therefore could exclude the possibility that PAI1 might inhibit snFPITE through an irreversible inhibition of a minor confounding component of snFPITE. Please see Page 6 Line 99-102 and Fig. 1b-d, 5j.

We have tried several approaches to look for some simple substrates that can be used for competition with A2AP. Unfortunately, we have not identified a substrate suitable so far. Nevertheless, after the co-incubation of snFPITE with A2AP, there is a decrease in the inhibition effect of A2AP whereas an increase of fibrin(ogen) or plasminogen, indicating that it is a competition between A2AP and fibrin(ogen) or plasminogen. Please Page 15, 16 Line 282-285, 297-298, 303-305, Fig. 5j and Supplementary Fig. 8a, 9e.

Kind regards,
Ruian Xu, PhD

Dear Reviewers,

Thanks for your positive comments and valuable suggestions. In line with your comments and suggestions, our responses are as follows:

Response to Reviewer#1

Overall comments:

In their revised manuscript, Ma et al. made a thorough effort to address most of the comments previously raised. The revised manuscript has improved overall, especially in its characterisation of the enzymatic properties of snFPITE, the clarity of figures and the readability of the text. Nevertheless, several additional issues were identified, mainly in the new data added, which should be clarified and carefully addressed before the manuscript can be approved for publication.

Response:

Thank you very much for positive assessment on our work. All the comments have been carefully addressed.

Comment 1:

Enzyme characterisation methods: While the authors simplified the figure legends and methodologies (as recommended), these paragraphs do not contain the essential information that is required for reproduction of each experiment. Dose/concentration/volumes of reagents, incubation times, statistical analyses and the number of independent experiments are all critically missing (especially in the newly added methods). The biochemical methods and figure legends (both in the main manuscript and the supp file) require a thorough revision.

Response:

According to your suggestion, the biochemical methods for this enzyme were revised thoroughly and added in Methods section in main text page 54, line 877-896; page 55, line 897-901, 907-916; page 56, line 917-921; page 63, line 1055, 1056, 1063-1065, 1072, 1073; page 64, line 1074-1078.

The figure legends were also revised thoroughly. Please see the main text page 34, line 641-649; page 36, line 657-668; page 38, line 674-683; page 45, line 732-736; page 71, line 1218, 1220; page 72, line 1232, 1233, 1236-1247; page 73, line 1258; page 74, line 1284; page 75, line 1295; page 76, line 1324-1326; page 77, line 1336-1338, 1341.

Comment 2:

Please add a dedicated “Statistical analysis” paragraph in the Methods and perform + display the stats in relevant figures and legends (also see in the comments below).

Response:

A “Statistical analysis” section was added. All the figures and legends were displayed with statistics information. Please see the “Statistical analysis” section, and relevant figures and legends in the revised manuscript page 34, line 647; page 36, line 657, 660, 662, 663, 665, 666, 668; page 38, line 674, 676, 679, 683; page 54, line 884, 885, 894, 895, page 55, line 900, 901; page 64, line 1074-1078; page 71, line 1212-1215, 1222-1225; page 72, line 1237, 1239, 1241; page 75, line 1291-1294; page 77, line 1342, 1343.

Comment 3:

It is not always clear what the authors make of the new enzyme data that was included (e.g., the degradation of albumin and collagen). Please add a clear concluding statement following each set of experiments, and more clarity in the Discussion regarding the strength and limitation of snFPITE as a possible new thrombolytic agent.

Response:

Thanks for your constructive suggestion. A clear concluding statement, strength and limitation of snFPITE were added in the “Results” and “Discussions” sections. Please see main text page 9, line 160-162; page 24, line 459-463.

Comment 4:

Fig. 1e and 1f: The lysis zones are very difficult to observe. Can the contrast be improved (similar to Fig. 7d)? The same issue exists with Supp Fig. 1d and 1e.

Response:

Done. A set of clearer figures were provided. Please see Fig. 1e and 1f, Supp Fig. 1d and 1e.

Comment 5:

Fig. 1g and 1h, and Supp Fig. 1b and 1c: Why did the authors vary the concentrations of S-2251 and S-2288 to find K_m and V_{max} of tPA and snFPITE towards plasminogen and fibrin? This seems incorrect. Instead, the concentrations of tPA, snFPITE and the chromogenic peptide needs to be fixed, and the substrate (for example, plasminogen) needs to be varied (see the method in <https://pubmed.ncbi.nlm.nih.gov/18469556/>). Can you please clarify. Additionally, add units to the values of K_m and V_{max} , also in Supp. Fig. 15b. Please also check the methodology for S_i evaluation in Supp Fig. 9a and 9b.

Response:

Thanks a lot. After carefully checking relative literatures (<https://pubmed.ncbi.nlm.nih.gov/18469556/>), we found that substrate concentrations in all the methodologies were varied. In this study, both S-2251 and S-2288 are not their direct substrates, therefore an indirect substrate, such as plasminogen, was used in this study. Furthermore, snFPITE could directly act on both S-2251 and S-2288, in such circumstance, we have to set up the different concentrations of S-2251 and S-2288 for experiments required.

The units to the values of K_m and V_{max} of relative figures were added. Please

see Fig. 1g and h, Supp. Fig. 1b and c, Supp. Fig. 15b. Experiment methods were provided in the “Methods” section. Please see main text page 54, line 886-896; page 55, line 897-901.

Comment 6:

Fig. 1i, Fig. 3d, Supp Fig. 4, Supp Fig. 8a (especially important) – while the use of (+) symbols under each lane has improved the figures, it is hard sometimes to associate the (+) with the material used due to the panel length. Please add lines under each reagent to assist with this.

Response:

Done. All the figures used (+) symbols were added in the corresponding lines. Please see Fig. 1e, f and i, Fig. 3d, Fig. 7d, Supp. Fig. 2b and c, Supp. Fig. 4, Supp. Fig. 8a, Supp. Fig. 9c.

Comment 7:

Fig. 2a and 2b: Please add statistical analysis and “n” numbers in the legends.

Response:

Corrected. Please see Fig. 2a and b, and its figure legends.

Comment 8:

Fig. 2d: The Halo whole blood assay is not only quantitative, but also qualitative (e.g., T0.5, CLRmax and 'Activation time' can be extracted). Can the authors display some quantitative data and statistical analysis?

Response:

The activation times (At), maximal clot lysis rates (CLRmax), and 50% lysis times (T0.5) were calculated and discussed. Please see Fig. 2e -g, and the main text page 7-8, line 122-128.

Comment 9:

Fig. 3b – this column graph is very dense. Is there a way to separate it to the different parameters tested?

Response:

Corrected. Fig. 3b was separated into 3 independent figures with wider space. Please see Fig. 3b, new added Supp. Fig. 3b and c.

Comment 10:

Fig. 3c – Can the authors display the two overlapping histograms for each platelet group? Also, it is unclear how the platelet experiment was performed, since there is no method description. Please add. Finally, what is the explanation for the apparent platelet inhibition by snFPITE?

Response:

Thanks for your suggestions. Unlike those microscopic pictures within the same frame, the platelet activation experiment of the snFPITE and saline groups were

bio-independent samples. Therefore, flow cytometry plots of two groups could not be merged to get into an overlapping map. As a result, the changes in activated platelet for both groups have been displayed in the Fig. 3c, and clarified in figure legend. Please see Fig. 3c and figure legends in the main text.

The method to describe this experiment was added in the “Methods” section of main text. Please see the main text page 55, line 907-913. A possible explanation was added. Please see the main text page 11, line 192-194.

Comment 11:

Fig. 3d – If snFPITE activates prothrombin, why is the ‘time of the first fibrin formed’ is extended in its presence?

Response:

Corrected. We apologize for this mistake that happened in generating Fig. 3d. i.e., a “+” symbol corresponding to snFPITE was marked to saline. Please see Fig. 3d.

Comment 12:

Fig. 7d – why does snFPITE-p1 needs to be mutated, if it has no Serine residue in its catalytic triad anyway (apologies if there is a misunderstanding here, but worth clarifying).

Response:

We did this additional mutation mainly based the following aspects.

Firstly, since both TAA and TGA are common termination codon, we therefore confirmed this phenomenon by two different approaches.

Secondly, we also found there was an ultra-small fibrinolytic agent (usFA) with a similar molecular weight as snFPITE-p1 in the peanut worm. The abundance of usFA was negative correlated with normal snFPITE. Thus, we speculated that an unknown conversion relationship might exist between usFA and normal snFPITE. We therefore replaced the TAA with a conditional stop codon TGA to see whether snFPITE-p1 would be pre-terminated or readthrough in natural peanut worms, since TGA could code for Selenocysteine once selenocysteine insertion sequence (SECIS) and sufficient selenium are presented in the host.

Comment 13:

Supp Fig. 8a: Lane 3 – plasminogen degradation by snFPITE does not seem to be blocked by PAI-1 at all, and only partially by A2AP (lane 4). The same is true with fibrinogen (last 4 lanes). Can the authors modify their statements to reflect this? In fact, the authors should examine carefully their claims with regards to these inhibitions. As mentioned by the other reviewers, A2AP most likely serves as a substrate for A2AP, rather than an inhibitor (also based on Supp Fig. 9c).

Response:

Yes. A2AP seemed to be a substrate of snFPITE.

We have modified all the statements to reflect its feature more property. Please see the main text page 16, line 293-297, 300, 301, 306-308, and Supp. Fig. 8a and its

legend.

Comment 14:

The Abstract, Introduction and to a lesser extent other sections require another round of English editing.

Response:

This revised manuscript edition was further polished with a professional organization for the second round.

Additional comment 1:

Line 137: “markedly improved patients” – what patients? Please clarify.

Response:

Thanks. Corrected. Please see the main text page 8, line 138.

Additional comment 2:

Line 138: “a better protective effect than rtPA” – protective effect on what?

Response:

Protective effect on thrombosis. This information was added. Please see the main text page 8, line 138.

Additional comment 3:

The argument in paragraph 433- 436 does not make sense. Clinically administered tPA (dose: 0.9 mg/kg over 1 h of infusion for stroke) easily overcomes endogenous PAI-1 and creates a large amount of plasmin. The clinical dose of tPA is definitely sufficient in this context. Please amend.

Response:

Thanks. This paragraph was revised. Please see the main text page 24, line 464-466; page 25, line 467-470.

Additional comment 4:

Line 450: There are large plasminogen levels in the circulation (2 μ M). Please amend the argument.

Response:

Corrected. Please see the main text page 25, line 480, 481.

Additional comment 5:

Line 464: Do you mean “serine-free” (i.e., a protease without serine)?

Response:

It means “a serine protease without the key residue serine in its canonical catalytic triads”.

Additional comment 6:

Line 727: “S-2251” need to be replaced with “S-2288”.

Response:

Yes, corrected.

Additional comment 7:

Supp Fig. 1d and 1e legend: Change “9=day 10” to “9=day 100”.

Response:

“9=day 10” has been corrected to “9=day 90”.

Additional comment 8:

Supp Fig. 2 legend: It is unclear what is “Abnormal plasminogen”. Also, there is no obvious difference between the ‘Dashed circle’ for abnormal plasminogen and ‘Solid circle’ for normal plasminogen; they both appear as dashed circles.

Response:

More information was added in this legend. One “dashed circle” was changed to “solid circle”. Please see Supp. Fig. 2 and its legend.

Additional comment 9:

Supp. Fig. 3b: Any stats performed?

Response:

Yes. Data were presented as the mean \pm s.d. of 6 replicates. $P>0.05$. The statistics information and “n” were added in “Statistical analysis” Section and Supp. Fig. 3b legend, respectively. Please see “Statistical analysis” Section and Supp. Fig. 3b legend.

Additional comment 10:

Supp Fig. 4, right panel: There seems to be a small lytic zone in the Flaa region also with plasminogen alone (lane 1). How can the authors explain this?

Response:

Many thanks. We apologize for this mistake occurred in generating Fig. 4. i.e. Now this mistake has been corrected. Please see Fig. 4.

Response to Reviewer #2

Overall comments:

The revised version is much improved. All the reviewers' questions are adequately addressed, no more question from me.

Response:

Thanks for your professional assessment and valuable comments.

Response to Reviewer #3

Overall comments:

The authors have done much to improve the manuscript. There is a lot to like about it, and it is pretty interesting. Now that some points have been clarified, it is easier to find some of these underlying issues with the characterization of enzymes.

Response:

Thank your kind comments. According to your valuable suggestions, we have revised our manuscript and provide the following response. We appreciate for your insightful feedback, which has significantly contributed to enhance the quality of our work.

Comment 1:

kinetics -- issues include (including what is shown in the main text and SI figures): where is the experimental methods section for kinetics (anywhere k_i , v_{max} , or k_m is shown)?

Response:

The procedures for K_i , V_{max} , K_m determination were added in the Methods section. Please see the main text page 54-55, line 886-901.

Comment 2:

no units for v_{max} , k_i , or k_m

Response:

Corrected. Please see Fig. 1g and h, Supp. Fig. 1b and c, Supp. Fig. 9a and b, Suppl. Fig. 15b.

Comment 3:

no error bars

Response:

The "Statistical analysis" section was added. Please see the main text page 64, line 1074-1078; Fig. 1g and h, Fig. 2a, b, d, e-g, Fig. 3b, Supp. Fig. 1b, c, f, g, Supp. Fig. 3d, Supp. Fig. 9a and b, Supp. 15b, and their corresponding figure legends.

Comment 4:

no indication of (or difficult to find) number of replicates per point

Response:

The statistical analysis section was added. Please see the main text page 34, line 647; page 36, line 657, 660, 662, 663, 665, 666, 668; page 38, line 674, 676, 679, 683; page 54, line 884, 885, 894, 895, page 55, line 900, 901; page 64, line 1074-1078; page 71, line 1212-1215, 1222-1225; page 72, line 1237, 1239, 1241; page 75, line 1291-1294; page 77, line 1342, 1343; Fig. 1g and h, Fig. 2a, b, d, e-g, Fig. 3b, Supp. Fig. 1b, c, f, g, Supp. Fig. 3d, Supp. Fig. 9a and b, Supp. 15b, and their corresponding

figure legends.

Comment 5:

too many sig figs (implied precision greater than possible)

Response:

Thanks for your comment. Two decimal places were kept in the revised version in the light with experimental data. Please see Fig. 1g and h, Supp. Fig. 1b and c, Supp. Fig. 9 a and b, Suppl. Fig. 15b.

Comment 6:

figure 5h: no indication of where it came from in terms of experimental method.

Response:

Thank you for your advice. The method of Fig. 5h was described in the “Methods” section. Please see the main text page 54, line 877-885.

Comment 7:

where is the method written up underlying this figure?

Response:

The methods of K_i , V_{max} , K_m and K_d were added in the “Methods” section. Please see the main text page 54-55, line 886-901.

Comment 8:

no mention of k_d in manuscript

Response:

The K_d values were mentioned in “Results” section. While the procedure for K_d experiment was described in “Methods” section. Please see the main text page 11, line 207; page 15, line 283.

Comment 9:

no replicates indicated

Response:

All the number of replicates were added. Please see the main text page 34, line 647; page 36, line 657, 660, 662, 663, 665, 666, 668; page 38, line 674, 676, 679, 683; page 54, line 884, 885, 894, 895, page 55, line 900, 901; page 64, line 1074-1078; page 71, line 1212-1215, 1222-1225; page 72, line 1237, 1239, 1241; page 75, line 1291-1294; page 77, line 1342, 1343; Fig. 1g and h, Fig. 2a, b, d, e-g, Fig. 3b, Supp. Fig. 1b, c, f, g, Supp. Fig. 3d, Supp. Fig. 9a and b, Supp. 15b, and their corresponding figure legends.

Comment 10:

I am still nervous about the activity of serine proteases when serine is not present. This should not happen. Could the authors more clearly present the controls, methods, and underlying data supporting this aspect?

Response:

Yes, indeed, we were all surprised when we first time observed such a phenomenon in our experiment, i.e., two special snFPITE (snFPITE-p1 and snFPITE-n1^{565T>A}) could exhibit fibrinolytic activity even lack the key residue Ser in the canonical catalytic triads (Fig 7d). We therefore validated this finding step by step as follows.

1). To ascertain the experimental result is reliable and reproducible and to exclude any personal performance mistaken, two individuals repeatedly run the same experiment parallelly at same time. The details of methods, controls and data were described in “Methods” section. Please see the main text page 20-22, line 385-410; page 63, line 1055, 1056, 1063-1065, 1072, 1073; revised Fig. 7d, new added Supp. Fig. 14e.

2). To ensure the test method is foolproof, snFPITE-p1 was recombinantly expressed with two different expression systems, i.e., the HEK293 and the Tn-baculovirus systems described in main text. Similarly, snFPITE-n1^{565T>A} was recombinantly expressed using the Tn-baculovirus systems reported in main text. Please see Page 20-22, Line 386-410. This striking discovery motivated us to examine the stable expression of snFPITE-p1 (without canonical catalytic triad) and two others normal snFPITE genes (snFPITE-8 and snFPITE-15 with canonical catalytic triad) in the HEK293 system (Fig. 7c-middle). Western blot analysis revealed that stable expression of all three snFPITE genes. Please see Page 21, Line 389-393, Supplementary Fig. 14c-bottom.

3). We further examined whether snFPITE-p1 exhibited fibrinolytic activity as traditional proteases (snFPITE-8 & snFPITE-15). Equal amounts of FPITE-8, snFPITE-15 and snFPITE-p1 samples were incubated with the same amount of rat blood clot under identical experimental conditions for 8h. The results showed that all clots in each sample tube were entirely cleared, except for those in the negative control group (non-transfected HEK293), Please see Page 21, Line 394-398, Supplementary Fig. 14c-top.

4). Further the SDS-PAGE test revealed that snFPITE-8 & snFPITE-15 and snFPITE-p1 degraded the A α , B β and γ chains of fibrin efficiently expect for the negative control. Please see Page 21, Line 398-400, Supplementary Fig. 14d.

5). A double check experiment on snFPITE-p1 and two others normal snFPITE genes [snFPITE-n1 and microplasmin (ocriplasmin, oPlm) with canonical catalytic triad] expressed in the Tn-baculovirus system (Fig. 7c-bottom) also reconfirmed that the data were reproducible. Please see Page 21 Line 400-403, Supplementary Fig. 14e.

6). To verify the repeatability of the outcome, two individual experiment was carried out 3 times. As a consequence, the outcomes of all performances are reproducible.

7). Moreover, we carried out another batch of snFPITE-p1 expression in the Tn/baculovirus system soon after receiving reviewer’s latest comments. The results also reconfirmed that it was only the snFPITE-p1 group but not the blank control Tn group exhibited fibrinolytic activity (Figure 1).

Figure 1 Fibrinolytic analysis of re-expressed snFPITE with plasminogen-rich (left) and plasminogen-free (right) fibrin plate.

Notes: Well 1, 2, 9, 10: Cell lysate of the snFPITE transfected Tn cell; Well 3, 4, 11, 12: Cell culture media of the snFPITE transfected Tn cell; Well 5, 6, 13, 14: Cell lysate of the non-transfected Tn cell; Well 7, 8, 15, 16: Cell culture media of the non-transfected Tn cell; Well 17: Saline. The plates were photographed 6 h after adding samples. It was clearly that only the snFPITE transfected Tn cell but not the control Tn cells exhibited fibrinolytic activities. Meanwhile, a fibrin-lysing and plasminogen-activating activities were observed, as a bigger lysing zone was presented in the plasminogen-rich plate.

8). Latterly, to exclude any contamination from cell lines, we further assessed the fibrinolytic activity of five other cell lines (Jeko-1, Mino, Vero e6, Huvec, Jurkat). All cell lines did not show any fibrinolytic activity (**Figure 2**).

Figure 2 Fibrinolytic analysis of the non-transfected cell lines with plasminogen-rich fibrin plate.

Notes: Well 1, 2, 3, 4, 5: Cell lysate of the non-transfected Jeko-1, Mino, Vero e6, Huvec, Jurkat cell lines. It was clear that there was no expression of fibrinolytic agents in the common cell lines.

On the other hand, after carefully checking the literatures, there are some relevant literatures [1-2] to support such a phenomenon.

This discovery is supported by previous research on trypsin. Even though mutating Ser195 in trypsin significantly reduces its activity, water can substitute for the missing functional group [1]. It is essential to recognize that the catalytic triad is not the sole source of a protease's catalytic power as trypsin can still exhibit some activity even

when the catalytic triad is disabled [2]. This finding is also further supported by a latest report for a well-established adenylate kinase that has also been shortened to functional polypeptides (P-loop prototypes) without the canonical catalytic core [3]. Such an enzyme that achieves this without the canonical catalytic core would suggest alternative mechanisms of catalysis, potentially offering new pathways for enzyme design and biotechnological applications. Please see discussion section, Page 26, Lane 493-503.

References

1. Corey DR, Craik CS. An investigation into the minimum requirements for peptide hydrolysis by mutation of the catalytic triad of trypsin. *Journal of the American Chemical Society* 1992; 114, 1784-1790 (1992).
2. Carter P, Wells JA. Dissecting the catalytic triad of a serine protease. *Nature* 1988; 332: 564-568.
3. Vyas P, Malitsky S, Itkin M, Tawfik DS. On the Origins of Enzymes: Phosphate-Binding Polypeptides Mediate Phosphoryl Transfer to Synthesize Adenosine Triphosphate. *J Am Chem Soc* 2023; 145: 8344-8354.

We sincerely thanks for your significantly contribution to enhancing the quality of our work.

Kind regards,

Ruian (Ray) Xu, PhD

Response to Reviewer#1

Dear Reviewer,

We really appreciate your extremely valuable evaluation, positive comments and constructive suggestions. Herein, we disclose our corresponds to your comments as follows:

Overall comments:

In their 2nd revision, Ma et al. made another solid attempt to address the concerns raised and the manuscript has overall improved. Unfortunately, significant mistakes and inaccuracies are still present, as highlighted below, which the authors need to correct before publication. It will be appreciated if the next revision will include all the correct information to allow the conclusion of this review.

Response:

Thank you very much for the positive assessment on our work. All the comments have been carefully addressed. Changes were highlighted by yellow in the revised manuscript.

Comment 1:

In Fig. 1g and 1h, the high molar values of K_m (8.71 M for S-2251 and 41.39 M for S-2288) clearly do not represent the K_m of snFPITE towards its substrates, plasminogen and fibrin, but instead of plasmin and fibrin towards their amyolytic peptides.

As highlighted to the authors previously, these assays were designed incorrectly (see comment #5 in the previous set of comments) and the data generated was not relevant to the questions it was meant to test. The authors need to correct these assays in the following way:

- Vary plasminogen and fibrin concentrations on a fixed snFPITE and reporter peptide concentrations.
- These reactions are from a second order (i.e., snFPITE activates a substrate, which then activates a reporter peptide)
- Fit the raw parabolic curves (before the reaction “curls” due to reporter peptide depletion) to a polynomial second order equation ($y=AX^2+Bx+C$).

- Plot the activation rate V (represented by the constant A ; in $\Delta\text{Abs}_{405\text{nm}} \text{ s}^{-2}$) against the plasminogen or fibrin concentration to calculate the K_m and V_{max} of snFPITE towards these natural substrates (not towards the reporter peptides).
- Please refer to Niego et al. Blood Coagul Fibrinolysis 2008 for example.

Response:

Thanks. We redesigned this experiment according to your instructions. The K_m and V_{max} of snFPITE towards fibrin were recalculated with plasmin activity assay kit (Colorimetric) (Abcam, MA, United States, ab273301). The K_m and V_{max} of snFPITE towards tPA were recalculated with tissue type plasminogen activator activity assay kit (Colorimetric) (Abcam, MA, United States, ab108905). The experiments were performed according to the manufacturer's protocols. The new data indicated that the K_m and V_{max} of snFPITE towards fibrin were 0.1482 μM and 0.172e-7 $\mu\text{M}\cdot\text{S}^{-2}$, respectively. The K_m and V_{max} of snFPITE towards plasminogen were 0.5106 μM and 0.276e-7 $\mu\text{M}\cdot\text{S}^{-2}$, respectively. Please see Results section (page 6, lines 97-101), Figure Legends section (Fig. 1g and 1h in page 32, lines 600-602; page 33, lines 606-607), Methods section (page 54, lines 888-897; page 55, lines 898-909), Methods References section (page 66, lines 1126-1128), and Extended Data Figures and Tables Legends section (page 73, lines 1244-1261; page 80, lines 1379-1383).

Comment 2:

The contrast in Fig. 1e and 1f, as well as Supp Fig. 1d and 1e, is still not satisfying. However, if this is the best the authors can do, then it could be accepted.

Response:

We have repeated the same experiment by different methods without staining and hope their contrast would meet the requirement for publication.

Please see in more details in the revised Fig. 1e and 1f and the changes in Figure Legends section (page 32, lines 597-599; page 33, lines 606-607); Supp Fig. 1d and 1e in Supplementary Figures.

Comment 3:

New lines 160-162, please amend the sentence: "Such broader protease activities of snFPITE may not only strengthen its thrombolytic properties, but could also raise

potential bleeding issues, which will need to be carefully assessed in future clinical exploration”.

Response:

Thanks for your wonderful suggestions. The changes is reflected in the revised main text (page 9, lines 158-160).

Comment 4:

New line 460: “...of snFPITE could act as a double-edged sword while it enhances the speed and potency...”

Response:

Done.

Please see the main text in page 23, line 438.

Comment 5:

As requested before, please add at the end of each figure legend and supplementary figure legend a sentence describing the statistical test and level of significance (e.g., Bars represent Mean \pm SD; * $p < 0.05$, ** $p < 0.01$ by one-way ANOVA). The legends need to contain all the statistical information irrespective of the Statistical Analysis section.

Response:

Thanks a lot. The statistical information was added in the figure legend and supplementary figure legend. Please see the revised Figure Legends section (page 32, lines 601-602; page 34, lines 612-626; page 35, lines 627-628; page 37, lines 636-642), and Extended Data Figures and Tables Legends section (page73, lines 1244-1252, 1258-1261; page 74, lines 1272-1280; page 77, lines 1330-1333; page 80, lines 1380-1383).

Comment 6:

It is recommended to use colour bars for new Fig. 3b, similar to Supp Fig. 3b and 3c.

Response:

Thanks for your suggestion.

Color bars were used in revised Fig. 3b. Please see the corresponding changes in page

36, line 634 and page 37, lines 645-646.

Comment 7:

The colour description in the legend of Fig. 3c has been switched: The **green** curve of flow cytometry represents the nonactivated platelet (PE-). The **blue** curve of flow cytometry represents the activated platelet (PE+). Please correct.

Response:

Corrected.

Please see the revised Fig. 3c and its figure legends in the main text page 36, lines 637-639.

Comment 8:

Fig. 3d has not been changed despite the authors' claim. the 'time of the first fibrin formed' is still extended in the presence of snFPITE relative to saline.

Response:

Corrected.

We apologize for forgetting to correct this mistake in the 2nd revision. It occurred during the generation of Fig. 3d. i.e., a "+" symbol corresponding to snFPITE was marked to saline. Please see the revised Fig 3d in main text page 36, line 639 and page 37, lines 645-646.

Comment 9:

Please specify what is the "negative control group" in page 8, line 138. Is this a group of healthy patients without thrombosis?

Response:

The "negative control group" is the group of healthy mice only treated with saline, which was specified. Please see the changes in the main text (page 8, lines 133-135).

Comment 10:

Supp Fig. 4, middle and right panels: There still seems to be a small lytic zone in the Flaa region also with snFPITE alone (lane 1). How can the authors explain this? According to the data currently presented, Flaa may in fact be an active fragment of snFPITE, which might be generated by plasmin after plasminogen activation by

snFPITE. This possibility requires a careful examination.

Response:

A smaller lytic zone was present in snFPITE alone. snFPITE is a fibrin-degrading agent (24.925 kD), which could lyse the fibrinogen-polyacrylamide gel. When snFPITE was incubated with plasminogen (90-93 kD), plasminogen would be cleaved into Flaa (26.83 kD) which is also a fibrin-degrading agent. As the dose of plasminogen (10 μ g) was much larger than snFPITE (0.5 μ g), a larger lytic zone would be presented.

The statement was modified in Results section. Please see the main text in page 12, lines 211-212.

We sincerely appreciate for your significantly contribution to enhancing the quality of our work.

Response to Reviewer #2

Dear Reviewer,

Thank you very much for your extremely valuable comments and suggestions. In light of your opinions, we revised the manuscript thoroughly.

Overall comments:

I liked this paper on the initial review and will not restate the reasons why here. The authors have done a reasonable job of trying to address reviewer comments. I would urge the authors to consider the following:

Response:

We thank you for your kind comments. According to your valuable suggestions, we revised our manuscript and disclose the following response to address each point raised. We appreciate your insightful feedback, which has significantly contributed to enhance the quality of our work. Changes were highlighted by yellow.

Comment 1:

1) Some of the conclusions are stated maybe more firmly than the data support. Examples:

a) Line 133 "...is therefore more advantageous.... in patients." Preliminary experiments demonstrating initial efficacy and pharmacology in animal models probably should not be used to state that it would be advantageous in patients. A more cautious approach is suggested.

Response:

Thank you for your nice suggestion. This sentence was modified. Please see the main text in page 8, lines 137-139.

Comment 2:

Line 140, the data do not confirm an advantage in thrombosis treatment. They instead show that the proteins exhibit the desired properties in the assays that you are performing, suggesting that it is worth investigating whether they might have an advantage in thrombosis treatment.

Response:

Corrected. Please see the revised main text in page 8, lines 137- 139.

Comment 3:

Line 119, snFPITE is said to do "considerably better", but no significance is indicated between the conditions in question.

Response:

Yes, it is done. There was no significance between tPA and snFPITE, so the sentence "but snFPITE performed better" was removed. Please see the change in page 7, line 116.

Comment 4:

And elsewhere, I would suggest greater caution in making concluding statements.

Response:

Thanks for your valuable suggestions. All the statements were checked and revised. Please see the main text in page 3, line 42; page 5, lines 65, 72-73; page 6, lines 92-93; page 7, line 104; page 8, lines 126, 129-130, 137; page 10, lines 164, 172-173; page 11, line 192; page 12, lines 205-208, 216, 219; page 13, lines 225, 229, 240-243; page 14, lines 244, 250, 254-255, 262; page 15, lines 274, 277, 278, 283; page 16, line 289; page 18, line 338; page 19, line 348; page 20, lines 372, 374, 380; page 23, line 438.

Comment 5:

The methods could use improvement. They are not well linked to the figures and main text; as one example, sometimes, a concentration for an experiment or the number of replicates performed is given in the figure legend, but not in methods text. Not all the conditions are listed in the methods, so that I could not always picture how to do the experiments. Also, I did not see the mouse toxicity methods, but maybe I just missed them. If the authors could take one more look and try to address any issues that they see, it would be helpful.

Response:

Thank for your very nice suggestions.

We have double checked the methods thoroughly. The links between figures and main text, concentration, replicates were added. The methods for mouse toxicity and biodistribution of snFPITE in rats were added.

Please see the revised main in page 45, lines 711-718; page 46, lines 722, 723, 731; page 47, lines 741-758; page 48, lines 759-765, 772-773, 776-778; page 49, lines 779, 784-796; page 50, lines 798-800, 809-816; page 51, lines 821-822, 831-836; page 52, lines 840-857; page 53, lines 862-863, 866-867, 872-873, 876-877; page 54, lines 886-897; page 55, lines 898-916; page 56, lines 917-935; page 57, lines 940, 948; page 73, lines 1244-1261; page 74, lines 1272-1280; page 77, lines 1330-1333; page 79, line 1376; page 80, lines 1380-1383.

Comment 6:

The new enzyme mechanism is still preliminary.

a) In the papers cited by the authors, the proteolysis absent an active site serine is very minor - 10^5 or 10^6 times slower. Since the method followed release of fluorescent groups, such a large difference makes one wish that these experiments were redone with modern methods and with more proteins. Moreover, such a large difference would be considered to be inactivating in biochemical studies done with less precision. The authors of those papers themselves describe rigorously removing potential sources of protease contamination to be sure. Also, in the papers talking about wild-type proteins with different mechanisms, they were confirmed using solid biochemical experiments.

b) Typically, one would want the purified protein and significant characterization in order to conclude that a new mechanism is present. The proxies described here are

initial steps, but they do not prove a new mechanism. There is still a reasonable chance that something unexpected is going on.

c) For these reasons, I think it's fine for the authors to describe the data, but if I were them I would be more cautious in the interpretation. At this point, I suggest it's at the authors' discretion.

Response:

Thank you for your wonderful advice and suggestions. In this revised manuscript, we therefore focus on routine functional proteases only while excluding those without a serine residue. This also was the suggestion from senior editor. Please see the Abstract, Introduction, Results, Discussion, Figure and Legends section in the revised main.

a) “Notably, two sequences exhibited fibrinolytic activities despite the lack of the key Ser residue in the canonical catalytic triads, suggesting a brand new unknown fibrinolytic mechanism.” was removed in page 3, line 42.

b) “Further site mutation experiments displayed two snFPITE genes that exhibited fibrinolytic activities even when they lacked the key residue Ser in the canonical catalytic triads, indicating the presence of a brand new unknown fibrinolytic mechanism” was removed. Please see page 6, lines 84.

c) “Next, an additional 17 candidate snFPITE genes were identified in the assembled genome with 2 more putative pseudogenes.” was revised to “Next, an additional 16 candidate snFPITE genes were identified in the assembled genome with one more putative pseudogenes.” Please see page 19, lines 344-345.

d) “lacking the key amino acid Ser189 in the canonical catalytic triads of known serine proteases”, “This striking phenomenon motivated us to examine”, “snFPITE-p1 (without canonical catalytic triad) and two others normal snFPITE genes (snFPITE-8 and snFPITE-15 with canonical catalytic triad)”, “with canonical catalytic triad) expressed in the Tn-baculovirus system”, “To our best knowledge, snFPITE-p1 therefore might be the first fibrinolytic enzyme without any canonical catalytic triad structure^{26,27}, providing a novel insight into the structural requirement of the catalytic core of serine proteases.” were removed or modified. Please see 21, lines 383-396.

e) “Ser189 in the canonical catalytic triads”, “a unique serine fibrinolytic enzyme might exist without the traditional catalytic triad structure. This finding was supported by a previous study on trypsin⁴⁴. Although mutating Ser195 in trypsin significantly reduced its activity, water could substitute for the missing functional group⁴⁴. The catalytic triad was not the sole source of the catalytic power of a protease as trypsin could still exhibit some activity even when the catalytic triad was disabled⁴⁵. This discovery was further supported by a latest report on a well-established adenylate kinase that was shortened to functional polypeptides (P-loop prototypes) without the canonical catalytic core⁴⁶. Such an enzyme that achieved this without the canonical catalytic core suggested alternative mechanisms of catalysis, potentially offering new pathways for enzyme design and biotechnological applications.” were removed or

modified. Please see 25, lines 496.

f) Revised the figure 7c and d. Please see page 43, lines 691-693; page 44, line 700.

g) Revised the methods. Please see the page 64, line 1093.

h) Revised the supplementary figure 14. Please see the page 79, lines 1365-1376.

i) Revised the supplementary figure 15. Please see the page 80, lines 1379-1383.

Response to Reviewer#1

Dear Reviewer,

We sincerely appreciate your extremely valuable evaluation, positive comments and constructive suggestions. All the comments have been carefully addressed. Changes were highlighted in yellow in the revised manuscript.

Overall comments:

Most of the comments have been addressed by Ma et al. in their 3rd revision, but a major remaining point is the kinetics assays for K_m and V_{max} , that have still not been performed correctly. If the authors decide to keep this information in the paper, they need to revise the methodology and data. However, because the authors seem to find these assays confusing, they could choose to omit these kinetics data (and its associated sections) from the paper altogether.

Note the assays for K_i are correct and can remain in the paper.

Response:

Thank you very much for the positive assessment on our work. Your suggestions are scientific and extremely valuable. On one hand, the snFPITE described here as a novel fibrinolytic agent. Despite snFPITE displayed plasminogen-activating and direct fibrin(ogen)-lysing activities, its working mechanism differs from the well-known tPA (represents the plasminogen-activating agents) and plasmin (represents the direct fibrin(ogen)-lysing agents). Therefore, it was inappropriate to use the tPA reporter peptide (ab108905) and plasmin reporter peptide (ab273301) to calculate snFPITE's kinetic parameters K_m and V_{max} . On the other hand, as you have reminded, it is too complex and difficult to generate a snFPITE specific fluorescent or colorimetric reporter which can reflect the plasminogen-activating and direct fibrin(ogen)-lysing activities of snFPITE precisely and timely. Thus, we choose to omit these kinetics data (and its associated sections) from the paper altogether according to your suggestion.

The kinetics data of the K_i remained in the paper.

Comment 1:

Since snFPITE is NOT described as a tPA activator, what is the logic of using a tPA reporter peptide (ab108905) to analyse enzyme kinetic parameters of snFPITE towards plasminogen? The data has little relevance to the paper and should be omitted.

Response:

Thanks. We have omitted this part from the paper.

Comment 2:

The plasmin reporter peptide (ab273301) is a good choice to assess the enzyme kinetic parameters (K_m , V_{max}) of snFPITE towards PLASMINOGEN (but not towards fibrin(ogen)). Nevertheless, as explained in the previous revisions, it is incorrect to vary the concentration of the reporter peptide in these assays. Instead, the concentrations of the activating enzyme (tPA or snFPITE; low nM) AND the reporter peptide (1 mM) should be fixed, and the plasminogen levels vary (e.g., from 0.2-2 μ M). Please also note that t-PA will not work properly without its fibrin cofactor (or soluble CNBr-digested fibrinogen), so it is better to compare snFPITE to uPA (urokinase) in this assay

Response:

Thank you for your valuable suggestions. As mentioned above, we choose to omit these kinetics data (and its associated sections) from the paper altogether.

Comment 3:

To measure the enzyme kinetic parameters (K_m , V_{max}) of snFPITE towards fibrin(ogen), the authors could use a fluorescent fibrinogen reporter at varying concentrations (where the fluorescence is quenched before cleavage). After direct incubation of this substrate with a fixed concentration of snFPITE (and plasmin as control; low nM), the rates in these simple reactions would be the first derivatives (slopes) of the linear graphs generated within the early minutes (~15 min post-initiation). The slopes should then be plotted against the activator concentrations to calculate the K_m and V_{max} . Since it could be challenging to source the quenched fluorescent fibrinogen, a possible alternative could be DQ gelatin (ThermoFisher Scientific #D12054)

Response:

Thanks for your wonderful suggestions. As mentioned before, we choose to omit these kinetics data (and its associated sections) from the paper altogether according to your suggestion.

Comment 4:

Supp Fig. 4: For clarity, please add the molecular weight marker, align the 3 panels according to the molecular weights and annotate which of the bands is Flaa, and which is snFPITE. The almost identical molecular weights of these agents make these gels confusing. Also, what is the higher MW band in the snFPITE alone lane in the middle panel? This band appears on both the snFPITE alone lane and snFPITE with plasminogen lane.

Response:

The molecular weight marker has been added. The bands of Flaa, snFPITE, plasmin, and plasminogen have been annotated. The middle and right panel have been aligned as they both were native-PAGE based. The left panel remained separated as it was SDS-PAGE based.

Please see the refined Supp Fig. 4.

The question why a new higher MW band in the snFPITE was exhibited in the middle panel. Because this band did not present in the SDS-PAGE experiment (left panel), while it displayed at higher position in the native-PAGE experiment (middle panel), it might be another form of snFPITE under different experimental conditions. In our previous protein crystallization experiments, we also found snFPITE could form a dimer in certain circumstances (snFPITE-n2 in Fig. 5b). Therefore, this higher band might be a dimer of snFPITE. When compared the middle panel with right panel, while this snFPITE dimer might be unable to lyse fibrin in the native-PAGE circumstance.

Comment 5:

Lines 755-756, 792-793: replace “Niego’s method” to “the method described by Niego et al.”.

Response:

Thanks a lot. Done. Please see page 46, line 747, 748.

Comment 6:

The Methods section should be organised in a more logical way to create a better flow and prevent repetition (e.g., place the “Kinetics analysis” section next to the “Fibrin(ogen) lysis analysis”, and ensure that the methods are edited to avoid unnecessary repetition).

Response:

Thanks for your suggestion.

Corrected . Please see page 46, lines 746-749; page 47, lines 750-761.

Comment 7:

New sentence in line 438: omit the coma after ‘snFPITE’.

Response:

Done.

Please see page 23, line 430.

Comment 8:

Line 132: change “protective influence” to “pro-fibrinolytic influence”; Line 135: change “improved” to “prolonged” (because in healthy mice, it is unclear if the haemostatic effect of snFPITE is actually beneficial).

Response:

Corrected.

Please see page 8, lines 126, 129.

We sincerely appreciate for your significantly contribution to enhancing the quality of our work.

Kind regards

Ruian Xu, PhD

Response to Reviewer#4

Dear Reviewer,

We really appreciate your extremely valuable evaluation, positive comments and constructive suggestions.

Overall comments:

This manuscript describes the discovery and characterization of a plasminogen-independent thrombolytic protease(snFPITE) isolated from *Sipunculus nudus*. The authors provide evidence that the enzyme hydrolyzes fibrin and generates a novel, smaller fibrinolytic-active agent from plasminogen. Specific comments concerning the enzymology are listed below.

Response:

Thank you very much for the positive assessment on our work. All the comments have been carefully addressed. Changes were highlighted in yellow in the revised manuscript.

Comment 1:

Figure 1 E, F and Suppl Figure 1 B, C. It is not clear why V_{max} is reported here rather than k_{cat} . As seen in Fig. 1 B, C, the enzyme has been purified. If the purified enzyme is used, it would be helpful to calculate k_{cat} , which is much more instructive than V_{max} .

Response:

Thanks for the wonderful suggestions. K_{cat} is much more instructive than V_{max} . On one hand, this kinetic parameter could be converted from the V_{max} and K_m values with kinetic equations. On the other hand, other reviewers have suggested that it had better to omit the kinetic data from the paper. Because the snFPITE described here was a novel fibrinolytic agent. Despite snFPITE displayed both plasminogen-activating and direct fibrin(ogen)-lysing activities, its working mechanism differs from the well-known tPA (represents the plasminogen-activating agents) and plasmin (represents the direct fibrin(ogen)-lysing agents). Therefore, it was unsuitable to use

the tPA reporter peptide (ab108905) and plasmin reporter peptide (ab273301) to calculate the kinetic parameters K_m and V_{max} . Meanwhile, it is too complex and difficult to generate a snFPITE specific fluorescent or colorimetric reporter which can reflect the plasminogen-activating and direct fibrin(ogen)-lysing activities of snFPITE precisely and timely. Thus, we choose to omit these kinetics data (and its associated sections) from the paper altogether.

Comment 2:

The methods section describing the enzyme kinetics results in Fig. 1 B, C are incomplete. It is assumed that the purified enzyme was used, but this is not explicitly stated. Also, what concentration of purified enzyme was used? What buffer and pH was used for the reaction?

Response:

Thank you very much for the scientific and valuable suggestions. As mentioned above, we choose to omit these kinetics data (and its associated sections) from the paper altogether.

Comment 3:

Supplemental Fig. 9 A, B. It is not clear how the K_i is determined since there does not seem to be any dose response in the presence of inhibitor. The units for K_i are not given. Is it ng/ul or a μM value? It would be better if all values were given in molar units so the reader does not have to calculate it. The model used for fitting should be provided rather than referencing a software package.

Response:

Thanks for your wonderful suggestions.

Yes, the fitting lines in K_i calculation did not vary much. Because the K_i value of PAI1 towards snFPITE was large, which implies that the activity of snFPITE would not change significantly with PAI1, especially under low dose experimental conditions. This weak inhibition effects of PAI1 on snFPITE were also substantiated by our molecular modeling of the snFPITE crystal structure with PAI1 (supplementary figure 9d). The method of K_i determination was described in the “Kinetics analysis” section. Please see the page 46, lines 746-749; page 47, lines 750-761.

The K_i was recalculated using μM as its unit. The model used for fitting also indicated in the figure legends. Please see the revised Supplementary Fig. 9 A, B.

Comment 4:

Lines 316-317 states, “However, the catalytic Ser189 of snFPITE-n1 was found not to form a covalent bond with the Arg376 of A2AP, indicating a different manner of interaction”. Since A2AP is a substrate (Fig. 9C), the covalent acyl-enzyme intermediate may not be detected in the time frame of the experiment because it is deacylated to form the product. This does not imply a different manner of interaction but rather simply that A2AP is being turned over. However, this is also a hypothesis and the data presented does not allow mechanistic interpretation.

Response:

Thank you for the wonderful suggestions. The hypothesis and mechanistic interpretation of A2AP have been omitted from the paper. Please see page 17, lines 308-310.

Comment 5:

Supplemental Figures 5A and 11. It would be helpful if the catalytic triad Asp, His, Ser could be labeled directly in the sequence alignment. The numbering scheme in Fig S11 does not place a serine at position 189, which makes it difficult to identify the triad residues.

Response:

Thanks a lot. The catalytic triad Asp, His, Ser of snFPITE-n1 and snFPITE-n2 have been labeled directly in the sequence alignment. Please see revised Supplemental Figure 5A. The snFPITE1-15 in Supplemental Figure 11 have been aligned again with a new numbering scheme. However, because snFPITE 1-15 sequences were identified by genome sequencing or cloning, we could not get their accurate structural information as snFPITE-n1 and snFPITE-n2 that crystallized by us. Therefore, it is inaccurate to specify their catalytic Asp, His, Ser only by sequence alignment or structural analysis with software at this stage. Please see revised Supplemental Figure 11.

We sincerely appreciate for your significantly contribution to enhancing the quality of our work.

Kind regards

Ruian Xu

Most of the comments have been addressed by Ma et al. in their 3rd revision, but a major remaining point is the kinetics assays for K_m and V_{max} , that have still not been performed correctly. If the authors decide to keep this information in the paper, they need to revise the methodology and data. However, because the authors seem to find these assays confusing, they could choose to omit these kinetics data (and its associated sections) from the paper altogether.

Note the assays for K_i are correct and can remain in the paper.

Major comments:

1. Since snFPITE is NOT described as a tPA activator, what is the logic of using a tPA reporter peptide (ab108905) to analyse enzyme kinetic parameters of snFPITE towards plasminogen? The data has little relevance to the paper and should be omitted.
2. The plasmin reporter peptide (ab273301) is a good choice to assess the enzyme kinetic parameters (K_m , V_{max}) of snFPITE towards PLASMINOGEN (but not towards fibrin(ogen)). Nevertheless, as explained in the previous revisions, it is incorrect to vary the concentration of the reporter peptide in these assays. Instead, the concentrations of the activating enzyme (tPA or snFPITE; low nM) AND the reporter peptide (1 mM) should be fixed, and the plasminogen levels vary (e.g., from 0.2-2 μ M). Please also note that t-PA will not work properly without its fibrin cofactor (or soluble CNBr-digested fibrinogen), so it is better to compare snFPITE to uPA (urokinase) in this assay.
3. To measure the enzyme kinetic parameters (K_m , V_{max}) of snFPITE towards fibrin(ogen), the authors could use a fluorescent fibrinogen reporter at varying concentrations (where the fluorescence is quenched before cleavage). After direct incubation of this substrate with a fixed concentration of snFPITE (and plasmin as control; low nM), the rates in these simple reactions would be the first derivatives (slopes) of the linear graphs generated within the early minutes (~15 min post-initiation). The slopes should then be plotted against the activator concentrations to calculate the K_m and V_{max} . Since it could be challenging to source the quenched fluorescent fibrinogen, a possible alternative could be DQ gelatin (ThermoFisher Scientific #D12054).
4. Supp Fig. 4: For clarity, please add the molecular weight marker, align the 3 panels according to the molecular weights and annotate which of the bands is Flaa, and which is snFPITE. The almost identical molecular weights of these agents make these gels confusing. Also, what is the higher MW band in the snFPITE alone lane in the middle panel? This band appears on both the snFPITE alone lane and snFPITE with plasminogen lane.

Additional comments:

5. Lines 755-756, 792-793: replace "Niego's method" to "the method described by Niego et al.".
6. The Methods section should be organised in a more logical way to create a better flow and prevent repetition (e.g., place the "Kinetics analysis" section next to the "Fibrin(ogen) lysis analysis", and ensure that the methods are edited to avoid unnecessary repetition).

7. New sentence in line 438: omit the coma after 'snFPITE'.
8. Line 132: change "protective influence" to "pro-fibrinolytic influence"; Line 135: change "improved" to "prolonged" (because in healthy mice, it is unclear if the haemostatic effect of snFPITE is actually beneficial).